# PyramidKV: Dynamic KV Cache Compression based on Pyramidal Information Funneling

## Abstract

In this study, we investigate whether attention-based information flow inside large language models (LLMs) is aggregated through noticeable patterns for long context processing. Our observations reveal that LLMs aggregate information through **Pyramidal** Information Funneling where attention is scattering widely in lower layers, progressively consolidating within specific contexts, and ultimately focusing on critical tokens (a.k.a massive activation or attention sink) in higher layers. Motivated by these insights, we developed PyramidKV, a novel and effective KV cache compression method. This approach dynamically adjusts the KV cache size across different layers, allocating more cache in lower layers and less in higher ones, diverging from traditional methods that maintain a uniform KV cache size. Our experimental evaluations, utilizing the LongBench benchmark, show that PyramidKV matches the performance of models with a full KV cache while retaining only 12% of the KV cache, thus significantly reducing memory usage. In scenarios emphasizing memory efficiency, where only 0.7% of the KV cache is maintained, PyramidKV surpasses other KV cache compression techniques, achieving up to a 20.5 absolute accuracy improvement on TREC dataset. In the Needle-in-a-Haystack experiment, PyramidKV outperforms competing methods in maintaining long-context comprehension in LLMs; notably, retaining just 128 KV cache entries enables the LLAMA-3-70B model to achieve 100.0 Acc. performance, matching that of a full KV cache.

## 1 Introduction

Large language models (LLMs) (Achiam et al., 2023; Touvron et al., 2023a;b; Jiang et al., 2023) are integral to various natural language processing applications, including dialogue systems (Chiang et al., 2023), document summarization (Fabbri et al., 2019a), and code completion (Roziere et al., 2023). These models have recently been scaled up to handle long contexts (Fu et al., 2024; Ding et al., 2024; Zhu et al., 2023; Chen et al., 2023), with GPT-4 processing up to 128K tokens and Gemini-pro-1.5 handling 1M tokens. However, scaling LLMs to extremely long contexts naturally leads to a significant delay due to the quadratic computation of attention over long contexts. A common solution to mitigate such inference delays involves caching the key and value states (KV) of previous tokens (Waddington et al., 2013), with the trade-off of requiring extensive GPU memory storage. For instance, maintaining a KV cache for 100K tokens in LLaMA-2 7B requires over 50GB of memory, while a 2K context requires less than 1GB of memory (Wu et al., 2024).

To tackle these memory constraints, recent studies have explored the optimization of KV caching, including approaches such as low-rank decomposition of the KV cache (Dong et al., 2024) or pruning non-essential KV cache (Zhang et al., 2024; Li et al., 2024; Ge et al., 2023). Notably, it has been shown that maintaining merely 20% of the KV cache can preserve a substantial level of performance (Zhang et al., 2024). Moreover, extreme compression of the KV cache for tasks of longer contexts (e.g., retrieval augmented generation or RAG for short) can drastically improve efficiency and further reduce resource use. However, questions about the universal applicability of these strategies across all layers of an LLM remain open. (1) *Are these KV cache strategies applicable to all layers?* (2) *Is it computationally efficient to use the same KV cache size across layers as previous studies have done?* These considerations suggest a need for an in-depth, more nuanced understanding of KV cache optimization in LLMs.

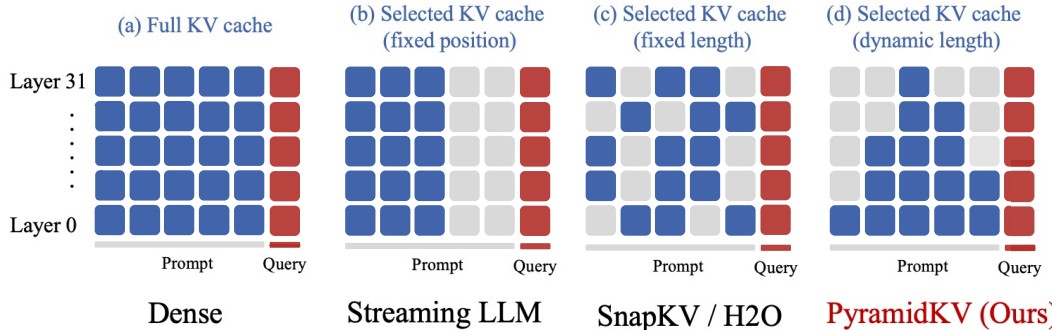

Figure 1: Illustration of PyramidKV compared with existing KV cache compression methods. (a) Full KV has all tokens stored in the KV cache in each layer; cache size increases as the input length increases. (b) StreamingLLM (Xiao et al., 2023) only keeps few initial tokens with a fixed cache size in each layer. (c) SnapKV (Li et al., 2024) and H2O (Zhang et al., 2024) keep a fixed cache size across Transformer layers, and their selection is based on the attention score. (d) PyramidKV maintains pyramid-like cache sizes, allocating more cache budget to lower layers and less to higher layers. This approach to KV cache selection better aligns with the increasing attention sparsity observed in multi-layer Transformers (§3).

To examine these questions, we aim to systematically investigate the design principles of the KV cache compression across different layers, specifically tailored to the behaviors of the attention mechanism. We first investigate how information flow is aggregated via attention mechanisms across different layers in multi-document question answering (QA), a classic task involving long contexts. Our analysis identifies a notable transition of attention distribution from a broad coverage of global contexts to a narrow focus of local tokens over layers in LLMs. This pattern suggests an aggregated information flow where information is initially gathered broadly and subsequently narrowed down to key tokens, epitomizing the massive attention phenomenon. Our findings provide unique insights beyond the previously documented "massive activation" (Sun et al., 2024) that very few activations exhibit significantly larger values than others when calculating multi-head attention in LLMs and "attention sink" (Xiao et al., 2023) that keeping the KV of initial tokens will largely recover the performance of window attention.

Building on these insights on how information flows are aggregated through a pyramid pattern, we design a novel and effective KV cache pruning approach that mirrors the geometric shape, named PyramidKV. As shown in Figure 1, unlike the fixed-and-same length KV cache pruning common in prior works (Zhang et al., 2024; Ge et al., 2023; Li et al., 2024), PyramidKV allocates more KV cache to the lower layers where information is more dispersed and each KV holds less information while reducing the KV cache in higher layers where information becomes concentrated in fewer key tokens. To the best of our knowledge, PyramidKV is the first KV cache compression method with varied cache retention across layers, tailoring cache amounts to the informational needs of each layer and paving the way for future research.

We conducted comprehensive experiments on LongBench (Bai et al., 2023) using 17 datasets across various tasks and domains with three backbone models (LLaMa-3-8B-Instruct, LLaMa-3-70B-Instruct and Mistral-7B (Jiang et al., 2023)). The results show that PyramidKV preserves performance using just 12.0% of the KV cache (KV Cache size = 2048) on the LongBench benchmark and significantly outperforms other methods in extreme conditions, retaining only 0.7% of the KV cache. Moreover, PyramidKV outperforms baseline models (H2O (Zhang et al., 2024), SnapKV (Li et al., 2024), StreamingLLM (Xiao et al., 2023)) across all tested cache sizes (64, 96, 128, 256), with its advantages most pronounced at smaller cache sizes. In the Needle In A Haystack experiment, PyramidKV effectively maintains the long-context comprehension in LLMs, outperforming than competing methods. Remarkably, with PyramidKV, retaining only 128 KV cache entries allows the LLaMa-3-70B-Instruct model to achieve 100.0 Acc. performance, matching the performance of a full KV cache.

## 2 RELATED WORK

**Interpretation of LLMs** Prior research has shown that attention matrices in LLMs are typically sparse (Chen et al., 2024a; Xiao et al., 2023; Zhang et al., 2024), focusing disproportionately on a few tokens. For instance, Xiao et al. (2023) identified an "attention sink" phenomenon, where maintaining the Key and Value (KV) states of the first few tokens can substantially restore the performance of windowed attention, despite these tokens not being semantically crucial. Similarly, Sun et al. (2024) identified a "massive activations" pattern, where a minority of activations show significantly larger values than others within LLMs. Interestingly, these values remain relatively constant across different inputs and act as critical bias terms in the model.

Further explorations in this field reveal distinct patterns across various attention heads and layers. Li et al. (2024) observed that certain attention heads consistently target specific prompt attention features during decoding. Additionally, Wang et al. (2023) discovered that in In-Context Learning scenarios, label words in demonstration examples serve as semantic anchors. In the lower layers of an LLM, shallow semantic information coalesces around these label words, which subsequently guide the LLMs' final output predictions by serving as reference points. Recently, Wu et al. (2024) revealed that a special type of attention head, the so-called retrieval head, is largely responsible for retrieving information. Inspired by these findings that the attention mechanism exhibits varying behaviors across different layers, we discovered that "Massive Activation" does not consistently manifest across all layers in long context sequences; instead, it predominantly occurs in the upper layers. Additionally, we identified a novel trend of information aggregation specific to long-context inputs, which will be further explained in §3.

**KV Cache Compression** There has been a growing interest in addressing LLMs' memory constraints on processing long context inputs. FastGen (Ge et al., 2023) introduces an adaptive KV cache management strategy that optimizes memory use by tailoring retention tactics to the specific nature of attention heads. This method involves evicting long-range contexts from heads that prioritize local interactions, discarding non-special tokens from heads focused on special tokens, and maintaining a standard KV cache for heads that engage broadly across tokens. SnapKV (Li et al., 2024) improves efficiency by compressing KV caches via selecting/clustering significant KV positions based on their attention scores. Heavy Hitter Oracle (H2O) (Zhang et al., 2024) implements a dynamic eviction policy that effectively balances the retention of recent and historically significant tokens, optimizing memory usage while preserving essential information. StreamingLLM (Xiao et al., 2023) enables LLMs trained on finite attention windows to handle infinite sequence lengths without fine-tuning, thus expanding the models' applicability to broader contexts. LM-Infinite (Han et al., 2023) allows LLMs pre-trained with 2K or 4K-long segments to generalize to up to 200M length inputs while retaining perplexity without parameter updates.

While these approaches have significantly advanced the efficient management of memory for LLMs, they generally apply a fixed KV cache size across all layers. In contrast, our investigations into the attention mechanisms across different layers of LLMs reveal that the attention patterns vary from layer to layer, making a one-size-fits-all approach to KV cache management suboptimal. In response to this inefficiency, we propose a novel KV cache compression method, called PyramidKV that allocates different KV cache budgets across different layers, tailored to the unique demands and operational logic of each layer's attention mechanism. This layer-specific strategy takes a significant step toward balancing both memory efficiency and model performance, addressing a key limitation in existing methodologies.

## 3 PYRAMIDAL INFORMATION FUNNELING

To systematically understand the attention mechanism over layers in LLMs for long-context inputs, we conduct a fine-grained study focusing on the multi-document question answering (QA) task. The model is presented with multiple interrelated documents and prompted to generate an answer for the given query. The main target is to investigate how the model aggregates dispersed information within these retrieved documents for accurate responses.

In particular, we focus on our analysis of the LLaMa (Touvron et al., 2023a;b) and visualize the distribution and behavior of attention scores over layers. To assess the distinct behaviors of each

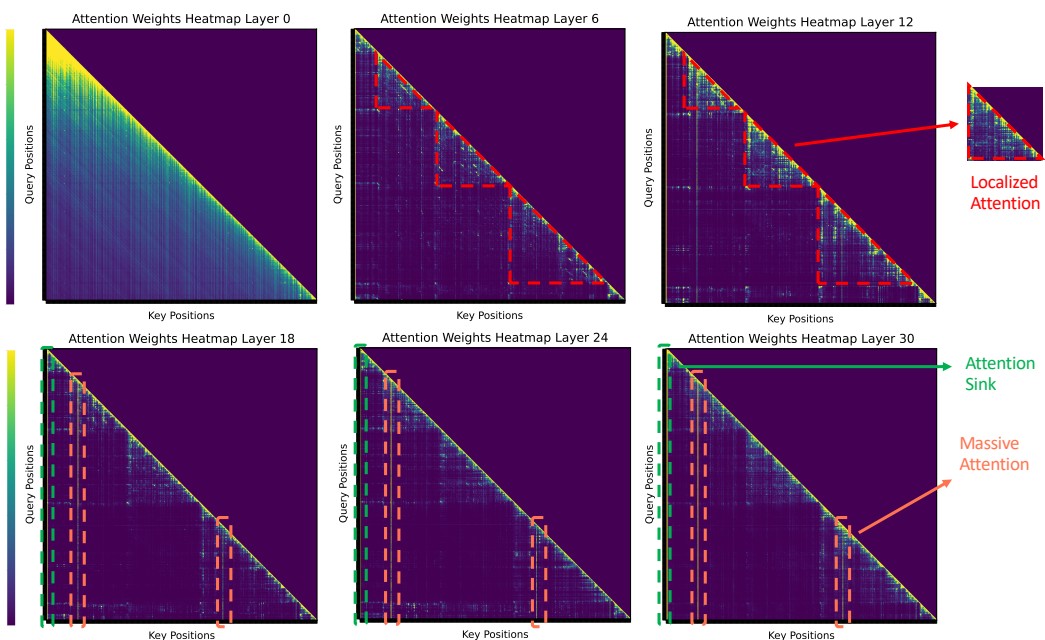

Figure 2: Attention patterns of retrieval-augmented generation across layers in LlaMa (Touvron et al., 2023a;b) reveal that in the lower layers, the model exhibits a broad-spectrum mode of attention, distributing attention scores uniformly across all content. In the middle layers, attention becomes more localized within each document, indicating refined information aggregation (dotted red triangular shapes in layers 6 and 10). This culminates in the upper layers, where "massive attention" focuses on a few key tokens (concentrated attention bars after layer 18), efficiently extracting essential information for answers.

multi-head self-attention layer, we compute the average attention from all heads within each layer. Figure 2 shows the attention patterns of one QA example over six different layers (i.e., 0, 6, 12, 18, 24, and 30).

We identify an approximately uniform distribution of attention scores from the lower layers (e.g., the 0th layer). This suggests that the model operates in a broad-spectrum mode at the lower layers, aggregating information globally from all available content without prioritizing its attention on specific input segments. Notably, a distinct transition to a more localized attention pattern within each document emerges, as the model progresses to encode information at the middle layers (6th to 18th layers). In this phase, attention is predominantly directed towards tokens within the same document, suggesting a more refined aggregation of information within individual contexts.

This trend continues and intensifies in the upper layers (from the 24th to the 30th layer), where we observed the emergence of 'massive attention' phenomena. In these layers, the attention mechanism concentrates overwhelmingly on a few key tokens. This pattern of attention allocation, where extremely high attention scores are registered, signifies that the model has aggregated the essential information into these focal tokens. Such behavior underscores a sophisticated mechanism by which LLMs manage and streamline complex and voluminous information, culminating in the efficient extraction of the most pertinent data points necessary for generating accurate answers.

## 4 PYRAMIDKV

### 4.1 PRELIMINARIES AND PROBLEM FORMULATION

In an autoregressive transformer-based LLM, the generation of the $i$-th token requires that the attention module computes the query, key, and value vectors for all previous $i - 1$ tokens. To speed

up the inference process and avoid duplicate computations, the key and value matrices are typically stored in the GPU memory. While the KV cache enhances inference speed and reduces redundant computations, it can consume significant memory when dealing with long input contexts. To optimize memory usage, a strategy called KV cache compression is proposed (Zhang et al., 2024; Xiao et al., 2023; Li et al., 2024), which involves retaining only a minimal amount of KV cache while preserving as much information as possible.

In a language model with $m$ transformer layers, we denote the key and value matrices in the $l$-th attention layer respectively as $\boldsymbol{K}^l, \boldsymbol{V}^l \in \mathbb{R}^{n \times d}, \forall l \in [0, m-1]$ when encoding a sequence of $n$ tokens. The goal of KV cache compression is to seek two sub-matrices $\boldsymbol{K}_s^l, \boldsymbol{V}_s^l \in \mathbb{R}^{k^l \times d}$ from the full matrices $\boldsymbol{K}^l$ and $\boldsymbol{V}^l$, given a cache budget $k^l < n$ for each layer $l \in [0, m-1]$ while maximizing performance preservation. That is, a language model with KV cache compression only uses $\boldsymbol{K}_s^l$ and $\boldsymbol{V}_s^l$ in the GPU memory for inference on a dataset $\mathcal{D}$, and obtains a similar result to a full model according to an evaluation scoring metric, i.e., $\text{score}(\boldsymbol{K}^l, \boldsymbol{V}^l, \mathcal{D}) \approx \text{score}(\boldsymbol{K}_s^l, \boldsymbol{V}_s^l, \mathcal{D})$.

## 4.2 PROPOSED METHOD

In this section, we introduce our method, PyramidKV, based on the pyramidal information funneling observed across different layers in §3. PyramidKV consists of two steps: (1) Dynamically allocating different KV cache sizes/budgets across different layers (§4.2.1); and (2) Selecting important KV vectors in each attention head for caching (§4.2.2).

### 4.2.1 KV CACHE SIZE/BUDGET ALLOCATION

Previous work on KV cache compression (Li et al., 2024; Zhang et al., 2024; Xiao et al., 2023) often allocates a fixed KV cache size across LLM layers. However, as our analysis in §3 demonstrates, attention patterns are not identical across different layers. Particularly dense attention is observed in the lower layers, and sparse attention in higher layers. Therefore, using a fixed KV cache size across different layers may lead to suboptimal performance. These approaches may retain many unimportant tokens in the higher layers of sparser attentions while potentially overlooking many crucial tokens in the lower layers of denser attentions.

Thus, we propose to increase compression efficiency by dynamically allocating the cache budgets across layers to reflect the aggregated information flow based on attention patterns. Specifically, PyramidKV allocates more KV cache to the lower layers where information is more dispersed and each KV state contains less information, while reducing the KV cache in higher layers where information becomes concentrated in a few key tokens.

Following the common practice in KV cache compression (Li et al., 2024; Xiao et al., 2023), we first retain the KV cache for the last $\alpha$ tokens of the input across all layers, as these tokens have been shown to contain the most immediate task-related information, where $\alpha$ is a hyperparameter, controlling the number of last few tokens being included in the KV cache. For simplicity, we call these tokens "*instruction tokens*", which is also referred to as "*local window*" in previous literature (Zhang et al., 2024; Li et al., 2024; Xiao et al., 2023).

Subsequently, given the remaining total cache budget $k^{\text{total}} = \sum_{l \in [0, m-1]} k^l$ that can be used over all transformer layers (noted as $m$), we first determine the cache sizes for the top and bottom layers, and use an arithmetic sequence to compute the cache sizes for the intermediate layers to form the pyramidal shape. The key intuition is to follow the attention pattern in aggregated information flow, reflecting a monotonically decreasing pattern of important tokens for attention from lower layers to upper layers. We allocate $k^{m-1} = k^{\text{total}}/(\beta \cdot m)$ for the top layer and $k^0 = (2 \cdot k^{\text{total}})/m - k^{m-1}$ for the bottom layer,, where $\beta$ is a hyperparameter to adjust the pyramid's shape. The hyperparameter $\beta$ is still required to determine the top layer. Once the top layer is identified, the budget of the bottom layer can be calculated by summing the budgets across all layers and equating this sum to the total budget. Once the cache sizes of the bottom and top layers are determined, the cache sizes for all intermediate layers are set according to an arithmetic sequence, defined as

$$k^l = k^0 - \frac{k^0 - k^{m-1}}{m-1} \times l. \tag{1}$$

### 4.2.2 KV Cache Selection

Once the KV cache budget is determined for each layer, our method needs to select specific KV states for caching within each layer in LLMs. As described in the previous section, the KV cache of the last $\alpha$ tokens, referred to as instruction tokens, are retained across all layers. Following SnapKV (Li et al., 2024), the selection of the remaining tokens is then guided by the attention scores derived from these instruction tokens—tokens receiving higher attention scores are deemed more relevant to the generation process and are thus their KV states are prioritized for retention in the GPU cache.

In a typical LLM, the attention mechanism in each head $h$ is calculated using the formula:

$$\boldsymbol{A}^h = \text{softmax}(\boldsymbol{Q}^h \cdot (\boldsymbol{K}^h)^\top / \sqrt{d_k}), \tag{2}$$

where $d_k$ denotes the dimension of the key vectors. Following (Li et al., 2024), we utilize a pooling layer at $\boldsymbol{A}^h$ to avoid the risk of being misled by some massive activation scores.

To quantify the importance of each token during the generation process, we measure the level of attention each token receives from the instruction tokens, and use this measurement to select important tokens for KV caching. Specifically, we compute the score of selecting $i$-th token for retention in the KV cache as $s_i^h$ in each attention head $h$ by:

$$s_i^h = \sum_{j \in [n-\alpha, n]} \boldsymbol{A}_{ij}^h \tag{3}$$

where $[n - \alpha, n]$ is the range of the instruction tokens. In each layer $l$ and for each head $h$, the top $k^l$ tokens with the highest scores are selected, and their respective KV caches are retained. All other KV caches are discarded and will not be utilized in any subsequent computations throughout the generation process.

## 5 Experiment

We conduct comprehensive experiments to evaluate the effectiveness of PyramidKV on performance preserving and memory reduction. First, we introduce the experiment setup (§5.1) as backbone LLMs (§5.1.1), the evaluation datasets (§5.1.2), and the baselines in comparison (§5.1.3). Next, we report the performance in a memory-oriented scenario and a performance-oriented scenario experiments in §5.2. We also do some insight experiment (§5.3) to test if the model can preserve the performance on long-context inputs on the Needle-in-the-haystack experiment (§5.3.1). Finally, we discuss the trade-off between memory, time, and performance in §5.3.2.

### 5.1 Experiment Setup

We maintain a fixed constant KV cache size for each layer for the baseline methods. In contrast, PyramidKV employs varying KV cache sizes across different layers. To ensure a fair comparison, we adjusted the average KV cache size in PyramidKV to match that of the baseline models, to keep the total memory consumption of all methods the same. In our experiment, we set $\beta = 20$ and $\alpha = 8$. We use the same prompt for each dataset in all the experiments.

### 5.1.1 Backbone LLMs

We compare PyramidKV against baselines using state-of-the-art open-sourced LLMs, namely LLaMa-3-8B-Instruct, Mistral-7B-Instruct (Jiang et al., 2023) and LLaMa-3-70B-Instruct. Testing examples are evaluated in a generative format, with answers generated by greedy decoding across all tasks to ensure a fair comparison.

### 5.1.2 Datasets

We use LongBench (Bai et al., 2023) to assess the performance of PyramidKV on tasks involving long-context inputs. LongBench is a meticulously designed benchmark suite that tests the capabilities

of language models in handling extended documents and complex information sequences. This benchmark was created for comprehensive multi-task evaluation of long context inputs. It includes 17 datasets covering tasks such as single-document QA (Kočiský et al., 2018; Dasigi et al., 2021), multi-document QA (Yang et al., 2018; Ho et al., 2020), summarization (Huang et al., 2021; Zhong et al., 2021; Fabbri et al., 2019b), few-shot learning (Li and Roth, 2002; Gliwa et al., 2019; Joshi et al., 2017), synthetic, and code generation (Guo et al., 2023; Liu et al., 2023b). The datasets feature an average input length ranging from 1,235 to 18,409 tokens (detailed average lengths can be found in Table 1), necessitating substantial memory for KV cache management. For all these tasks, we adhered to the standard metrics recommended by LongBench (Bai et al., 2023) (i.e., F1 for QA, Rouge-L for summarization, Acc. for synthetic and Edit Sim. for code generation.) We refer readers to more details at Appendix E.

### 5.1.3 BASELINES

We compare PyramidKV with three baselines, all of which keep the same KV cache size across different layers, with different strategies for KV cache selection.

**StreamingLLM (SLM)** (Xiao et al., 2023) is an efficient framework that enables LLMs trained with a finite length attention window to generalize to infinite sequence length without any fine-tuning. They propose StreamingLLM based on the attention sink phenomenon that keeping the KV of the first few tokens will largely recover the performance of window attention. StreamingLLM is a competitive method to solve long-context tasks. In our experiments, to be consistent with other methods, we simply keep the KV cache of the last $\alpha$ tokens and the first $k - \alpha$ tokens, as suggested in the paper.

**Heavy Hitter Oracle (H2O)** (Zhang et al., 2024) is a KV cache compression policy that dynamically retains a balance of recent and Heavy Hitter (H2) tokens. H2O keeps a fixed cache size of the Key and Value matrix across Transformer layers. The selection process for the KV cache is driven by attention scores, specifically utilizing the average attention scores from all queries across all tokens to guide the selection.

**SnapKV (SKV)** (Li et al., 2024) automatically compresses KV caches by selecting clustered important tokens for each attention head. This method discerns the attention patterns of the Key and Value matrices using a localized observation window positioned at the end of the prompts. However, unlike H2O, SnapKV employs a more nuanced clustering algorithm that includes a pooling layer. Additionally, SnapKV captures attention signals using patterns from a localized window (Instruction Tokens), rather than aggregating attention across all queries, allowing for more targeted and efficient compression.

**FullKV (FKV)** caches all keys and values for each input token in each layer. All methods are compared to the FullKV simultaneously.

## 5.2 MAIN RESULTS

The evaluation results from LongBench (Bai et al., 2023) are shown in Table 1 and Figure 3. In Figure 3, we report the average score across datasets for 64, 96, 128, and 256 case sizes. In Table 1, we report the results for two different KV cache sizes with 64 and 2048. These two sizes represent two distinct operational scenarios—the memory-efficient scenario and the performance-preserving scenario, respectively for a trade-off between memory and model performance. In Appendix M, we report the results of KV cache sizes with 64, 96, 128 and 2048.

Overall, PyramidKV preserves the performance with only 12% of the KV cache and it consistently surpasses other method across a range of KV cache sizes and different backbone models, with its performance advantages becoming particularly pronounced in memory-constrained environments where only about 0.8% of the KV cache from the prompt is retained. Upon examining specific tasks, PyramidKV demonstrates a notably superior performance on the TREC task, a few-shot question answering challenge. This suggests that the model effectively aggregates information from the few-shot examples, highlighting the potential for further investigation into in-context learning tasks.

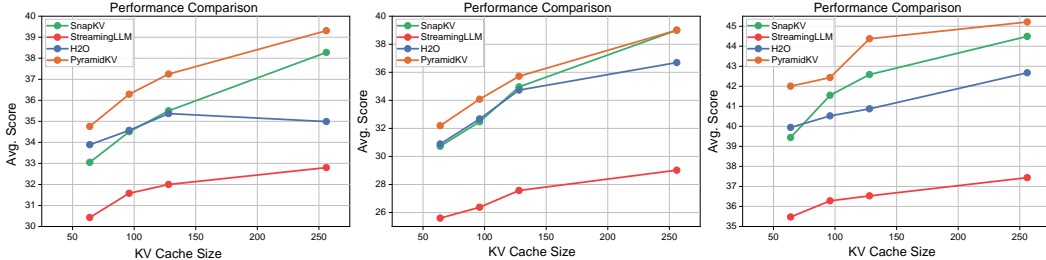

Figure 3: The evaluation results from LongBench (Bai et al., 2023) across 64, 96, 128 and 256 cache sizes at LLaMa-3-8B-Instruct (Left), Mistral-7B-Instruct (Middle) and LLaMa-3-70B-Instruct (Right). The evaluation metrics are the average score of LongBench across datasets. PyramidKV outperforms H2O (Zhang et al., 2024), SnapKV (Li et al., 2024) and StreamingLLM (Xiao et al., 2023), especially in small KV cache sizes.

| Method | Single-Document QA | | | Multi-Document QA | | | Summarization | | | Few-shot Learning | | | Synthetic | | Code | | |
| | NrtvQA | Qasper | MF-en | HotpotQA | 2WikiMQA | Musique | GovReport | QMSum | MultiNews | TREC | TriviaQA | SAMSum | PCount | PRe | Lcc | RB-P | Avg. |
|---|---|---|---|---|---|---|---|---|---|---|---|---|---|---|---|---|---|
| | 18409 | 3619 | 4559 | 9151 | 4887 | 11214 | 8734 | 10614 | 2113 | 5177 | 8209 | 6258 | 11141 | 9289 | 1235 | 4206 | |
| | | | | | | LLaMa-3-8B-Instruct, KV Size = Full | | | | | | | | | | | |
| FKV | 25.70 | 29.75 | 41.12 | 45.55 | 35.87 | 22.35 | 25.63 | 23.03 | 26.21 | 73.00 | 90.56 | 41.88 | 4.67 | 69.25 | 58.05 | 50.77 | 41.46 |
| | | | | | | LLaMa-3-8B-Instruct, KV Size = 64 | | | | | | | | | | | |
| SKV | 19.86 | 9.09 | 27.89 | **37.34** | 28.35 | **18.17** | 15.86 | 20.80 | 16.41 | 38.50 | 85.92 | 36.32 | 5.22 | 69.00 | 51.78 | 48.38 | 33.05 |
| H2O | 20.80 | 11.34 | 27.03 | 37.25 | 30.01 | 17.94 | 18.29 | 21.49 | 19.13 | 38.00 | 84.70 | **37.76** | **5.63** | 69.33 | **53.44** | **50.15** | 33.89 |
| SLM | 17.44 | 8.68 | 22.25 | 35.37 | **31.51** | 15.97 | 15.46 | 20.06 | 14.64 | 38.00 | 72.33 | 29.10 | 5.42 | **69.50** | 46.14 | 45.09 | 30.43 |
| Ours | **21.13** | **14.18** | **30.26** | 35.12 | 23.76 | 16.17 | **18.33** | **21.65** | **19.23** | **58.00** | **88.31** | 37.07 | 5.23 | **69.50** | 52.61 | 45.74 | **34.76** |
| | | | | | | LLaMa-3-8B-Instruct, KV Size = 2048 | | | | | | | | | | | |
| SKV | **25.86** | 29.55 | **41.10** | 44.99 | **35.80** | 21.81 | 25.98 | **23.40** | 26.46 | **73.50** | 90.56 | 41.66 | 5.17 | **69.25** | 56.65 | 49.94 | 41.35 |
| SLM | 21.71 | 25.78 | 38.13 | 40.12 | 32.01 | 16.86 | 23.14 | 22.64 | **26.48** | 70.00 | 83.22 | 31.75 | **5.74** | 68.50 | 53.50 | 45.58 | 37.82 |
| H2O | 25.56 | 26.85 | 39.54 | 44.30 | 32.92 | 21.09 | 24.68 | 23.01 | 26.16 | 53.00 | 90.56 | 41.84 | 4.91 | **69.25** | 56.40 | 49.68 | 39.35 |
| Ours | 25.40 | **29.71** | 40.25 | 44.76 | 35.32 | **21.98** | **26.83** | 23.30 | 26.19 | 73.00 | 90.56 | **42.14** | 5.22 | **69.25** | **58.76** | **51.18** | **41.49** |
| | | | | | | Mistral-7B-Instruct, KV Size = Full | | | | | | | | | | | |
| FKV | 26.90 | 33.07 | 49.20 | 43.02 | 27.33 | 18.78 | 32.91 | 24.21 | 26.99 | 71.00 | 86.23 | 42.65 | 2.75 | 86.98 | 56.96 | 54.52 | 42.71 |
| | | | | | | Mistral-7B-Instruct, KV Size = 64 | | | | | | | | | | | |
| SKV | 16.94 | 17.17 | 39.51 | **36.87** | 22.26 | 15.18 | 14.75 | 20.35 | 21.45 | 37.50 | **84.16** | 37.28 | 4.50 | 61.13 | 42.40 | 38.44 | 30.72 |
| SLM | 15.01 | 13.84 | 28.74 | 30.97 | **24.50** | 13.42 | 13.25 | 19.46 | 19.17 | 35.50 | 76.91 | 29.61 | 4.67 | 27.33 | 38.71 | 35.29 | 25.60 |
| H2O | 18.19 | 19.04 | 37.40 | 30.18 | 22.22 | 13.77 | 16.60 | **21.52** | **21.98** | 37.00 | 81.02 | **38.62** | **5.00** | **66.03** | 43.54 | **40.46** | 30.88 |
| Ours | **20.91** | **20.21** | **39.94** | 33.57 | 22.87 | **15.70** | **17.31** | 21.23 | 21.41 | **54.00** | 81.98 | 36.96 | 3.58 | 60.83 | **44.52** | 37.99 | **32.19** |
| | | | | | | Mistral-7B-Instruct, KV Size = 2048 | | | | | | | | | | | |
| SKV | **25.89** | **32.93** | 48.56 | **42.96** | 27.42 | 19.02 | 26.56 | **24.47** | 26.69 | 70.00 | 86.27 | 42.57 | **5.50** | **88.90** | 50.42 | 46.72 | 41.56 |
| SLM | 20.31 | 26.64 | 45.72 | 35.25 | 24.31 | 12.20 | **27.47** | 21.57 | 24.51 | 68.50 | 71.95 | 31.19 | 5.00 | 22.56 | 43.38 | 37.08 | 32.35 |
| H2O | 25.76 | 31.10 | **49.03** | 40.76 | 26.52 | 17.07 | 24.81 | 23.64 | 26.60 | 55.00 | **86.35** | 42.48 | **5.50** | 88.15 | 49.93 | 46.57 | 39.95 |
| Ours | 25.53 | 32.21 | 48.97 | 42.26 | **27.50** | **19.36** | 26.60 | 23.97 | **26.73** | **71.00** | 86.25 | **42.94** | 4.50 | 87.90 | **53.12** | **47.21** | **41.63** |
| | | | | | | LLaMa-3-70B-Instruct, KV Size = Full | | | | | | | | | | | |
| FKV | 27.75 | 46.48 | 49.45 | 52.04 | 54.90 | 30.42 | 32.37 | 22.27 | 27.58 | 73.50 | 92.46 | 45.73 | 12.50 | 72.50 | 40.96 | 63.91 | 46.55 |
| | | | | | | LLaMa-3-70B-Instruct, KV Size = 64 | | | | | | | | | | | |
| SKV | 23.92 | 31.09 | 36.54 | 46.66 | 50.40 | 25.30 | 18.05 | 21.11 | 19.79 | 41.50 | **91.06** | 40.26 | 12.00 | **72.50** | 43.33 | 57.62 | 39.45 |
| SLM | 22.07 | 23.53 | 27.31 | 43.21 | **51.66** | 23.85 | 16.62 | 19.74 | 15.20 | 39.50 | 76.89 | 33.06 | 12.00 | **72.50** | 40.23 | 50.20 | 35.47 |
| H2O | 25.45 | 34.64 | 33.23 | **48.25** | 50.30 | 24.88 | 20.03 | 21.50 | 21.39 | 42.00 | 90.36 | **41.58** | 12.00 | 71.50 | 43.83 | **58.16** | 39.94 |
| Ours | **25.47** | **36.71** | **42.29** | 47.08 | 46.21 | **28.30** | 20.60 | **21.62** | 21.62 | **64.50** | 89.61 | 41.28 | 12.50 | **72.50** | **45.34** | 56.50 | **42.01** |
| | | | | | | LLaMa-3-70B-Instruct, KV Size = 2048 | | | | | | | | | | | |
| SKV | 26.73 | 45.18 | 47.91 | **52.00** | **55.24** | 30.48 | 28.76 | 22.35 | 27.31 | 72.50 | 92.38 | 45.58 | 12.00 | **72.50** | 41.52 | **69.27** | 46.36 |
| SLM | 26.69 | 41.01 | 35.97 | 46.55 | 52.98 | 25.71 | 27.81 | 20.81 | 27.16 | 69.00 | 91.55 | 44.02 | 12.00 | 72.00 | 41.44 | 68.73 | 43.96 |
| H2O | **27.67** | **46.51** | **49.54** | 51.49 | 53.85 | 29.97 | 28.57 | **22.79** | **27.53** | 59.00 | **92.63** | **45.94** | 12.00 | **72.50** | 41.39 | 63.90 | 45.33 |
| Ours | 27.22 | 46.19 | 48.72 | 51.62 | 54.56 | **31.11** | **29.76** | 22.50 | 27.27 | **73.50** | 91.88 | 45.47 | 12.00 | **72.50** | 41.36 | 69.12 | **46.55** |

Table 1: Performance comparison of PyramidKV (Ours) with SnapKV (SKV), H2O, StreamingLLM (SLM) and FullKV (FKV) on LongBench for LLaMa-3-8B-Instruct, Mistral-7B-Instruct and LLaMa-3-70B-Instruct. PyramidKV generally outperforms other KV Cache compression methods across various KV Cache sizes and LLMs. The performance strengths of PyramidKV are more evident in small KV Cache sizes (i.e. KV Size = 64). Bold text represents the best performance.

Notably, we initially observe the pyramidal attention patterns from the visualization analysis on the multi-document QA task (Figure 2), but the pyramid heuristic has demonstrated its effectiveness on a range of other LongBench tasks (e.g., single-document QA, In-Context Learning), suggesting its promising generalizability beyond multi-document QA.

The performance advantage of PyramidKV increases as the KV cache memory decreases. By focusing on optimizing budget allocation across layers, PyramidKV accurately allocates resources in memory-constrained scenarios, ensuring that retained information is effectively preserved to maintain model performance. Moreover, as in long bench results shown in Table 1, even in the performance-preserving scenario (i.e., KV cache size = 2048), PyramidKV still improves the performance over baseline methods and even outperforms the performance of FullKV.

Among the 16 datasets, the tasks where our proposed method performs slightly worse than the baseline are mostly saturated (e.g., HotpotQA, Musique, etc under the LlaMa-3-8B-Instruct setting with KV Size = 64, as shown in Table 1). In these cases, our method is only marginally inferior to the baseline and remains competitive. Conversely, on tasks with greater potential for improvement (e.g., Qasper, MF-en, TREC, TriviaQA, etc under the same setting), our method significantly outperforms the baseline. Consequently, the overall average performance of our method surpasses that of the baselines. Notably, these tasks include several In-Context Learning tasks (i.e., TREC), our method enjoys best performance gain at In-Context Learning tasks, where the method demonstrates the ability to leverage provided examples to adapt effectively. The importance of In-Context Learning tasks lies in their widespread use in real-world applications, where dynamic adaptation to new inputs is critical.

### 5.3 DISCUSSION AND INSIGHTS

#### 5.3.1 PYRAMIDKV PRESERVES THE LONG-CONTEXT UNDERSTANDING ABILITY

We conduct the "Fact Retrieval Across Context Lengths" (Needle In A Haystack) experiment (Liu et al., 2023a; Fu et al., 2024), which is a dataset designed to test whether a model can find key information in long input sequences, to evaluate the in-context retrieval capabilities of LLMs when utilizing various KV cache compression methods. For this purpose, we employ **LlaMa-3-70B-Instruct** as our base, with context lengths extending up to 8k. We compared several KV cache compression techniques (PyramidKV, SnapKV (Li et al., 2024), and H2O (Zhang et al., 2024)) at cache sizes of **128** and full cache. The results, presented in Figure 4 [1]. The results demonstrate that with only 128 KV cache retained, PyramidKV effectively maintains the model's ability to understand short contexts, and shows only modest degradation for longer contexts. In contrast, other KV cache compression methods significantly hinder the performance of LLMs. Notably, for the larger model (**LlaMa-3-70B-Instruct**), PyramidKV achieves 100.0 Acc. performance, matching the results of FullKV, thereby demonstrating its ability to preserve long-context comprehension with a substantially reduced KV cache. We adopt the haystack setting of haystack formed from a long corpus for the Needle In A Haystack task as Wu et al. (2024).

#### 5.3.2 PYRAMIDKV SIGNIFICANTLY REDUCES MEMORY WITH LIMITED PERFORMANCE DROP

In this section, we study how sensitive the methods are with different sizes of KV cache. We report the KV cache memory reduction in Table 2. We evaluate the memory consumption of LLaMa-3-8B-Instruct. Specifically, we evaluate the memory consumption of all methods with a fixed batch size of 1, a sequence length of 8192, and model weights in fp16 format. We observe that PyramidKV substantially reduces the KV cache memory across different numbers of cache sizes.

We also present that the allocation strategy and score-based selection add minimal complexity in the inference phase compared to the computation required for next-token predictions as Appendix K.

## 6 CONCLUSION

In this study, we investigate the intrinsic attention patterns of Large Language Models (LLMs) when processing long context inputs. Our empirical analysis leads us to discover the existence of Pyramidal Information Funneling. Motivated by this discovery, we design a novel KV cache

---

[1]Additional results with 64, 96 and 128 KV cache sizes with **LlaMa-3-8B-Instruct** at 8k context length, **LlaMa-3-70B-Instruct** at 8k context length, and **Mistral-7B-Instruct** (Jiang et al., 2023) at 32k context length are available in Appendix O

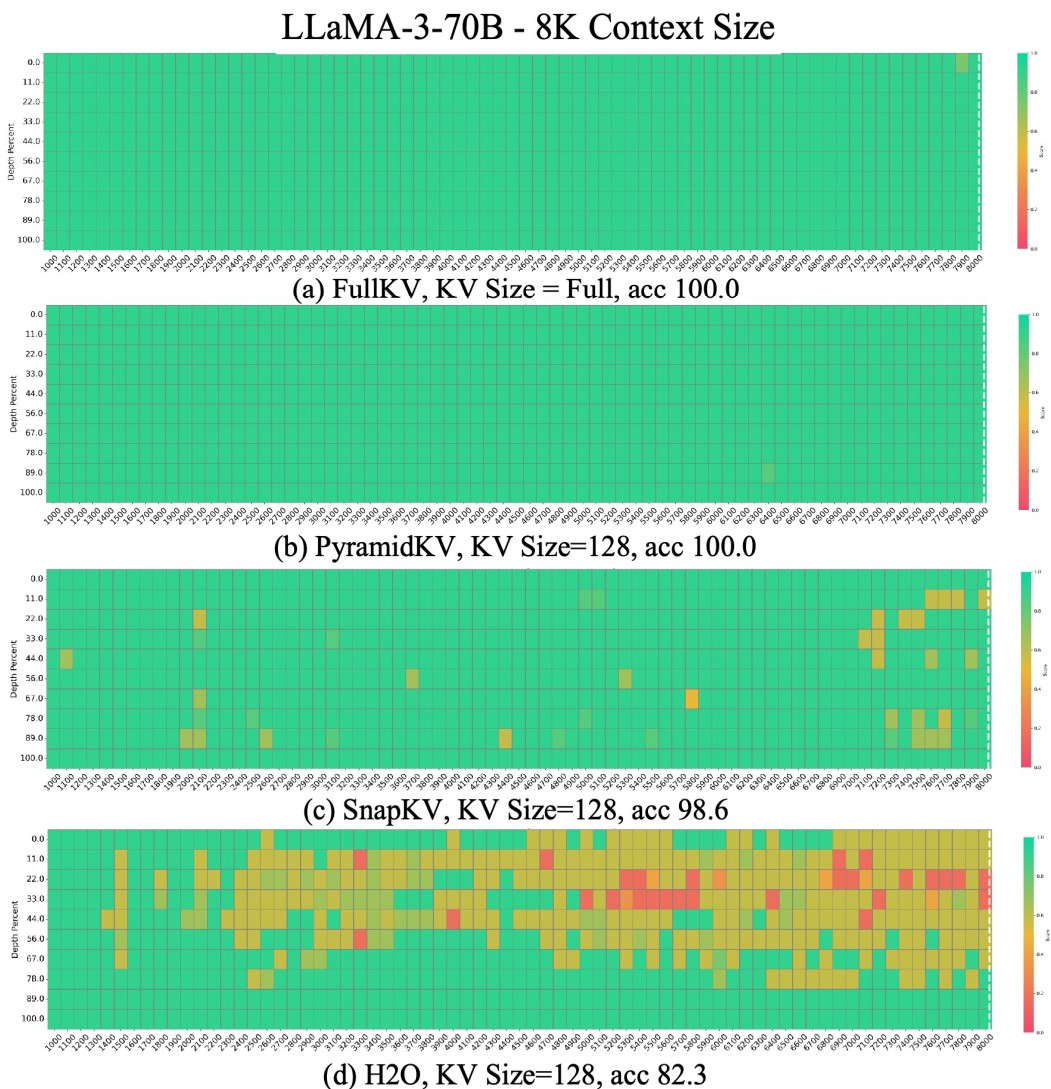

Figure 4: Results of the Fact Retrieval Across Context Lengths ("Needle In A HayStack") test in **LlaMa-3-70B-Instruct** with **8k** context size in **128** KV cache size. The vertical axis of the table represents the depth percentage, and the horizontal axis represents the token length.

| cache size | Memory | Compression Ratio | QMSum | TREC | TriviaQA | PCount | PRe | Lcc |
|------------|--------|-------------------|-------|------|----------|--------|-------|-------|
| 512 | 428M | 6.3% | 22.80 | 71.50 | 90.61 | 5.91 | 69.50 | 58.16 |
| 1024 | 856M | 12.5% | 22.55 | 71.50 | 90.61 | 5.91 | 69.50 | 58.16 |
| 2048 | 1712M | 25.0% | 22.55 | 72.00 | 90.56 | 5.58 | 69.25 | 56.79 |
| Full | 6848M | 100.0% | 23.30 | 73.00 | 90.56 | 5.22 | 69.25 | 58.76 |

Table 2: Memory reduction effect and benchmark result by using PyramidKV. We conducted a comparison of memory consumption between the Llama-3-8B-Instruct model utilizing the Full KV cache and the Llama-3-8B-Instruct model compressed with the PyramidKV.

compression approach PyramidKV that utilizes this information flow pattern. By leveraging the Pyramidal Information Funneling into KV cache compression design, our method excels in memory-constrained settings, preserves long-context understanding ability, and significantly reduces memory usage with minimal performance trade-offs compared to the baselines.

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

# A    LIMITATIONS

Our experiments were limited to three base models: LLAMA-3-8B-Instruct, LLAMA-3-70B-Instruct and Mistral-7B-Instruct. While these models demonstrated consistent trends, the robustness of our findings could be enhanced by testing a broader array of model families, should resources permit. Additionally, our research was conducted exclusively in English, with no investigations into how these findings might be transferred to other languages. Expanding the linguistic scope of our experiments could provide a more comprehensive understanding of the applicability of our results globally. Based on our results at LongBench and Needle-in-a-HayStack experiment, PyramidKV generally works decently in most of the language tasks (i.e., Single-Document QA, Multi-Document QA, Summerization, Few-Shot In-Context Learning, etc.). Although we observe that PyramidKV performs better in some tasks (i.e., Few-Shot In-Context Learning) compared with some other tasks (i.e., Summerization), we have not observed cases that the decoding result collapses at some tasks. This remains a new topic for future work to explore.

# B    FUTURE WORK

Our investigation on PyramidKV highlights considerable opportunities for optimizing KV cache compression by adjusting the number of KV caches retained according to the distinct attention patterns of each layer (or even for each head). For instance, the retention of KV cache for each layer could be dynamically modified based on real-time analysis of the attention matrices, ensuring that the compression strategy is consistently aligned with the changing attention dynamics within LLMs. Furthermore, our experiments indicate that PyramidKV significantly surpasses other methods in few-shot learning tasks, suggesting promising applications of KV cache in in-context learning. This approach could potentially enable the use of more shots within constrained memory limits.

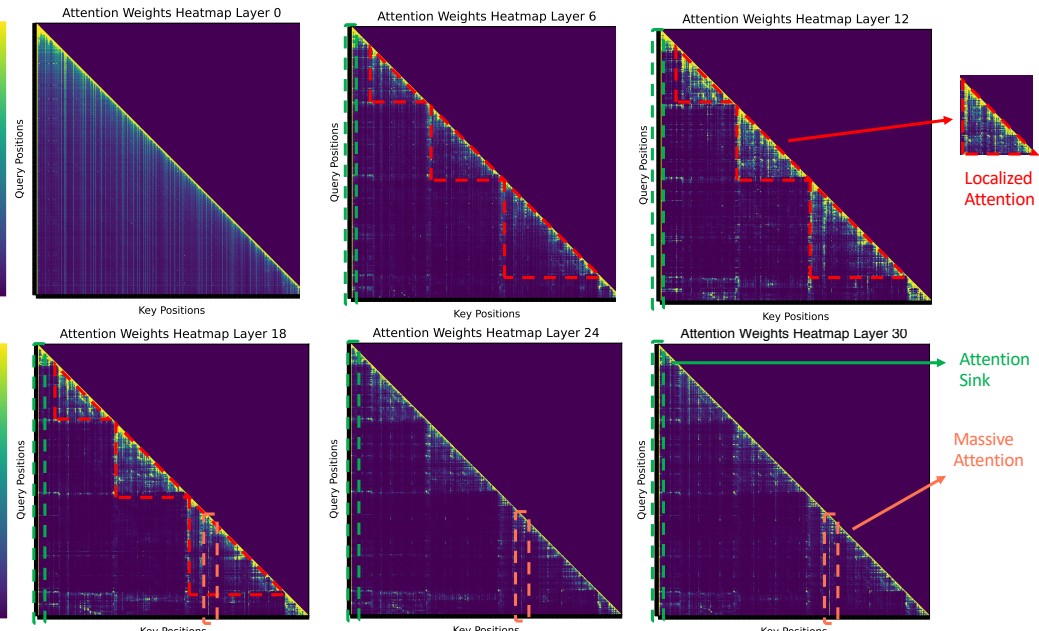

Figure 5: Attention patterns of retrieval-augmented generation across layers in Mistral-7B-Instruct model (Jiang et al., 2023)

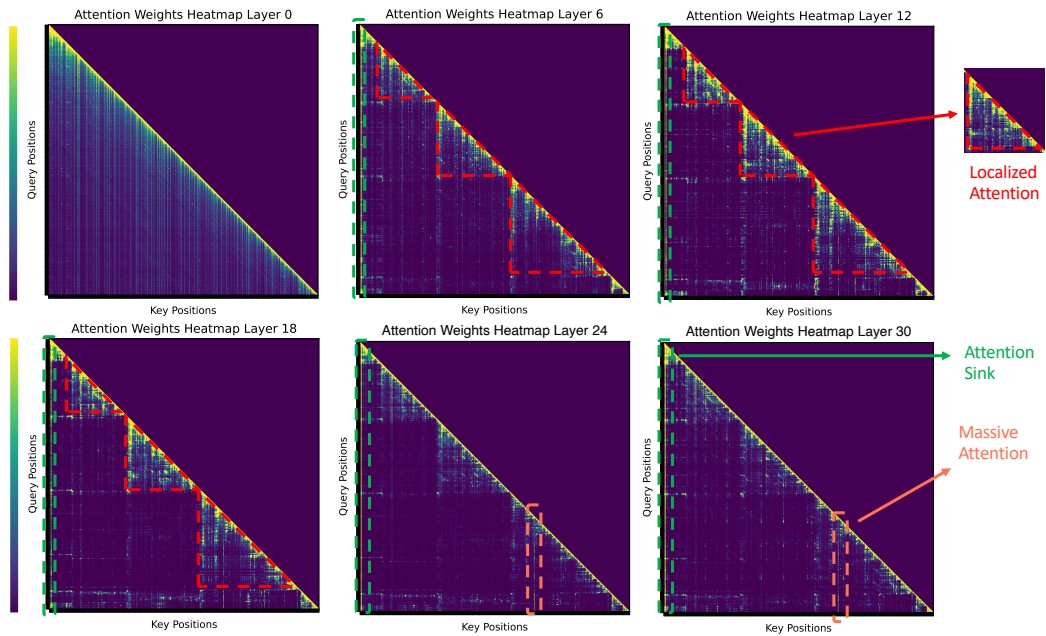

Figure 6: Attention patterns of retrieval-augmented generation across layers in Mixtral-8x7B-Instruct Mixture-of-Experts model.

## C PYRAMIDAL INFORMATION FUNNELING

Figure 5 and Figure 6 shows the attention patterns of one QA example over six different layers (i.e., 0, 6, 12, 18, 24, and 30) for Mistral-7B-Instruct model and Mixtral-8x7B-Instruct Mixture-of-Experts model. Figure 5 and Figure 6 demonstrate that the Pyramidal Information Funneling phenomenon is also evident in both the Mistral model and Mixtral model . The results reveal that, akin to Llama-like models, Mistral exhibit a progressively narrowing attention focus across layers. This supports the universality of the Pyramidal Information Funneling phenomenon across diverse model families. We hope this addresses your concern and underscores the generalizability of our findings.

Our analysis uniquely examines attention metrics across all transformer layers, from 0 to 30, leading to the discovery of a key phenomenon we term Pyramidal Information Funneling.

 Lee et al. (2024) conducted a limited investigation into attention patterns, focusing only on the lower layer (layer 0) and a single upper layer (layer 18). While  Lee et al. (2024) noted that attention becomes more skewed in upper layers, it did not provide a fine-grained observation of attention patterns across all layers. In contrast, our study reveals several novel findings:

- **Localized Attention**: We observe that attention progressively narrows its focus, targeting specific components within the input sequence.
- **Massive Attention Mechanism**: In the upper layers, attention heavily concentrates on a small set of critical tokens. Notably, these tokens are not limited to the leading positions, as observed in  Lee et al. (2024), but also appear at regular intervals across the sequence. The discrepancy arises from differences in input settings, with  Lee et al. (2024) identifying massive attention only at the initial tokens.

These insights motivated us to propose a token-selection method based on the highest attention scores in the upper layers, rather than solely relying on tokens from earlier positions.

To the best of our knowledge,  Chen et al. (2024b) has not analyzed attention patterns across transformer layers.

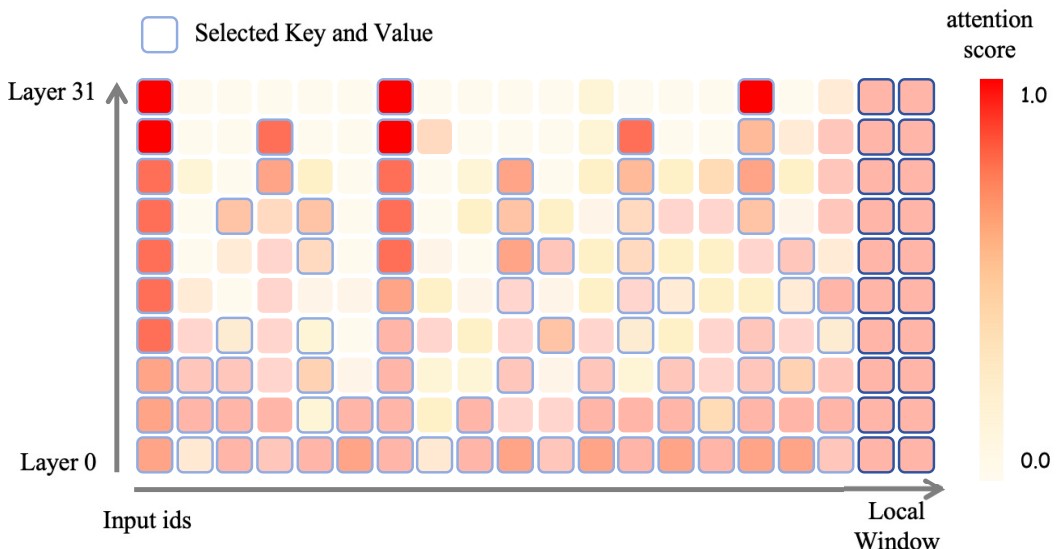

Figure 7: Illustration of PyramidKV. At the lower level of the transformer, the PyramidKV selects more keys and values based on the exhibited average attention pattern. Fewer keys and values at the higher level are selected based on the massive activation pattern, where we observe that attention scores are concentrated over local regions.

Therefore, although Lee et al. (2024) and Chen et al. (2024b) are considered contemporaneous with our work, making a comparison unnecessary, the perspective of our observation is considered novel compared with Lee et al. (2024) and Chen et al. (2024b). Moreover, although Lee et al. (2024) also observed attention patterns, the method we proposed based on our observations is significantly different from Lee et al. (2024), further highlighting the novelty of our work.

# D  DETAILS OF PROPOSED METHOD

Based on the pyramidal information funneling observed across different layers, PyramidKV consists of two steps: (1) Dynamically allocating different KV cache sizes/budgets across different layers; and (2) Selecting important KV vectors in each attention head for caching as Figure 7.

Our decision to use an arithmetic sequence is driven by three key factors:

- **Alignment with Pyramidal Information Funneling Pattern**: Empirical observations reveal a pyramidal information funneling pattern, where lower layers exhibit dispersed attention while higher layers concentrate on fewer tokens. Inspired by this, we adopt the arithmetic sequence design to align with this natural progression.

- **Superior Empirical Performance**: Through extensive experimentation across diverse datasets, we compared various methods, including the arithmetic sequence and adaptive approaches. Results consistently showed that the arithmetic sequence method outperformed others.

- **Computational Efficiency**: The arithmetic sequence method introduces minimal computational overhead compared to adaptive approaches, which require dynamically computing cache budgets across layers.

To perform KV cache eviction, we use torch.gather. Below, we outline the memory allocation and release process of torch.gather:

- **Index Selection**: Identify the positions of the elements to extract from the input tensor.

- **Memory Location Calculation**: Compute the specific memory locations of the elements to be extracted using the strides of the input tensor across each dimension.

- **Output Tensor Creation**: Allocate memory to create a new output tensor and copy the selected elements to their corresponding positions in the output tensor.

- **Memory Management**: Since torch.gather is not an in-place operation, it creates a new tensor to store the results, while the memory of the original input tensor is released.

The speed-up offered by PyramidKV is complementary to that achieved through tensor parallelism and pipeline parallelism, as these approaches are not mutually exclusive. PyramidKV can be seamlessly integrated with both tensor parallelism and pipeline parallelism.

## E  DETAILS OF EVALUATION

We use LongBench (Bai et al., 2023) to assess the performance of PyramidKV on tasks involving long-context inputs. LongBench is a meticulously designed benchmark suite that tests the capabilities of language models in handling extended documents and complex information sequences. This benchmark was created for multi-task evaluation of long context inputs.

We present the details of metrics, language and data for LongBench at Table 3.

We run all the experiments on NVIDIA A100.

| Dataset | Source | Avg len | Metric | Language | #data |
|---|---|---|---|---|---|
| *Single-Document QA* | | | | | |
| NarrativeQA | Literature, Film | 18,409 | F1 | English | 200 |
| Qasper | Science | 3,619 | F1 | English | 200 |
| MultiFieldQA-en | Multi-field | 4,559 | F1 | English | 150 |
| *Multi-Document QA* | | | | | |
| HotpotQA | Wikipedia | 9,151 | F1 | English | 200 |
| 2WikiMultihopQA | Wikipedia | 4,887 | F1 | English | 200 |
| MuSiQue | Wikipedia | 11,214 | F1 | English | 200 |
| *Summarization* | | | | | |
| GovReport | Government report | 8,734 | Rouge-L | English | 200 |
| QMSum | Meeting | 10,614 | Rouge-L | English | 200 |
| MultiNews | News | 2,113 | Rouge-L | English | 200 |
| *Few-shot Learning* | | | | | |
| TREC | Web question | 5,177 | Accuracy (CLS) | English | 200 |
| TriviaQA | Wikipedia, Web | 8,209 | F1 | English | 200 |
| SAMSum | Dialogue | 6,258 | Rouge-L | English | 200 |
| *Synthetic Task* | | | | | |
| PassageCount | Wikipedia | 11,141 | Accuracy (EM) | English | 200 |
| PassageRetrieval-en | Wikipedia | 9,289 | Accuracy (EM) | English | 200 |
| *Code Completion* | | | | | |
| LCC | Github | 1,235 | Edit Sim | Python/C#/Java | 500 |
| RepoBench-P | Github repository | 4,206 | Edit Sim | Python/Java | 500 |

Table 3: An overview of the dataset statistics in LongBench (Bai et al., 2023). 'Source' denotes the origin of the context. 'Accuracy (CLS)' refers to classification accuracy, while 'Accuracy (EM)' refers to exact match accuracy.

## F  LICENSE

LongBench: MIT

# G   HANDLE ROTARY EMBEDDING AFTER TOKENS ARE REMOVED IN PYRAMIDKV

We keep the rotary embedding unchanged after tokens are removed, so that LLMs can still capture the exact position information even if the tokens are removed. StreamingLLM (Xiao et al., 2023) shows that rolling kv cache with the correct relative position is crucial for maintaining performance. This is because StreamingLLM is designed to mainly handle unlimited context sizes, where contexts exceed the LLM's fixed context length. Without changing the rotary embedding after token removal, LLMs would receive rotary embedding of a non-monotonic position sequence. For example, after the first KV cache compression, LLMs might receive the input position embedding as $[0, 1, 2, 3, 3096, 3097, \cdots, 4096]$, and the position embedding of the generated sequences could be $[1005, 1006, 1007, \cdots]$. The position sequence of $[0, 1, 2, 3, 3096, \ldots, 4096, 1005, 1006, 1007, \cdots]$ is a non-monotonic sequence, which may negatively hurts the performance. In contrast, our targeting settings will not process unlimited context size. For example, given a input sequence of 4012 length, after KV cache compression, the position sequence would be $[0, 4, 6, 16, \cdots, 3927, 3987, 4012]$, and the position sequence of the generated tokens would be $[4013, 4014, \cdots]$. By keeping the rotary embedding unchanged after the tokens are removed, the LLM avoids non-monotonic position sequences, and the LLM can capture the exact position information even if the tokens are shifted. Our preliminary results show that rolling KV cache with the correct relative position will slightly decrease the performance.

# H   ABLATION STUDY

In this section, we present an ablation study for hyperparameters and allocation strategies.

Based on our observations of the attention pattern, we find that a relatively stable, linear arithmetic decrease aligns more closely with the underlying structure of the pattern. We conduct experiments comparing various allocation strategies.

We conducted hyperparameter testing on the original development sets of 16 datasets in LongBench. The parameter $\beta$ demonstrated remarkable stability, showing minimal sensitivity to varying hyperparameter settings, which highlights its robustness. Conversely, $\alpha$ consistently produced superior results when set to 8 or 16. Consequently, these values were adopted for subsequent experiments. In Appendix H.2 and H.3, we further analyzed the impact of hyperparameter selection on KV cache budget allocation across different layers. The experiments reaffirmed that $\beta$ had negligible influence on the outcomes, underscoring its stability. Meanwhile, $\alpha$ continued to deliver optimal results at values of 8 and 16.

## H.1   ALLOCATION SRATEGIES

Based on our observations of the attention pattern, we find that a relatively stable, linear arithmetic decrease aligns more closely with the underlying structure of the pattern.

We conduct experiments comparing various pyramidal allocation strategies (i.e., linear decay strategy, geometric decay strategy and exponential decay strategy) with a cache size of 64 as Table 4 to confirm that a linear strategy is indeed optimal or preferable.

We also propose three adaptive allocation baselines, which are based on the entropy, Gini coefficient, and sparsity of the attention values at each layer. The weight of each layer is calculated based on its corresponding metric (entropy, Gini coefficient, or sparsity), and the budget is allocated accordingly. Specifically:

- **Entropy-based allocation**: Layers with higher entropy receive higher weights. Each layer's entropy is calculated based on the the layer's attention.

- **Gini coefficient-based allocation**: Layers with higher Gini coefficients receive higher weights. Each layer's Gini coefficient is calculated based on the the layer's attention

The empirical results as Table 4 consistently showed that the linear strategy outperformed its counterparts, establishing it as the most effective approach for our use case. The experiment strengthens the rationale for choosing the specific allocation method.

| Stra. | Single-Document QA | | | Multi-Document QA | | | Summarization | | | Few-shot Learning | | | Synthetic | | Code | | Avg. |
|---|---|---|---|---|---|---|---|---|---|---|---|---|---|---|---|---|---|
| | NrtvQA | Qasper | MF-en | HotpotQA | 2WikiMQA | Musique | GovReport | QMSum | MultiNews | TREC | TriviaQA | SAMSum | PCount | PRe | Lcc | RB-P | |
| Geo. | 20.51 | 15.04 | 29.4 | 34.93 | 26.41 | 16.6 | 18.32 | 21.68 | 18.81 | 52 | 87.51 | 36.15 | 5.18 | 69.17 | 53.11 | 44.91 | 34.36 |
| Exp. | 20.58 | 14.82 | 28.74 | 34.34 | 26.24 | 16.11 | 18.41 | 21.63 | 18.75 | 52.00 | 87.94 | 36.26 | 5.19 | 69.17 | 54.34 | 43.21 | 34.23 |
| Lin. | 21.13 | 14.18 | 30.26 | 35.12 | 23.76 | 16.17 | 18.33 | 21.65 | 19.23 | 58.00 | 88.31 | 37.07 | 5.23 | 69.50 | 52.61 | 45.74 | 34.76 |
| Entropy. | 18.12 | 14.12 | 27.22 | 33.21 | 21.16 | 15.16 | 17.76 | 19.87 | 17.09 | 51 | 87.31 | 34.29 | 5.09 | 68.91 | 50.12 | 42.98 | 32.71 |
| Gini. | 17.92 | 14.61 | 28.21 | 32.67 | 19.98 | 15.98 | 16.20 | 19.29 | 18.21 | 51.00 | 86.21 | 34.97 | 5.11 | 65.51 | 51.98 | 43.37 | 32.58 |

Table 4: Ablation study of allocation strategies.

## H.2 HYPER PARAMETER $\alpha$

We present the study of $\alpha$ for LlaMa-3-8B-Instruct in 128 KV cache size budget at Table 5. We find that a small alpha value (i.e., 8, 16) leads to better performance than a larger alpha value (i.e., 24, 32, 40, 48).

| $\alpha$ | Single-Document QA | | | Multi-Document QA | | | Summarization | | | Few-shot Learning | | | Synthetic | | Code | | Avg. |
|---|---|---|---|---|---|---|---|---|---|---|---|---|---|---|---|---|---|
| | NrtvQA | Qasper | MF-en | HotpotQA | 2WikiMQA | Musique | GovReport | QMSum | MultiNews | TREC | TriviaQA | SAMSum | PCount | PRe | Lcc | RB-P | |
| 8 | 21.40 | 16.92 | 31.62 | 38.45 | 28.72 | 18.59 | 19.96 | 22.49 | 20.96 | 66.50 | 89.35 | 38.43 | 5.92 | 69.00 | 57.86 | 51.80 | 37.37 |
| 16 | 23.37 | 16.21 | 33.93 | 38.24 | 27.28 | 20.57 | 19.71 | 21.93 | 20.86 | 60.00 | 88.75 | 38.34 | 5.48 | 69.12 | 57.84 | 53.42 | 37.19 |
| 24 | 22.85 | 14.51 | 32.26 | 38.38 | 28.36 | 20.33 | 19.55 | 21.72 | 20.72 | 54.50 | 88.71 | 38.46 | 5.48 | 69.50 | 56.83 | 53.65 | 36.61 |
| 32 | 23.01 | 14.54 | 31.68 | 38.86 | 29.90 | 19.16 | 19.20 | 21.83 | 20.52 | 49.50 | 87.01 | 38.01 | 5.75 | 69.50 | 57.02 | 54.54 | 36.25 |
| 40 | 21.70 | 13.06 | 30.14 | 36.78 | 27.34 | 18.88 | 18.72 | 21.37 | 19.79 | 44.00 | 87.74 | 38.43 | 6.08 | 69.25 | 56.11 | 53.89 | 35.21 |
| 48 | 21.51 | 12.30 | 29.77 | 39.04 | 26.76 | 17.97 | 18.65 | 21.20 | 20.29 | 44.50 | 87.73 | 38.44 | 5.51 | 69.25 | 56.73 | 53.88 | 35.22 |

Table 5: Ablation on $\alpha$.

## H.3 HYPER PARAMETER $\beta$

One topic we want to analyze for our ablation study is the selection of $\beta$, which can determine the staircase. The smaller $\beta$ is, the gentler the staircase is; the larger $\beta$ is, the steeper the staircase is. We want to investigate the effect of $\beta$ step size on the final result. Results on 128 KV cache size and LlaMa-3-8B-Instruct are shown in Table 6. The results at Table 6 show that using a relatively small value of $\beta$ yields better outcomes, and PyramidKV is generally robust to the selection of $\beta$.

| $\beta$ | Single-Document QA | | | Multi-Document QA | | | Summarization | | | Few-shot Learning | | | Synthetic | | Code | | Avg. |
|---|---|---|---|---|---|---|---|---|---|---|---|---|---|---|---|---|---|
| | NrtvQA | Qasper | MF-en | HotpotQA | 2WikiMQA | Musique | GovReport | QMSum | MultiNews | TREC | TriviaQA | SAMSum | PCount | PRe | Lcc | RB-P | |
| 20 | 21.40 | 16.92 | 33.79 | 39.73 | 28.72 | 18.59 | 19.86 | 22.48 | 20.95 | 66.50 | 89.35 | 38.39 | 5.92 | 69.00 | 56.49 | 47.95 | 37.25 |
| 18 | 21.71 | 16.24 | 33.59 | 39.89 | 27.94 | 18.38 | 19.76 | 22.32 | 21.20 | 66.50 | 88.98 | 38.93 | 5.46 | 69.50 | 56.47 | 49.23 | 37.25 |
| 16 | 21.74 | 14.86 | 33.64 | 39.18 | 28.17 | 18.77 | 19.57 | 22.25 | 21.48 | 66.50 | 89.69 | 38.87 | 5.82 | 69.50 | 57.02 | 50.11 | 37.32 |
| 14 | 22.53 | 16.31 | 33.50 | 40.50 | 28.15 | 19.26 | 19.66 | 22.39 | 21.38 | 65.50 | 90.02 | 38.56 | 5.75 | 69.50 | 57.51 | 49.71 | 37.51 |

Table 6: Ablation on $\beta$.

## I INTEGATION WITH MINFERENCE

We would like to clarify that PyramidKV and MInference Jiang et al. (2024) are complementary approaches addressing different aspects of KV cache optimization. Specifically:

- MInference focuses on accelerating the generation of KV caches during the pre-filling stage of LLM inference.

- In contrast, PyramidKV targets efficient KV cache management during LLM decoding.

To evaluate their respective strengths, we compared PyramidKV and MInference on Longbench using a KV cache size of 128. The results demonstrated the superior performance of PyramidKV.

Furthermore, we demonstrate that MInference and PyramidKV can be seamlessly integrated to achieve highly efficient inference while maintaining performance comparable to full attention. The results of MInference combined with PyramidKV, evaluated on Longbench with a KV cache size of 128, as PyramidKV + MInference hybrid approach.

| Stra. | Single-Document QA | | | Multi-Document QA | | | Summarization | | | Few-shot Learning | | | Synthetic | | Code | | Avg. |
|---|---|---|---|---|---|---|---|---|---|---|---|---|---|---|---|---|---|
| | NrtvQA | Qasper | MF-en | HotpotQA | 2WikiMQA | Musique | GovReport | QMSum | MultiNews | TREC | TriviaQA | SAMSum | PCount | PRe | Lcc | RB-P | |
| PyramidKV | 23.99 | 20.61 | 38.28 | 43.23 | 31.62 | 20.94 | 21.27 | 22.69 | 22.83 | 71 | 90.48 | 39.86 | 5.83 | 69.25 | 56.94 | 50.16 | 39.31 |
| MInference | 19.74 | 30.63 | 40.41 | 44.28 | 35.22 | 20.65 | 28.43 | 23.35 | 26.75 | 72.00 | 87.90 | 42.78 | 6.30 | 64.00 | 58.76 | 5.06 | 38.86 |
| M. + P. | 20.04 | 31.74 | 39.98 | 43.10 | 35.21 | 21.60 | 27.41 | 23.06 | 26.76 | 73.00 | 88.03 | 43.36 | 6.28 | 64.00 | 58.57 | 45.42 | 40.47 |

Table 7: Comparison between PyramidKV, MInference and MInference-PyramidKV hybrid method.

In summary, we demonstrate that PyramidKV outperforms MInference on Longbench. Furthermore, when integrated with MInference, PyramidKV enhances its performance even further.

## J  COMPARISON WITH PYRAMIDINFER

Our work differs from PyramidInfer in two key aspects:

- **Decay Strategy**: While PyramidInfer Yang et al. (2024) employs a geometric decay strategy, our method adopts an arithmetic decay strategy. We argue that the relatively stable and linear nature of arithmetic decay better aligns with the behavior of the attention mechanism. This strategy is derived from empirically observed attention patterns, aiming to closely match them. Notably, our approach also achieves superior results, as demonstrated in the experimental results presented in the table below.

- **Token Selection**: PyramidInfer discards tokens in earlier layers, preventing them from being reconsidered in later layers. In contrast, our method allows previously discarded tokens to be re-evaluated in higher layers, recognizing that these tokens may still hold relevance at different stages of the model's processing.

- **Pyramidal Information Funneling Pattern**: A key contribution of our work lies in identifying and leveraging the pyramidal information funneling phenomenon within attention mechanisms. Through in-depth analysis, we observe that attention tends to disperse in earlier layers and progressively concentrates on crucial tokens in higher layers. This insight forms the foundation of our arithmetic decay strategy, ensuring that our method aligns more naturally with these intrinsic patterns.

Despite some similarities between the two approaches, these differences lead to significantly distinct outcomes. As shown in Table 8, our method consistently outperforms PyramidInfer, highlighting the effectiveness of our design choices.

| Stra. | Single-Document QA | | | Multi-Document QA | | | Summarization | | | Few-shot Learning | | | Synthetic | | Code | | Avg. |
|---|---|---|---|---|---|---|---|---|---|---|---|---|---|---|---|---|---|
| | NrtvQA | Qasper | MF-en | HotpotQA | 2WikiMQA | Musique | GovReport | QMSum | MultiNews | TREC | TriviaQA | SAMSum | PCount | PRe | Lcc | RB-P | |
| Pyramidinfer | 20.42 | 12.77 | 25.21 | 35.81 | 25.83 | 16.88 | 18.27 | 21.78 | 18.52 | 51.00 | 88.54 | 35.76 | 5.61 | 69.25 | 53.21 | 44.12 | 33.94 |
| PyramidKV | 21.13 | 14.18 | 30.26 | 35.12 | 23.76 | 16.17 | 18.33 | 21.65 | 19.23 | 58.00 | 88.31 | 37.07 | 5.23 | 69.50 | 52.61 | 45.74 | 34.76 |

Table 8: Comparison between PyramidKV and Pyramidinfer.

## K  PYRAMIDKV WILL CAUSE MINIMAL EXTRA INFERENCE OVERHEAD.

The allocation strategy and score-based selection add minimal complexity in the inference phase compared to the computation required for next-token predictions as Table 9. Each row shows the

setting of using a specific "[Prompt length, Generation length]" combination. We show the inference speed comparison between total inference time, time for allocation strategy and time for score-based selection on LlaMa-3-8B-Instruct. Each cell is the latency measured in seconds. Furthermore, our budget allocation can be calculated before inference, requiring only a one-time computation. Thus, PyramidKV will cause minimal extra inference overhead.

| Prompt Length | Generation Length | Inference Time | Allocation Time | Selection Time |
|---|---|---|---|---|
| 512 | 512 | 18.26 | 0.0000003 | 0.0194 |
| 512 | 1024 | 34.69 | 0.000002 | 0.0133 |
| 512 | 2048 | 70.69 | 0.000003 | 0.013 |
| 512 | 4096 | 138.62 | 0.000005 | 0.013 |
| 1024 | 512 | 17.32 | 0.000002 | 0.0131 |
| 1024 | 1024 | 34.67 | 0.000002 | 0.01288 |
| 1024 | 2048 | 70.21 | 0.000005 | 0.01296 |
| 1024 | 4096 | 138.61 | 0.000003 | 0.01297 |
| 2048 | 512 | 17.48 | 0.000004 | 0.0128 |
| 2048 | 1024 | 34.78 | 0.000006 | 0.0129 |
| 2048 | 2048 | 69.50 | 0.000003 | 0.01297 |
| 2048 | 4096 | 138.59 | 0.000003 | 0.013 |
| 4096 | 512 | 17.58 | 0.000002 | 0.013 |
| 4096 | 1024 | 34.93 | 0.000004 | 0.0129 |
| 4096 | 2048 | 69.65 | 0.000002 | 0.013 |
| 4096 | 4096 | 138.87 | 0.000002 | 0.013 |

Table 9: Extra inference overhead of PyramidKV

## L INFERENCE SPEED COMPARISON

PyramidKV does not require extra computation time for budget allocation at inference by design. We show the inference speed comparison between PyramidKV and baselines on LlaMa-3-8B-Instruct as Table 10. Each row shows the setting of using a specific "[Prompt length, Generation length]" combination. Each cell is the latency measured in seconds. PyramidKV does not sacrifice the speed. PyramidKV provides performance improvement and memory saving while runs at a comparable speed compared with baselines (i.e. SnapKV (Li et al., 2024), StreamingLLM (Xiao et al., 2023) and H2O (Zhang et al., 2024)). That's because the allocation strategy requires very limited additional complexity in the inference/generation phase compared with computation required for generation as Appendix K.

| Prompt Length | Generation Length | H2O | SnapKV | StreamingLLM | PyramidKV |
|---|---|---|---|---|---|
| 512 | 512 | 18.47 | 18.25 | 18.96 | 18.26 |
| 512 | 1024 | 35.10 | 34.76 | 36.20 | 34.69 |
| 512 | 2048 | 70.21 | 69.60 | 72.35 | 70.69 |
| 512 | 4096 | 140.80 | 139.42 | 146.37 | 138.62 |
| 1024 | 512 | 17.63 | 17.34 | 18.12 | 17.32 |
| 1024 | 1024 | 35.16 | 34.61 | 36.17 | 34.67 |
| 1024 | 2048 | 71.02 | 69.17 | 72.37 | 70.21 |
| 1024 | 4096 | 140.51 | 138.83 | 146.09 | 138.61 |
| 2048 | 512 | 17.64 | 19.54 | 18.22 | 17.48 |
| 2048 | 1024 | 35.09 | 34.76 | 36.29 | 34.78 |
| 2048 | 2048 | 70.84 | 69.56 | 72.46 | 69.50 |
| 2048 | 4096 | 140.16 | 139.55 | 145.22 | 138.59 |
| 4096 | 512 | 17.75 | 17.67 | 18.40 | 17.58 |
| 4096 | 1024 | 35.20 | 35.08 | 36.46 | 34.93 |
| 4096 | 2048 | 70.02 | 69.26 | 72.58 | 69.65 |
| 4096 | 4096 | 139.87 | 138.57 | 144.98 | 138.87 |

Table 10: Performance comparison across different configurations and methods.

## M  PYRAMIDKV EXCELS IN ALL KV CACHE SIZE LIMITATION

The evaluation results from LongBench(Bai et al., 2023) are shown in Table 11, Table 12, andTable 13. We report the results using LlaMa-3-8B-Instruct, LlaMa-3-70B-Instruct and Mistral-7B-Instruct(Jiang et al., 2023) for different KV cache sizes.

Overall, PyramidKV consistently surpasses other method across a range of KV cache sizes and different backbone models, with its performance advantages becoming particularly pronounced in memory-constrained environments. Upon examining specific tasks, PyramidKV demonstrates a notably superior performance on the TREC task, a few-shot question answering challenge. This suggests that the model effectively aggregates information from the few-shot examples, highlighting the potential for further investigation into in-context learning tasks.

| Method | Single-Document QA | | | Multi-Document QA | | | Summarization | | | Few-shot Learning | | | Synthetic | | Code | | Avg. |
|---|---|---|---|---|---|---|---|---|---|---|---|---|---|---|---|---|---|
| | NrtvQA | Qasper | MF-en | HotpotQA | 2WikiMQA | Musique | GovReport | QMSum | MultiNews | TREC | TriviaQA | SAMSum | PCount | PRe | Lcc | RB-P | |
| | 18409 | 3619 | 4559 | 9151 | 4887 | 11214 | 8734 | 10614 | 2113 | 5177 | 8209 | 6258 | 11141 | 9289 | 1235 | 4206 | |
| LlaMa-3-8B-Instruct, KV Size = Full | | | | | | | | | | | | | | | | | |
| FKV | 25.70 | 29.75 | 41.12 | 45.55 | 35.87 | 22.35 | 25.63 | 23.03 | 26.21 | 73.00 | 90.56 | 41.88 | 04.67 | 69.25 | 58.05 | 50.77 | 41.46 |
| LlaMa-3-8B-Instruct, KV Size = 64 | | | | | | | | | | | | | | | | | |
| SKV | 19.86 | 9.09 | 27.89 | **37.34** | 28.35 | **18.17** | 15.86 | 20.80 | 16.41 | 38.50 | 85.92 | 36.32 | 5.22 | 69.00 | 51.78 | 48.38 | 33.05 |
| H2O | 20.80 | 11.34 | 27.03 | 37.25 | 30.01 | 17.94 | 18.29 | 21.49 | 19.13 | 38.00 | 84.70 | **37.76** | **5.63** | 69.33 | **53.44** | **50.15** | 33.89 |
| SLM | 17.44 | 8.68 | 22.25 | 35.37 | **31.51** | 15.97 | 15.46 | 20.06 | 14.64 | 38.00 | 72.33 | 29.10 | 5.42 | **69.50** | 46.14 | 45.09 | 30.43 |
| Ours | **21.13** | **14.18** | **30.26** | 35.12 | 23.76 | 16.17 | **18.33** | **21.65** | **19.23** | 58.00 | **88.31** | 37.07 | 5.23 | **69.50** | 52.61 | 45.74 | **34.76** |
| LlaMa-3-8B-Instruct, KV Size = 96 | | | | | | | | | | | | | | | | | |
| SKV | 20.45 | 10.34 | 31.84 | 37.85 | 28.65 | **18.52** | 17.90 | 21.26 | 19.07 | 41.50 | 86.95 | 37.82 | 5.08 | 69.12 | 54.69 | **51.31** | 34.51 |
| H2O | 21.55 | 11.21 | 28.73 | 37.66 | 30.12 | 18.47 | 19.57 | 21.57 | 20.44 | 38.50 | **87.63** | **38.47** | 5.60 | 69.00 | 54.51 | 50.16 | 34.57 |
| SLM | 18.67 | 8.43 | 24.98 | 38.35 | **30.59** | 16.37 | 17.33 | 19.84 | 18.41 | 41.00 | 73.92 | 29.38 | 5.80 | **69.50** | 47.15 | 45.61 | 31.58 |
| Ours | **21.67** | **15.10** | **33.50** | **39.73** | 26.48 | 17.47 | **19.64** | **22.28** | **20.49** | 61.50 | 87.38 | 38.18 | **6.00** | 69.25 | **55.30** | 46.78 | **36.29** |
| LlaMa-3-8B-Instruct, KV Size = 128 | | | | | | | | | | | | | | | | | |
| SKV | 21.19 | 13.55 | 32.64 | 38.75 | 29.64 | **18.73** | 18.98 | 21.62 | 20.26 | 45.00 | 88.36 | 37.64 | 5.13 | 68.85 | 55.84 | **51.82** | 35.50 |
| H2O | **22.12** | 13.20 | 31.61 | 37.79 | **32.71** | 18.45 | **20.32** | 22.02 | **21.10** | 38.50 | 87.75 | **39.14** | 5.83 | **69.50** | 55.06 | 50.97 | 35.37 |
| SLM | 18.61 | 9.65 | 25.99 | 37.95 | 29.39 | 16.34 | 18.03 | 20.11 | 20.08 | 43.50 | 74.08 | 29.86 | 5.90 | **69.50** | 47.47 | 45.60 | 32.00 |
| Ours | 21.40 | **16.92** | **33.79** | **39.73** | 28.72 | 18.59 | 19.86 | **22.48** | 20.95 | 66.50 | **89.35** | 38.39 | 5.92 | 69.00 | **56.49** | 47.95 | **37.25** |
| LlaMa-3-8B-Instruct, KV Size = 2048 | | | | | | | | | | | | | | | | | |
| SKV | **25.86** | 29.55 | **41.10** | **44.99** | **35.80** | 21.81 | 25.98 | **23.40** | 26.46 | **73.50** | 90.56 | 41.66 | 5.17 | **69.25** | 56.65 | 49.94 | 41.35 |
| SLM | 21.71 | 25.78 | 38.13 | 40.12 | 32.01 | 16.86 | 23.14 | 22.64 | **26.48** | 70.00 | 83.22 | 31.75 | **5.74** | 68.50 | 53.50 | 45.58 | 37.82 |
| H2O | 25.56 | 26.85 | 39.54 | 44.30 | 32.92 | 21.09 | 24.68 | 23.01 | 26.16 | 53.00 | 90.56 | 41.84 | 4.91 | **69.25** | 56.40 | 49.68 | 39.35 |
| Ours | 25.40 | **29.71** | 40.25 | 44.76 | 35.32 | **21.98** | **26.83** | 23.30 | 26.19 | 73.00 | 90.56 | **42.14** | 5.22 | **69.25** | **58.76** | **51.18** | **41.49** |

Table 11: Performance comparison of PyramidKV (Ours) with SnapKV (SKV), H2O, StreamingLLM (SLM) and FullKV (FKV) on LongBench for LlaMa-3-8B-Instruct. PyramidKV generally outperforms other KV Cache compression methods across various KV Cache sizes and LLMs. The performance strengths of PyramidKV are more evident in small KV Cache sizes. Bold text represents the best performance.

| Method | Single-Document QA | | | Multi-Document QA | | | Summarization | | | Few-shot Learning | | | Synthetic | | Code | | Avg. |
|---|---|---|---|---|---|---|---|---|---|---|---|---|---|---|---|---|---|
| | NrtvQA | Qasper | MF-en | HotpotQA | 2WikiMQA | Musique | GovReport | QMSum | MultiNews | TREC | TriviaQA | SAMSum | PCount | PRe | Lcc | RB-P | |
| | 18409 | 3619 | 4559 | 9151 | 4887 | 11214 | 8734 | 10614 | 2113 | 5177 | 8209 | 6258 | 11141 | 9289 | 1235 | 4206 | |
| Mistral-7B-Instruct, KV Size = Full | | | | | | | | | | | | | | | | | |
| FKV | 26.90 | 33.07 | 49.20 | 43.02 | 27.33 | 18.78 | 32.91 | 24.21 | 26.99 | 71.00 | 86.23 | 42.65 | 2.75 | 86.98 | 56.96 | 54.52 | 42.71 |
| Mistral-7B-Instruct, KV Size = 64 | | | | | | | | | | | | | | | | | |
| SKV | 16.94 | 17.17 | 39.51 | **36.87** | 22.26 | 15.18 | 14.75 | 20.35 | 21.45 | 37.50 | **84.16** | 37.28 | 4.50 | 61.13 | 42.40 | 38.44 | 30.72 |
| SLM | 15.01 | 13.84 | 28.74 | 30.97 | **24.50** | 13.42 | 13.25 | 19.46 | 19.17 | 35.50 | 76.91 | 29.61 | 4.67 | 27.33 | 38.71 | 35.29 | 25.60 |
| H2O | 18.19 | 19.04 | 37.40 | 30.18 | 22.22 | 13.77 | 16.60 | **21.52** | **21.98** | 37.00 | 81.02 | **38.62** | **5.00** | **66.03** | 43.54 | **40.46** | 30.88 |
| Ours | **20.91** | **20.21** | **39.94** | 33.57 | 22.87 | **15.70** | **17.31** | 21.23 | 21.41 | **54.00** | 81.98 | 36.96 | 3.58 | 60.83 | **44.52** | 37.99 | **32.19** |
| Mistral-7B-Instruct, KV Size = 96 | | | | | | | | | | | | | | | | | |
| SKV | 19.92 | 18.80 | **43.29** | **39.66** | 23.08 | **15.94** | 16.65 | 21.26 | 21.47 | 43.50 | 83.48 | 39.74 | 4.00 | 60.10 | 45.53 | 41.12 | 32.47 |
| SLM | 15.15 | 15.48 | 31.44 | 30.03 | **23.93** | 12.73 | 16.76 | 19.15 | 19.19 | 41.50 | 75.31 | 28.71 | **5.00** | 28.48 | 38.92 | 36.05 | 26.37 |
| H2O | 19.44 | 20.81 | 38.78 | 32.39 | 21.51 | 14.43 | 17.68 | **22.40** | **21.99** | 38.00 | 82.51 | **39.94** | **6.06** | **77.48** | 45.18 | **42.43** | 32.67 |
| Ours | **20.35** | **21.87** | 41.15 | 34.94 | 21.85 | 15.81 | **18.21** | 21.66 | 21.43 | **65.00** | **83.60** | 39.60 | 4.50 | 67.80 | **45.83** | 39.38 | **34.08** |
| Mistral-7B-Instruct, KV Size = 128 | | | | | | | | | | | | | | | | | |
| SKV | 19.16 | 21.46 | 43.52 | **38.60** | **23.35** | **16.09** | 17.66 | 21.84 | 21.47 | 47.50 | **84.15** | 40.24 | 5.00 | 69.31 | **46.98** | **42.97** | 34.96 |
| SLM | 16.57 | 14.68 | 32.40 | 30.19 | 22.64 | 12.34 | 18.08 | 18.96 | 19.19 | 43.50 | 74.22 | 29.02 | 4.50 | 29.48 | 39.23 | 36.16 | 27.57 |
| H2O | 21.20 | 21.90 | 41.55 | 33.56 | 21.28 | 12.93 | **18.59** | **22.61** | **21.99** | 39.00 | 82.37 | **40.44** | **6.00** | **83.19** | 46.41 | 42.66 | 34.73 |
| Ours | **21.75** | **22.03** | **44.32** | 34.06 | 22.79 | 15.77 | 18.58 | 21.89 | 21.43 | **66.00** | 83.46 | 39.75 | 4.50 | 66.90 | 46.96 | 41.28 | **35.72** |
| Mistral-7B-Instruct, KV Size = 2048 | | | | | | | | | | | | | | | | | |
| SKV | **25.89** | **32.93** | 48.56 | **42.96** | 27.42 | 19.02 | 26.56 | **24.47** | 26.69 | 70.00 | 86.27 | 42.57 | **5.50** | **88.90** | 50.42 | 46.72 | 41.56 |
| SLM | 20.31 | 26.64 | 45.72 | 35.25 | 24.31 | 12.20 | **27.47** | 21.57 | 24.51 | 68.50 | 71.95 | 31.19 | 5.00 | 22.56 | 43.38 | 37.08 | 32.35 |
| H2O | 25.76 | 31.10 | **49.03** | 40.76 | 26.52 | 17.07 | 24.81 | 23.64 | 26.60 | 55.00 | **86.35** | 42.48 | **5.50** | 88.15 | 49.93 | 46.57 | 39.95 |
| Ours | 25.53 | 32.21 | 48.97 | 42.26 | **27.50** | **19.36** | 26.60 | 23.97 | **26.73** | **71.00** | 86.25 | **42.94** | 4.50 | 87.90 | **53.12** | **47.21** | **41.63** |

Table 12: Performance comparison of PyramidKV (Ours) with SnapKV (SKV), H2O, StreamingLLM (SLM) and FullKV (FKV) on LongBench for Mistral-7B-Instruct. PyramidKV generally outperforms other KV Cache compression methods across various KV Cache sizes and LLMs. The performance strengths of PyramidKV are more evident in small KV Cache sizes. Bold text represents the best performance.

| Method | Single-Document QA | | | Multi-Document QA | | | Summarization | | | Few-shot Learning | | | Synthetic | | Code | | Avg. |
|---|---|---|---|---|---|---|---|---|---|---|---|---|---|---|---|---|---|
| | NrtvQA | Qasper | MF-en | HotpotQA | 2WikiMQA | Musique | GovReport | QMSum | MultiNews | TREC | TriviaQA | SAMSum | PCount | PRe | Lcc | RB-P | |
| | 18409 | 3619 | 4559 | 9151 | 4887 | 11214 | 8734 | 10614 | 2113 | 5177 | 8209 | 6258 | 11141 | 9289 | 1235 | 4206 | |
| | *LLaMa-3-70B-Instruct, KV Size = Full* | | | | | | | | | | | | | | | | |
| FKV | 27.75 | 46.48 | 49.45 | 52.04 | 54.9 | 30.42 | 32.37 | 22.27 | 27.58 | 73.5 | 92.46 | 45.73 | 12.5 | 72.5 | 40.96 | 63.91 | 46.55 |
| | *LLaMa-3-70B-Instruct, KV Size = 64* | | | | | | | | | | | | | | | | |
| SKV | 23.92 | 31.09 | 36.54 | 46.66 | 50.40 | 25.30 | 18.05 | 21.11 | 19.79 | 41.50 | **91.06** | 40.26 | 12.00 | **72.50** | 43.33 | 57.62 | 39.45 |
| SLM | 22.07 | 23.53 | 27.31 | 43.21 | **51.66** | 23.85 | 16.62 | 19.74 | 15.20 | 39.50 | 76.89 | 33.06 | 12.00 | **72.50** | 40.23 | 50.20 | 35.47 |
| H2O | 25.45 | 34.64 | 33.23 | **48.25** | 50.30 | 24.88 | 20.03 | 21.50 | 21.39 | 42.00 | 90.36 | **41.58** | 12.00 | 71.50 | 43.83 | **58.16** | 39.94 |
| Ours | **25.47** | **36.71** | **42.29** | 47.08 | 46.21 | **28.30** | **20.60** | 21.62 | 21.62 | **64.50** | 89.61 | 41.28 | **12.50** | **72.50** | **45.34** | 56.50 | **42.01** |
| | *LLaMa-3-70B-Instruct, KV Size = 96* | | | | | | | | | | | | | | | | |
| SKV | **25.78** | 35.71 | 42.13 | **50.38** | **51.46** | 26.68 | 19.61 | 21.40 | 21.98 | 48.50 | **92.11** | 41.21 | 12.00 | 72.00 | 44.85 | 59.05 | 41.55 |
| SLM | 23.31 | 29.46 | 29.21 | 41.85 | 45.92 | 23.00 | 18.42 | 19.71 | 18.57 | 45.00 | 76.79 | 33.54 | 12.00 | **72.50** | 40.49 | 50.73 | 36.28 |
| H2O | 25.30 | 35.13 | 35.54 | 47.39 | 50.61 | 26.20 | 20.87 | **21.80** | **22.93** | 41.00 | 90.47 | **43.42** | 12.00 | 72.00 | 43.84 | **59.86** | 40.52 |
| Ours | 25.47 | **37.61** | **44.00** | 47.33 | 45.36 | **27.91** | **21.05** | 21.60 | 22.31 | **66.00** | 91.45 | 42.36 | 12.00 | **72.50** | **45.12** | 56.88 | **42.43** |
| | *LLaMa-3-70B-Instruct, KV Size = 128* | | | | | | | | | | | | | | | | |
| SKV | **26.22** | 37.49 | **45.70** | 50.86 | 52.82 | 28.50 | 20.38 | 21.72 | 22.56 | 53.00 | 91.61 | 41.43 | 12.00 | 71.50 | 45.06 | **60.50** | 42.58 |
| SLM | 24.25 | 29.12 | 29.24 | 40.20 | 46.28 | 21.80 | 19.55 | 19.42 | 20.61 | 48.00 | 76.60 | 33.21 | 12.00 | **72.50** | 40.65 | 51.03 | 36.53 |
| H2O | 25.61 | 35.02 | 37.74 | 47.77 | 51.16 | 26.87 | 20.57 | 20.78 | 23.33 | 42.00 | 91.65 | 43.85 | 12.00 | **72.50** | 43.50 | 59.67 | 40.88 |
| Ours | 26.06 | **40.35** | 45.67 | 50.20 | 52.78 | **29.36** | **22.31** | **22.02** | **23.69** | **71.00** | **92.27** | **44.33** | 12.00 | **72.50** | **45.90** | 59.55 | **44.37** |
| | *LLaMa-3-70B-Instruct, KV Size = 2048* | | | | | | | | | | | | | | | | |
| SKV | 26.73 | 45.18 | 47.91 | **52.00** | **55.24** | 30.48 | 28.76 | 22.35 | 27.31 | 72.50 | 92.38 | 45.58 | 12.00 | **72.50** | 41.52 | **69.27** | 46.36 |
| SLM | 26.69 | 41.01 | 35.97 | 46.55 | 52.98 | 25.71 | 27.81 | 20.81 | 27.16 | 69.00 | 91.55 | 44.02 | 12.00 | 72.00 | 41.44 | 68.73 | 43.96 |
| H2O | **27.67** | **46.51** | **49.54** | 51.49 | 53.85 | 29.97 | 28.57 | **22.79** | **27.53** | 59.00 | **92.63** | **45.94** | 12.00 | **72.50** | 41.39 | 63.90 | 45.33 |
| Ours | 27.22 | 46.19 | 48.72 | 51.62 | 54.56 | **31.11** | **29.76** | 22.50 | 27.27 | **73.50** | 91.88 | 45.47 | 12.00 | **72.50** | 41.36 | 69.12 | **46.55** |

Table 13: Performance comparison of PyramidKV (Ours) with SnapKV (SKV), H2O, StreamingLLM (SLM) and FullKV (FKV) on LongBench for LLaMa-3-70B-Instruct. PyramidKV generally outperforms other KV Cache compression methods across various KV Cache sizes and LLMs. The performance strengths of PyramidKV are more evident in small KV Cache sizes. Bold text represents the best performance.

With a small budget, our proposed method enables more effective allocation, better preserving useful attention information. Second, with a large budget, such allocation becomes less critical, as it is sufficient to cover the necessary information. To further illustrate this phenomenon, we have included an ablation study titled "Attention Recall Rate Experiment" as Figure 8. The results show that with a small budget, PyramidKV improves the attention recall rate (the percentage of attention computed using the keys retrieved by the method and the query, relative to the attention computed using all keys and the query.). However, with a larger budget (i.e., 2k KV Cache Size), the improvement decreases. For 64, 128, 256, 512, 1024 and 2048 KV Cache sizes, PyramidKV's average attention recall rate improvements are 1.87%, 0.64%, 0.61%, 0.56%, 0.47% and 0.36%.

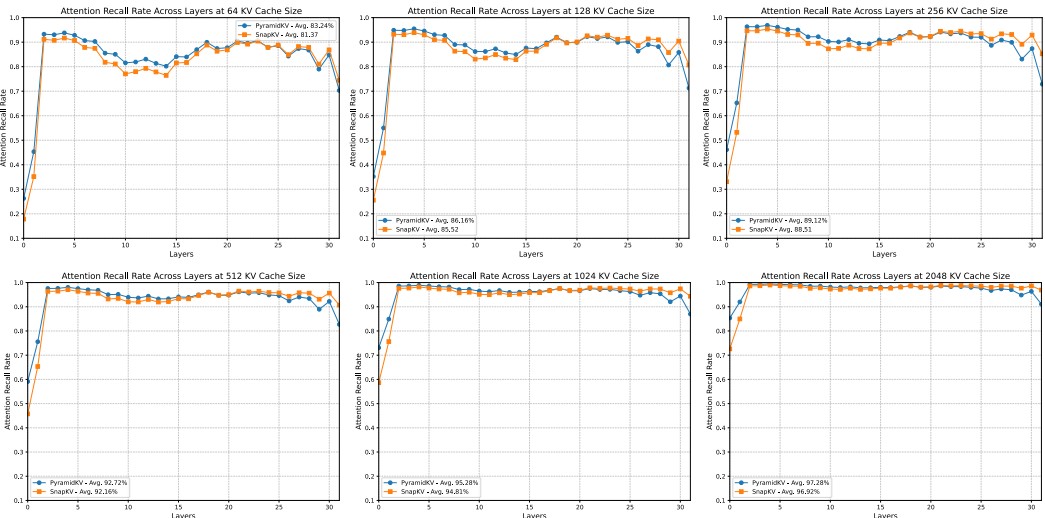

Figure 8: Attention recall rate (the percentage of attention computed using the keys retrieved by the method and the query, relative to the attention computed using all keys and the query.) comparison of PyramidKV and SnapKV.

## N    LONGBENCH RESULTS FOR 128 CONTEXT LENGTH

We conducted additional experiments using Llama-3-8B-Instruct-Gradient-1048k with a sequence length of 128k as Table 14. The results, summarized in the table below, showcase the model's performance with extended context lengths. These findings provide further validation of the scalability and robustness of our approach.

| Method | Single-Document QA | | | Multi-Document QA | | | Summarization | | | Few-shot Learning | | | Synthetic | | Code | | Avg. |
|---|---|---|---|---|---|---|---|---|---|---|---|---|---|---|---|---|---|
| | NrtvQA | Qasper | MF-en | HotpotQA | 2WikiMQA | Musique | GovReport | QMSum | MultiNews | TREC | TriviaQA | SAMSum | PCount | PRe | Lcc | RB-P | |
| SnapKV | 6.10 | 8.14 | 23.12 | 8.87 | 10.54 | 5.59 | 20.27 | 17.95 | 18.07 | 50.50 | 82.78 | 34.67 | 3.50 | 49.25 | 45.39 | 41.68 | 26.65 |
| H2O | 3.47 | 7.49 | 14.17 | 7.30 | 8.74 | 4.55 | 24.13 | 17.83 | 21.91 | 61.50 | 81.45 | 23.60 | 3.55 | 41.80 | 43.25 | 38.51 | 25.20 |
| StreamingLLM | 3.47 | 7.49 | 14.17 | 7.30 | 8.74 | 4.55 | 19.21 | 17.83 | 21.91 | 61.50 | 78.21 | 23.60 | 3.55 | 41.80 | 43.25 | 38.51 | 24.69 |
| PyramidKV | 5.41 | 8.42 | 22.61 | 9.71 | 10.73 | 5.82 | 20.37 | 18.24 | 18.32 | 54.00 | 85.33 | 34.60 | 3.50 | 52.75 | 47.23 | 42.58 | 27.48 |

Table 14: Comparison of PyramidKV with baselines at 128k context length.

## O    PYRAMIDKV PRESERVES THE LONG-CONTEXT UNDERSTANDING ABILITY

We perform Fact Retrieval Across Context Lengths ("Needle In A HayStack") (Liu et al., 2023a; Fu et al., 2024) to test the in-context retrieval ability of LLMs after leveraging different KV cache

methods. We conducted the Needle-in-a-Haystack experiment using various LLMs (i.e., Mistral-7B-Instruct-32k, LLaMA-3-8B-Instruct-8k, and LLaMA-3-70B-Instruct-8k), various KV cache sizes (i.e., 64, 96, and 128) and various methods (i.e., FullKV, PyramidKV, H2O and StreamingLLM). PyramidKV achieves Acc. performance closest to FullKV, while other methods show significant decreases. It is worth noting that PyramidKV with 128 KV cache size achieves the same 100.0 Acc. performance compared with FullKV with 8k context size for LLaMA-3-70B-Instruct.

Figure 9, Figure 10, Figure 11 show the results of **Mistral-7B-Instruct** (Jiang et al., 2023) with different cache size (64, 96 and 128, respectively).

Figure 12, Figure 13, Figure 14 show the results of **LLaMa-3-8B-Instruct** with different cache size (64, 96 and 128, respectively).

Figure 15, Figure 16, Figure 17 show the results of **LLaMa-3-70B-Instruct** with different cache size (64, 96 and 128, respectively).

| Model | Length | KV Cache | Full KV Acc. | PyramidKV Acc. | SnapKV Acc. | H2O Acc. |
|---|---|---|---|---|---|---|
| Mistral-7B | 32k | 64 | 100.00 | 80.50 | 43.90 | 48.40 |
| Mistral-7B | 32k | 96 | 100.00 | 90.50 | 72.20 | 59.10 |
| Mistral-7B | 32k | 128 | 100.00 | 91.60 | 80.10 | 64.90 |
| LLaMa-3-8B | 8k | 64 | 100.00 | 92.90 | 62.00 | 31.90 |
| LLaMa-3-8B | 8k | 96 | 100.00 | 95.80 | 80.70 | 44.20 |
| LLaMa-3-8B | 8k | 128 | 100.00 | 97.40 | 87.40 | 49.10 |
| LLaMa-3-70B | 8k | 64 | 100.00 | 99.60 | 76.20 | 47.30 |
| LLaMa-3-70B | 8k | 96 | 100.00 | 98.60 | 94.40 | 69.90 |
| LLaMa-3-70B | 8k | 128 | 100.00 | 100.00 | 98.60 | 82.30 |

Table 15: Recall Accuracy performance from Fact Retrieval Across Context Lengths ("Needle In A HayStack")

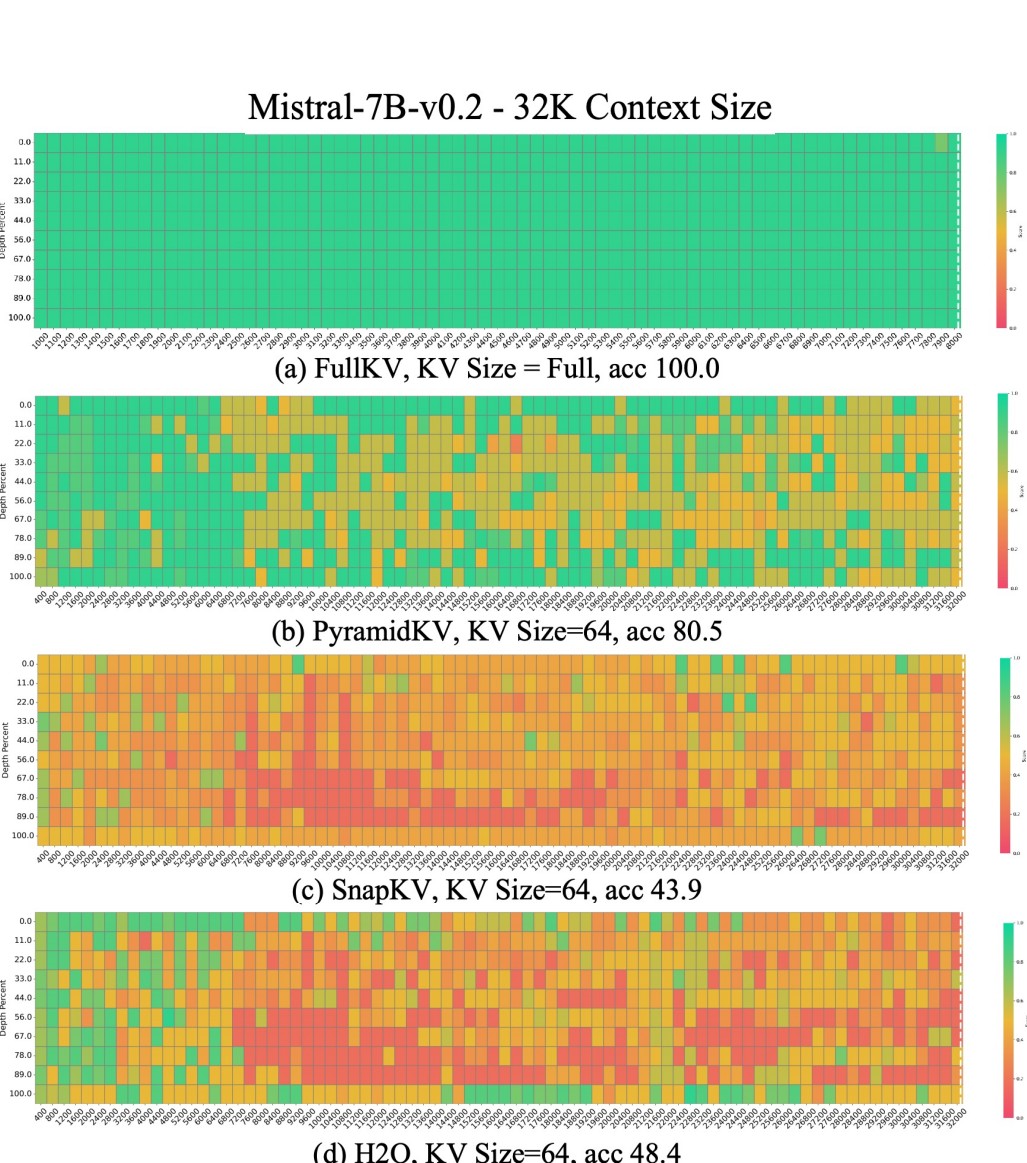

Figure 9: Results of the Fact Retrieval Across Context Lengths ("Needle In A HayStack") test in **Mistral-7B-Instruct** with **32k** context size in **64** KV cache size. The vertical axis of the table represents the depth percentage, and the horizontal axis represents the token length. PyramidKV mitigates the negative impact of KV cache compression on the long-context understanding capability of LLMs.

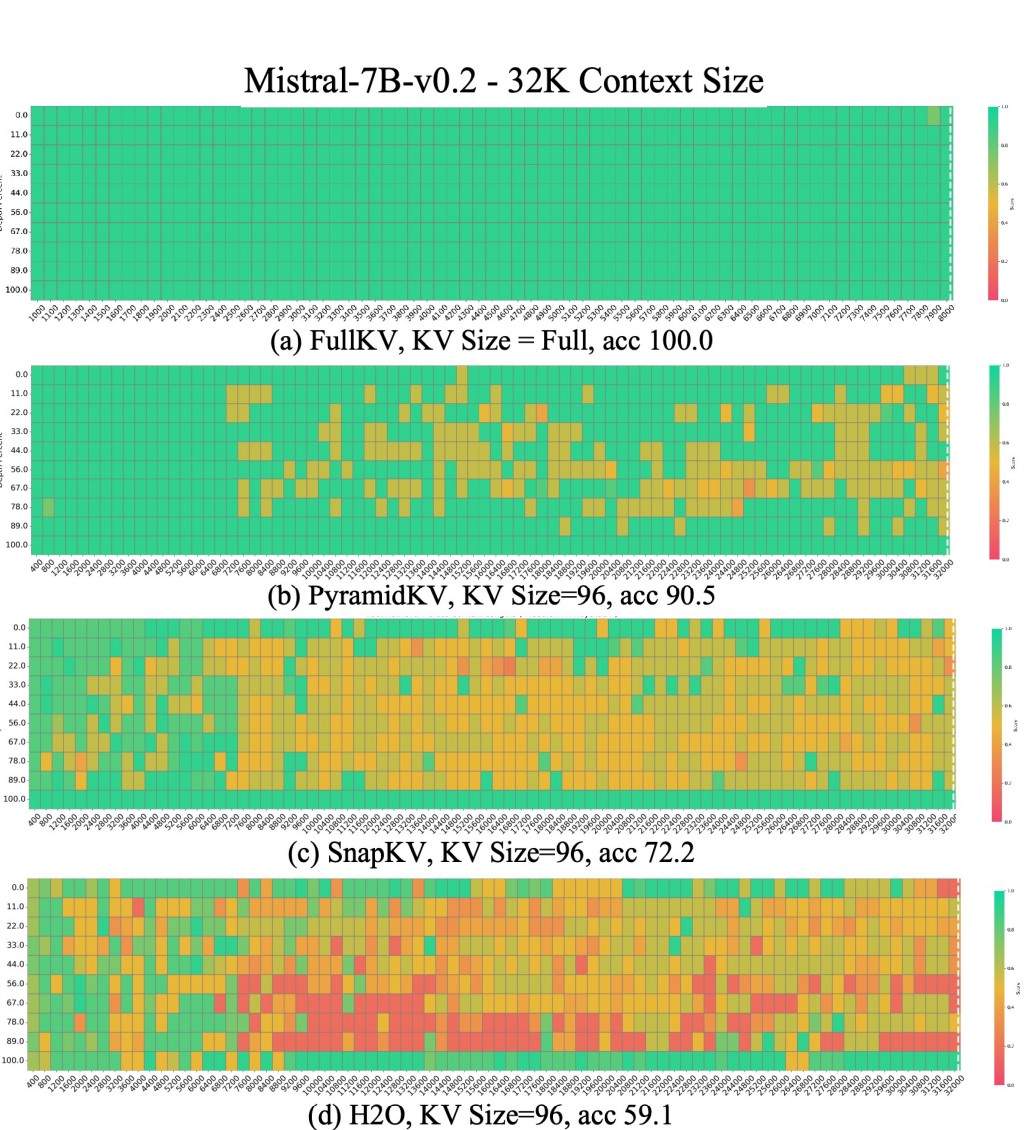

Figure 10: Results of the Fact Retrieval Across Context Lengths ("Needle In A HayStack") test in **Mistral-7B-Instruct** with **32k** context size in **96** KV cache size. The vertical axis of the table represents the depth percentage, and the horizontal axis represents the token length. PyramidKV mitigates the negative impact of KV cache compression on the long-context understanding capability of LLMs.

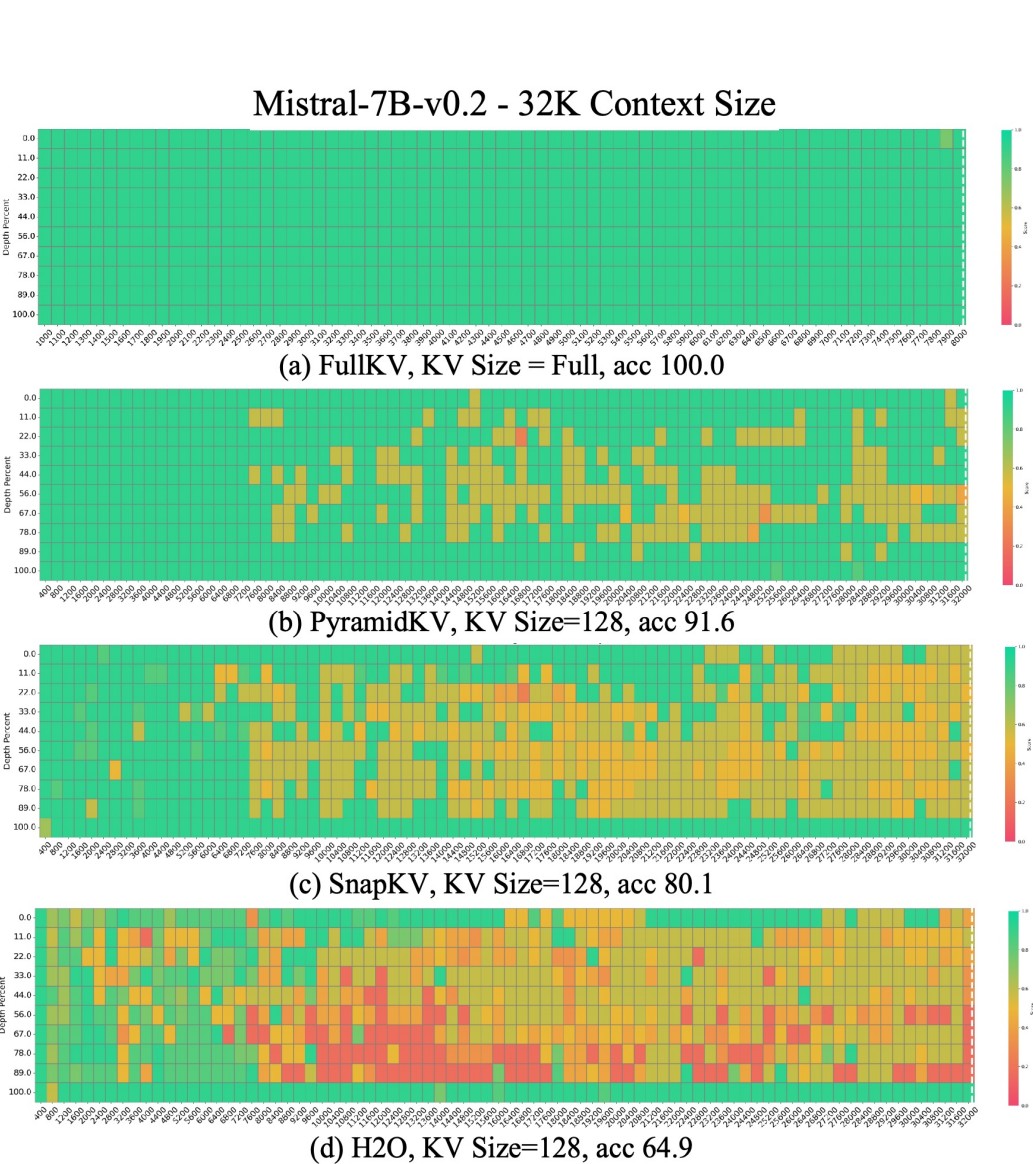

Figure 11: Results of the Fact Retrieval Across Context Lengths ("Needle In A HayStack") test in **Mistral-7B-Instruct** with **32k** context size in **128** KV cache size. The vertical axis of the table represents the depth percentage, and the horizontal axis represents the token length. PyramidKV mitigates the negative impact of KV cache compression on the long-context understanding capability of LLMs.

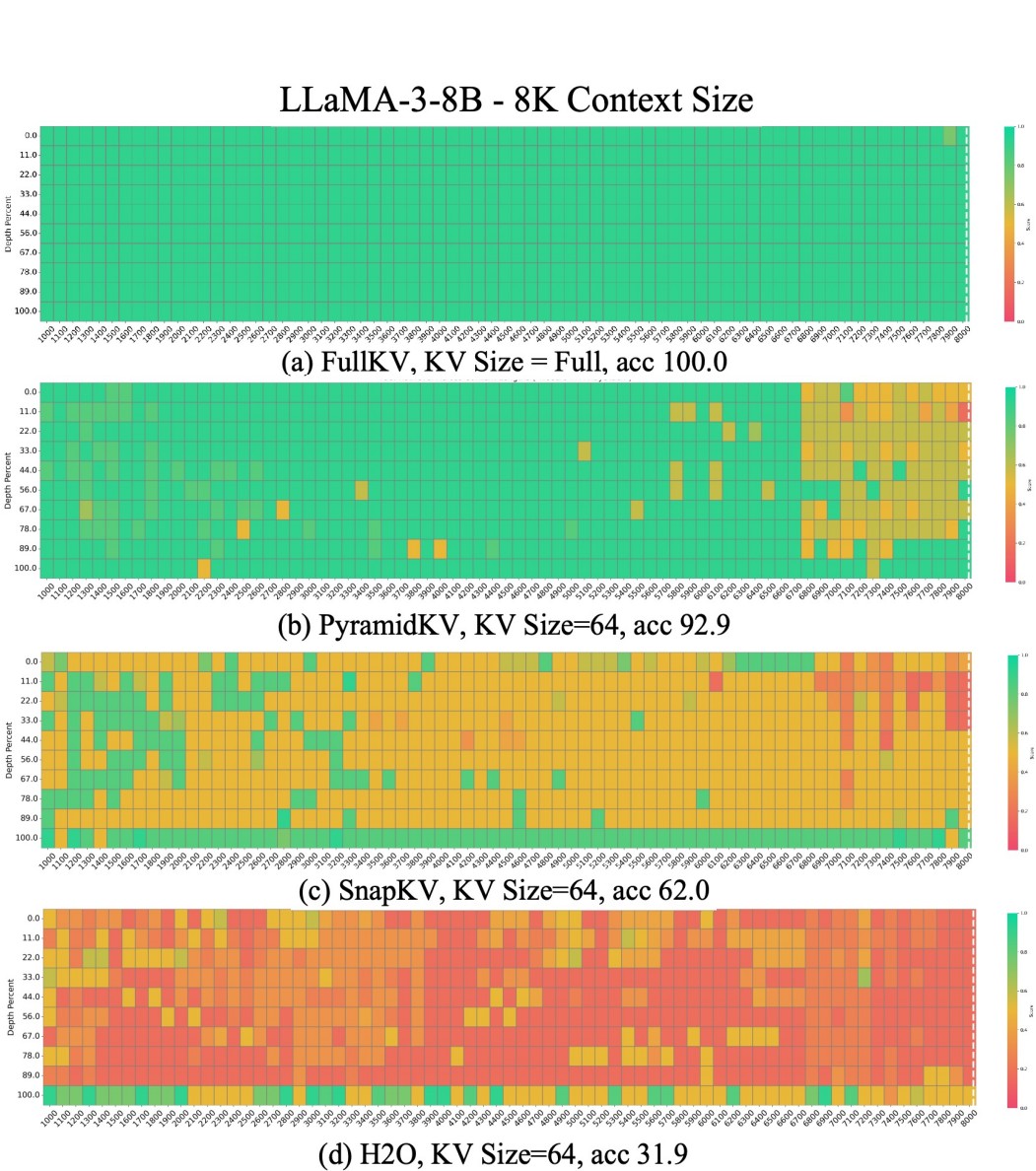

Figure 12: Results of the Fact Retrieval Across Context Lengths ("Needle In A HayStack") test in **LlaMa-3-8B-Instruct** with **8k** context size in **64** KV cache size. The vertical axis of the table represents the depth percentage, and the horizontal axis represents the token length. PyramidKV mitigates the negative impact of KV cache compression on the long-context understanding capability of LLMs.

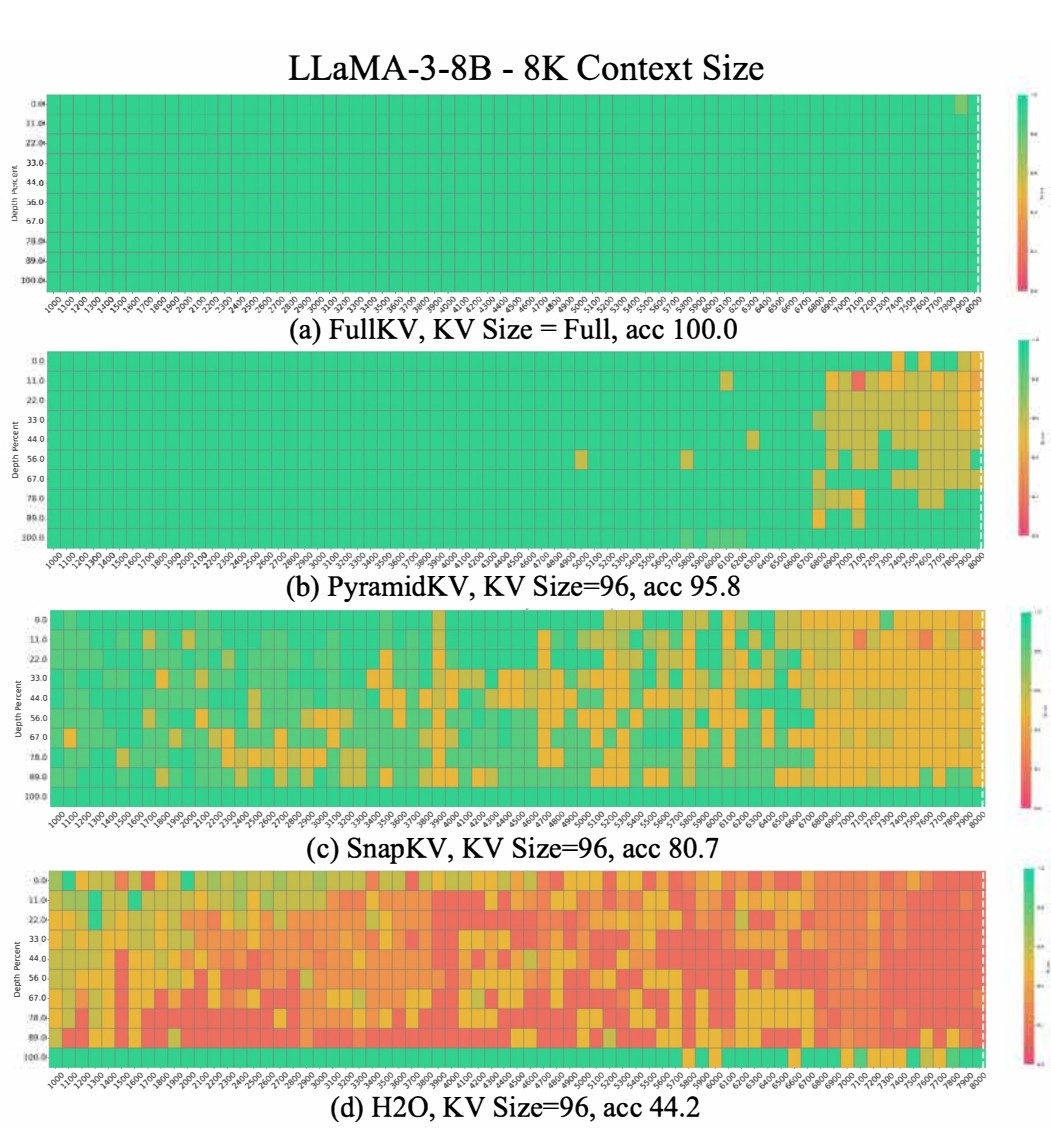

Figure 13: Results of the Fact Retrieval Across Context Lengths ("Needle In A HayStack") test in **LlaMa-3-8B-Instruct** with **8k** context size in **96** KV cache size. The vertical axis of the table represents the depth percentage, and the horizontal axis represents the token length. PyramidKV mitigates the negative impact of KV cache compression on the long-context understanding capability of LLMs.

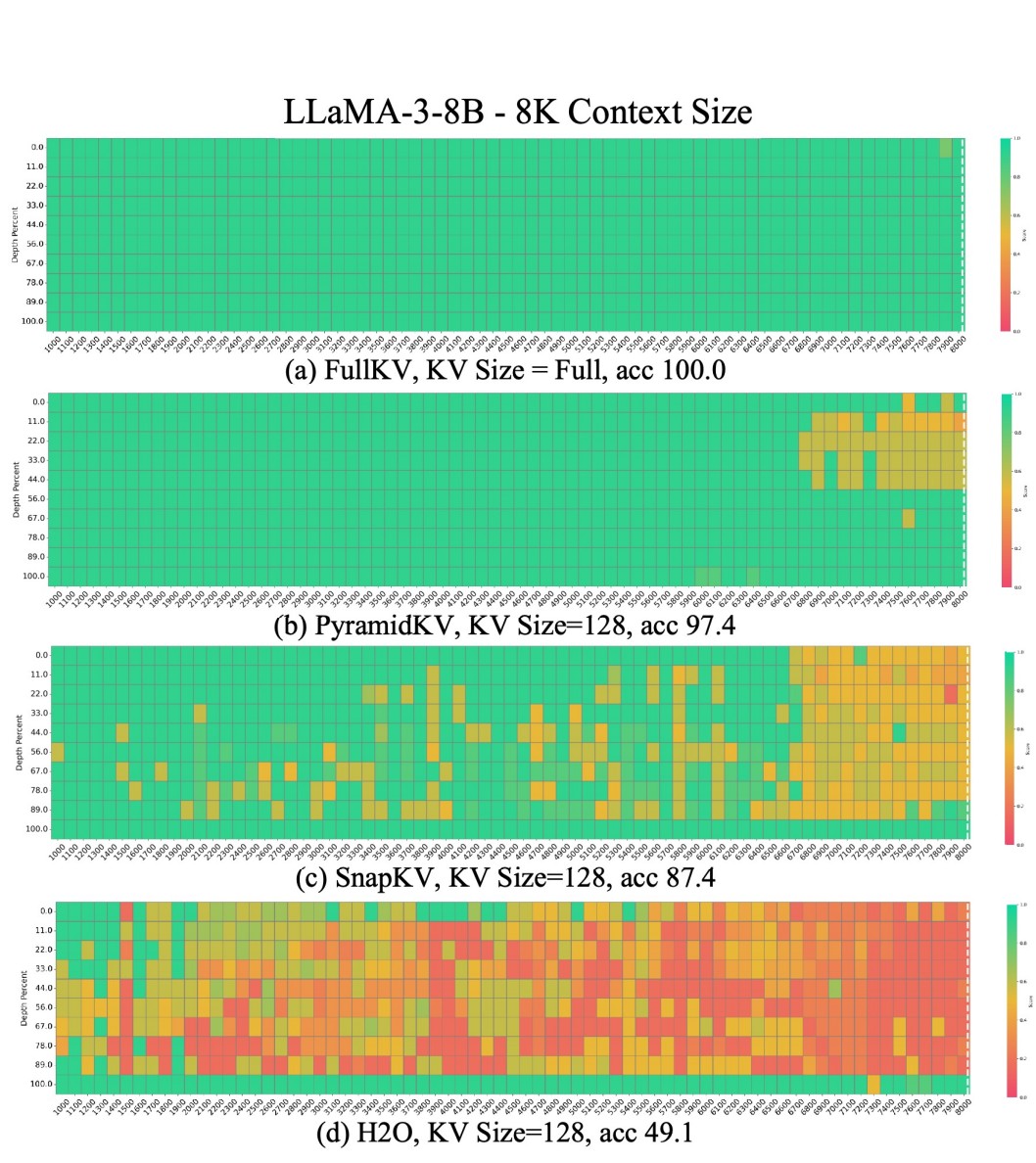

Figure 14: Results of the Fact Retrieval Across Context Lengths ("Needle In A HayStack") test in **LlaMa-3-8B-Instruct** with **8k** context size in **128** KV cache size. The vertical axis of the table represents the depth percentage, and the horizontal axis represents the token length. PyramidKV mitigates the negative impact of KV cache compression on the long-context understanding capability of LLMs.

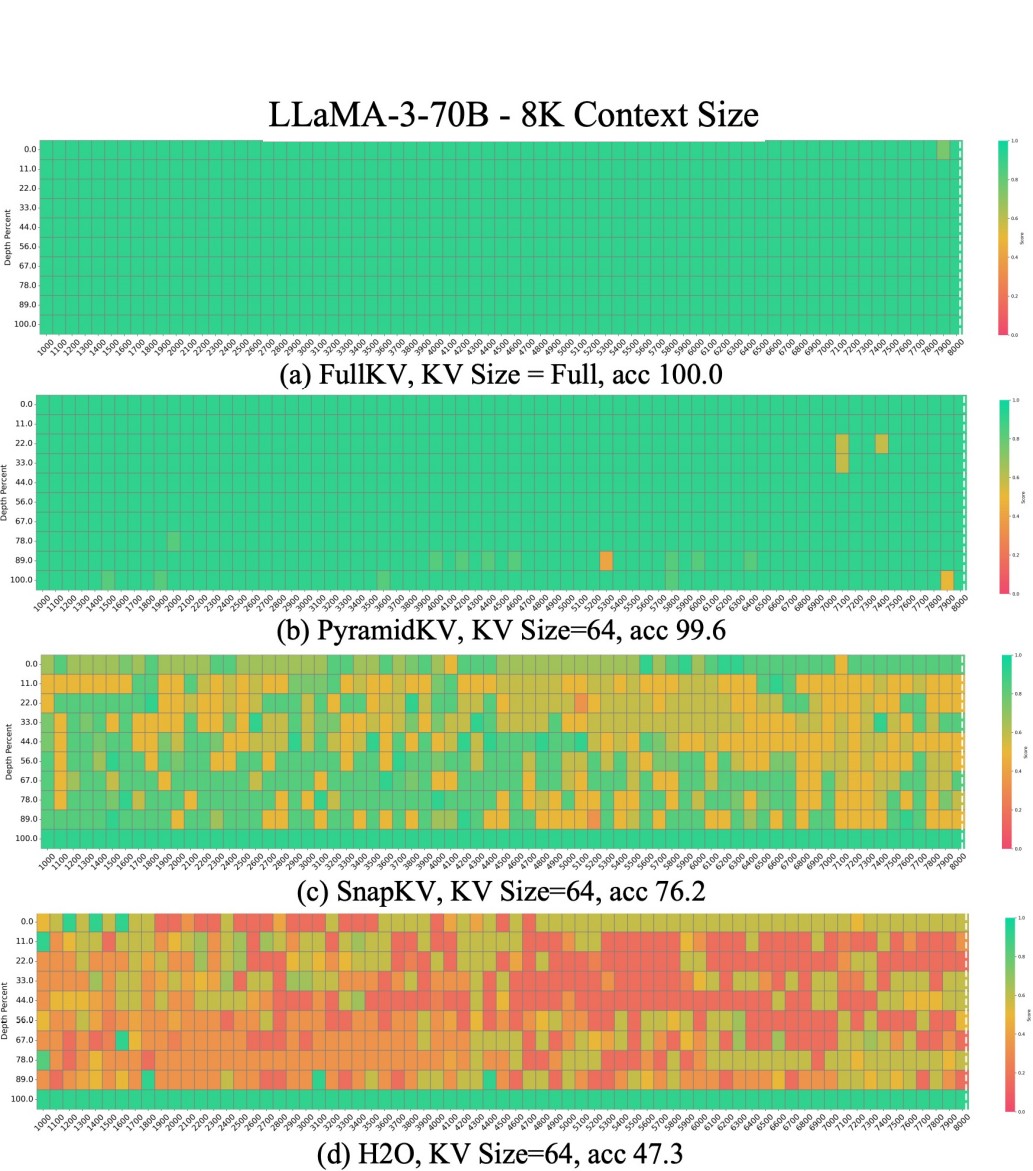

Figure 15: Results of the Fact Retrieval Across Context Lengths ("Needle In A HayStack") test in **LlaMa-3-70B** with **8k** context size in **64** KV cache size. The vertical axis of the table represents the depth percentage, and the horizontal axis represents the token length. PyramidKV mitigates the negative impact of KV cache compression on the long-context understanding capability of LLMs.

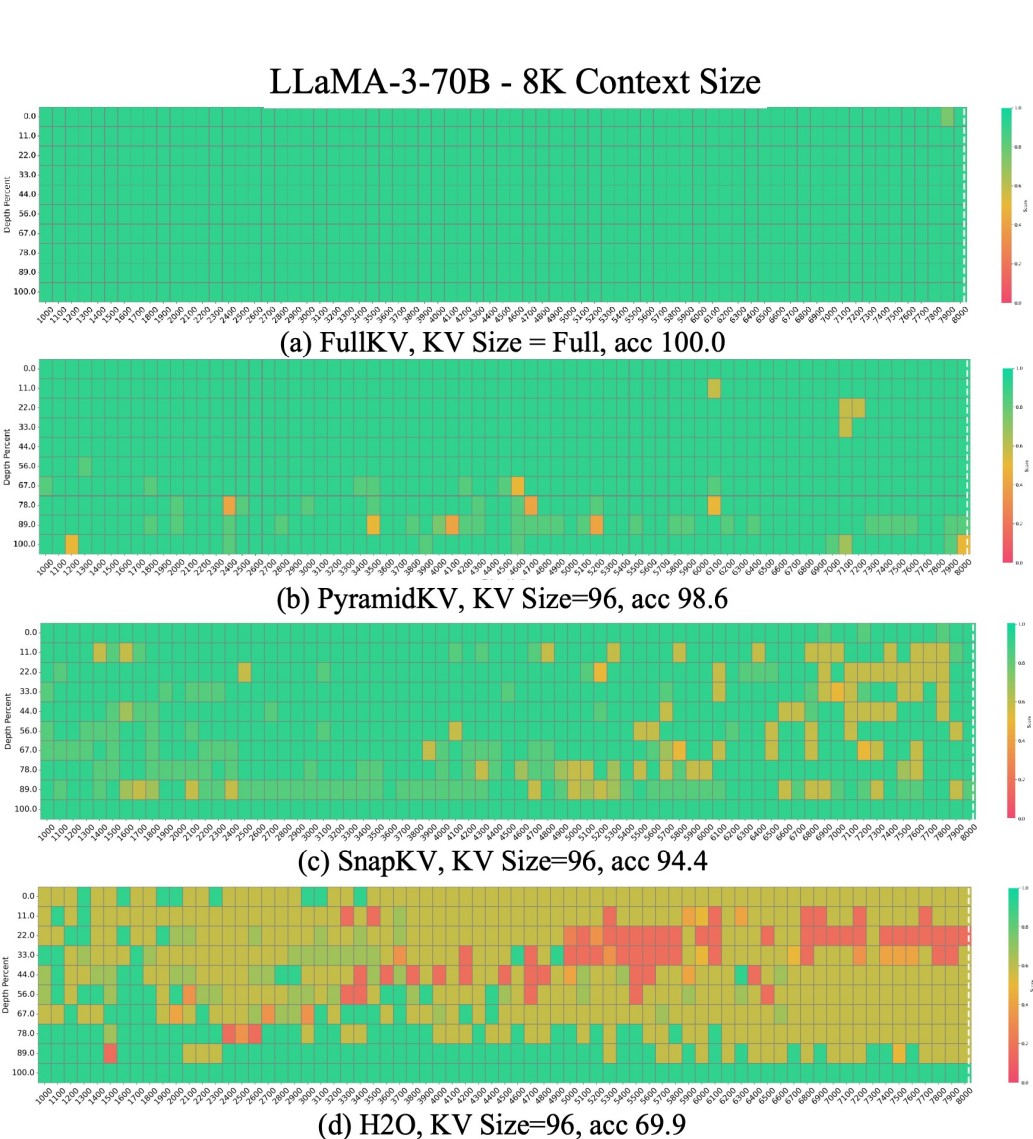

Figure 16: Results of the Fact Retrieval Across Context Lengths ("Needle In A HayStack") test in **LlaMa-3-70B** with **8k** context size in **96** KV cache size. The vertical axis of the table represents the depth percentage, and the horizontal axis represents the token length. PyramidKV mitigates the negative impact of KV cache compression on the long-context understanding capability of LLMs.

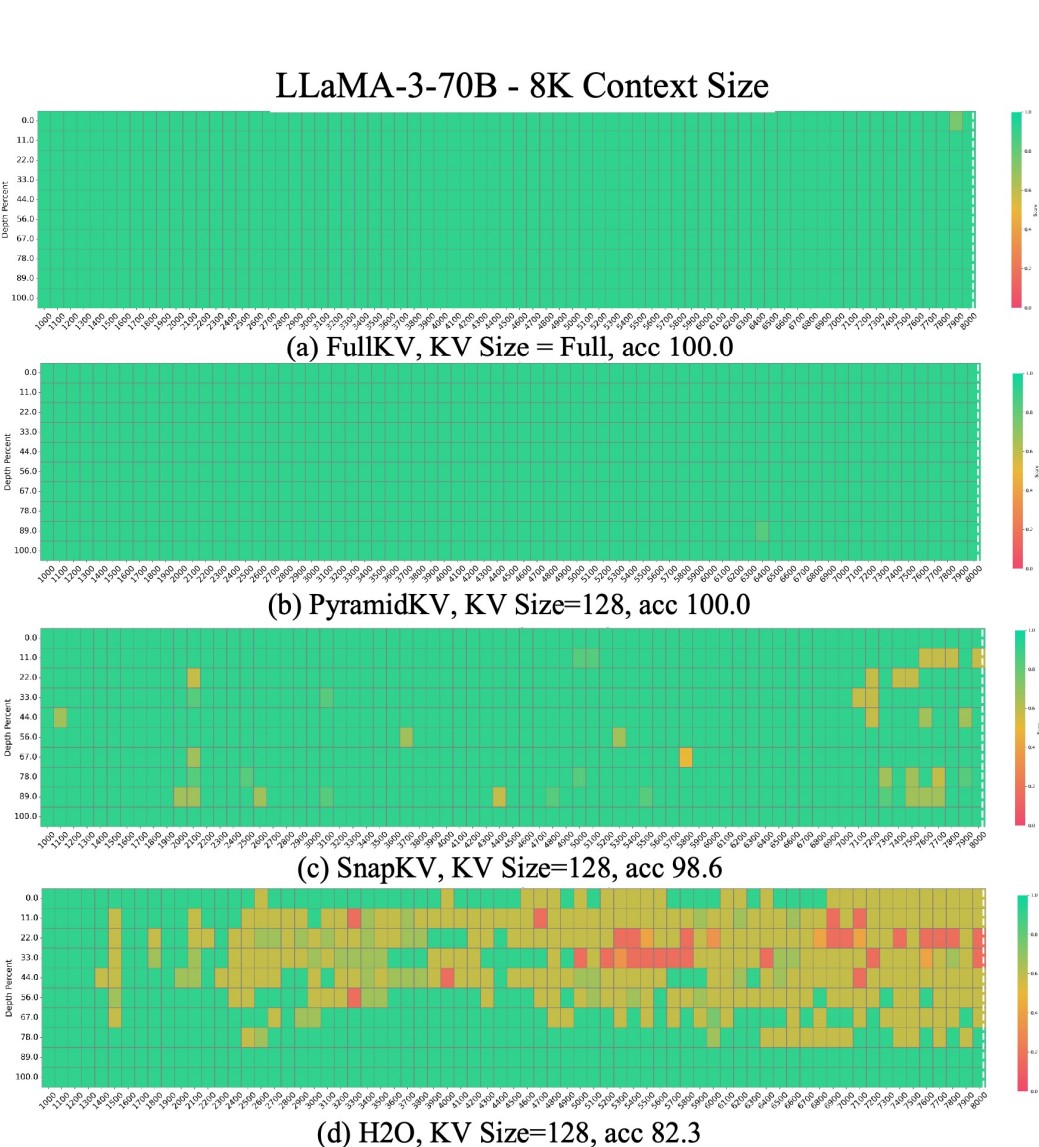

Figure 17: Results of the Fact Retrieval Across Context Lengths ("Needle In A HayStack") test in **LlaMa-3-70B** with **8k** context size in **128** KV cache size. The vertical axis of the table represents the depth percentage, and the horizontal axis represents the token length. PyramidKV mitigates the negative impact of KV cache compression on the long-context understanding capability of LLMs.

# P    ATTENTION PATTERNS ACROSS HEADS IN THE BOTTOM LAYER

Retrieval heads are predominantly located in the higher layers. Notably, no retrieval heads are observed in bottom layers. To further investigate, we conducted additional experiments on the bottom layer to analyze the attention patterns of the heads as Figure 18. Our findings indicate the absence of "massive attention" in any individual head.

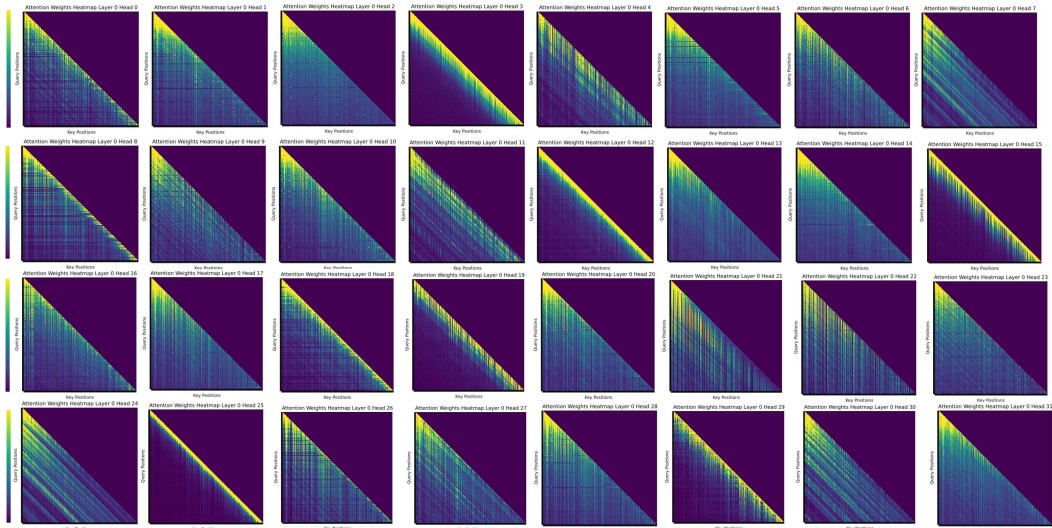

Figure 18:  Attention patterns of retrieval-augmented generation across heads in the bottom layer in LlaMa.

# Q    PYRAMIDKV IMPLEMENTATION AT VLLM

To help compare the vLLM implementation with the vanilla dense attention backend in terms of throughput, we perform the experiment. We present the throughput comparison between the PyramidKV vLLM implementation and the vanilla dense attention backend in a setting where the inputs have varying context lengths without shared prefixes.

In Figure Figure 19, we plot the throughput of the LlaMa 8b model by varying length. We observe that relative throughput under compression decreases as the new input context length approaches the limit, causing new sequences to wait longer before being added to the decoding batch.

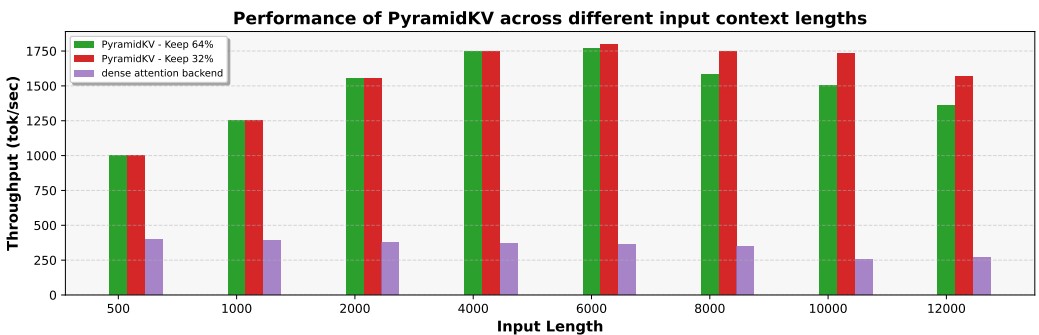

Figure 19:  Throughout performance of PyramidKV across different input context lengths using LlaMa-3-8b model.

We find that allocating/releasing/moving/accessing very small chunks of memory may cause ineffi-ciency and fragmentation in a naive implementation of PyramidKV at vLLM. As PyramidKV applies different allocation budgets for different layers. The top layers have less budget, while the bottom layers have more budget. The application of KV cache eviction with different budgets across layers at the standard paged attention frameworks (i.e., vLLM) is ineffective as it only reduces the cache size proportionally to the layer with the lowest compression rate, and all evictions beyond this rate merely increase cache fragmentation.

However, the problem could be solved by adapting paged attention to page out cache on a per-layer basis. We expand the block tables of each sequence to include block tables for each layer of the cache so that they can be retrieved for each layer's KV cache during attention without the use of fixed memory offsets.

The implementation of PyramidKV is orthogonal to multi-gpu setting at vLLM because vLLM shards attention by head in tensor parallel, so the performance bonus would not change too much with tensor parallel or pipeline parallel. Each TP rank will have it's corresponding heads of size ($num\_heads/tp\_size$). To help see the performance of PyramidKV implementation at vLLM compared with the vLLM dense attention backend at tensor parallel and pipeline parallel, we perform the experiment as Figure 20. We present the throughput comparison between PyramidKV vLLM implementation and vanilla dense attention backend at PP=1, TP=1; PP=1, TP=2; PP=1, TP=4; PP=2, TP=2 settings

Results show that the performance bonus remains similar with tensor parallel and pipeline parallel. Tensor parallelism demonstrates significant improvements when the degree is increased from 1 to 2. Notably, the result also reveals a non-linear relationship between the number of GPUs and throughput; doubling the GPU count did not yield a proportional doubling of throughput.

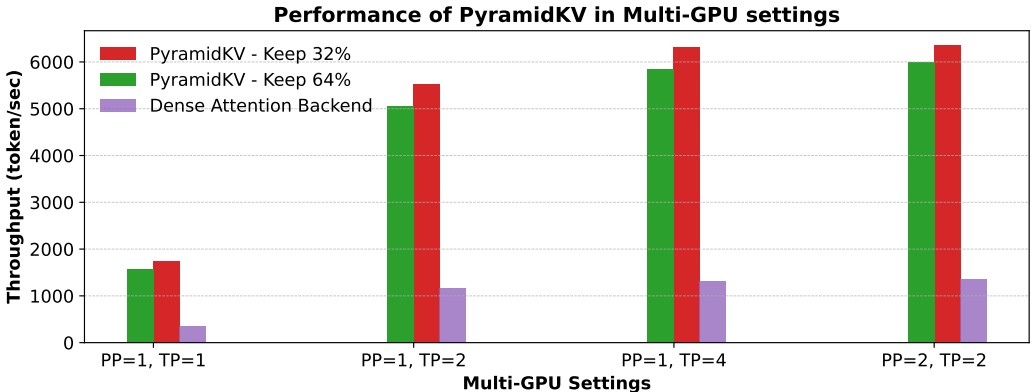

Figure 20: Throughout performance of PyramidKV across different multi-GPU settings using LlaMa-3-8b model at 8k context size.