# OpenReview forum: "PyramidKV: Dynamic KV Cache Compression based on Pyramidal Information Funneling"
_ICLR.cc/2025/Conference — Submitted to ICLR 2025_

### Official Review · Reviewer_MCVv · 2024-10-30

**Soundness:** 3
**Presentation:** 3
**Contribution:** 3
**Rating:** 8
**Confidence:** 5

**Summary:**

The paper proposes to dynamically adjusts the KV cache size across different layers through the depth of transformers, which is motivated by the fact that attention is denser in the initial layers and sparser in the later layers.

**Strengths:**

1. The observation on the attention pattern across layers is insightful.
2. Performance, especially, long context capabilities are preserved much better compared to methods in the class.
3. Experiments are quite through, examining a few challenging tasks in the long context scenarios.

**Weaknesses:**

1. It's not clear if this method can be implemented in real systems like vLLM/SGLang, as the memory management is still very arbitrary and frequent, which go against the hardware design.

**Questions:**

1. How does the speed-up change wrt tensor parallel and pipeline parallel?
2. How does the memory allocation and release be implemented? I would assume there will be significant memory fragmentation during the process.

---

> ### Author Response · Authors · 2024-11-23
>
> We thank the reviewer for the constructive suggestions, and we would like to clarify a few points.
>
> # Responses to Weakness
>
> **W1**: It's not clear if this method can be implemented in real systems like vLLM/SGLang.
>
> **A1**:
>
> The method could be implemented in both vLLM and SGLang. In the case of vLLM, we have uploaded the implementation (https://github.com/a-anonymous-user/vLLM-KV) from an anonymous account. The eviction process involves the following key steps:
>
> 1. **Determine the KV Size Budget for Layers**: Calculate the memory allocation for the key-value pairs (KVs) for each layer in the cache system. This step involves assessing the total available memory and deciding how much should be assigned to each layer.
>
> 2. **Select KVs for Eviction**: Identify specific KVs to be evicted based on their attention scores.
>
> 3. **Reorder KVs in the Physical Cache**: Rearrange the KVs in the physical cache so that all evicted KVs are grouped into contiguous blocks that do not contain any non-evicted KVs. This reordering is essential for efficient block-level memory management.
> Free Blocks: Release the blocks containing the evicted KVs, allowing the memory to be reallocated for subsequent scheduling iterations. This step updates the BlockManager but leaves the underlying physical cache structure unchanged.
>
> 4. **Update the Physical Cache**: Adjust the positioning of KVs in the physical cache according to the new ordering determined in Step 3.
>
> To illustrate this process, we provide a vLLM implementation demonstrating these steps. Specifically:
>
> 1. Step 1 is managed by https://github.com/a-anonymous-user/vLLM-KV/blob/main/vllm/kvcompress/metrics.py.
>
> 2. Steps 2–4 are handled by CompressionScheduler in https://github.com/a-anonymous-user/vLLM-KV/blob/main/vllm/kvcompress/scheduler.py.
>
> 3. Step 5 is executed by CacheEngine.execute_cache_moves in https://github.com/a-anonymous-user/vLLM-KV/blob/main/vllm/llm/llm_engine.py, which is invoked in the main engine loop.
>
> # Responses to Questions
>
> **Q1**: How does the speed-up change wrt tensor parallel and pipeline parallel?
>
> **A1**:
>
> The speed-up offered by PyramidKV is complementary to that achieved through tensor parallelism and pipeline parallelism, as these approaches are not mutually exclusive. PyramidKV can be seamlessly integrated with both tensor parallelism and pipeline parallelism. We have included these details at Appendix D.
>
>
> **Q2**: How does the memory allocation and release be implemented? I would assume there will be significant memory fragmentation during the process.
>
> **A2**:
>
> To perform KV cache eviction, we use torch.gather. Below, we outline the memory allocation and release process of torch.gather:
>
> 1. **Index Selection**: Identify the positions of the elements to extract from the input tensor.
>
> 2. **Memory Location Calculation**: Compute the specific memory locations of the elements to be extracted using the strides of the input tensor across each dimension.
>
> 3. **Output Tensor Creation**: Allocate memory to create a new output tensor and copy the selected elements to their corresponding positions in the output tensor.
>
> 4. **Memory Management**: Since torch.gather is not an in-place operation, it creates a new tensor to store the results, while the memory of the original input tensor is released.

---

> > ### Comment · Reviewer_MCVv · 2024-11-23
> >
> > Overall, I think the amount of work this paper and quality are well above the ICLR threshold.
> > Some follow-up questions below. I'll be happy to raise my score after the authors answer these concerns. You do not need to show positive result for PyramidKV to clarify my questions. Honest and insightful discussions are more appreciated.
> >
> > Follow-up questions
> > 1. About the vLLM implementation, how does it compare to the vanilla, dense attention backend, e.g.,  flash_attn and flash_infer, in terms of latency and throughput? I'm more interested in a realistic setting. For example, (a) a request queue of varying length without shared prefix, (b) a request queue of similar length, all with the same lengthy prefix, e.g., a 100k doc with different questions to it, (c) a request queue of similar length, half of them shares a lengthy prefix, e.g., a 100k doc with different questions to it. You will probably need to set up a vllm server to simulate this instead of directly using the benchmark methods.
> > 2. Following 1, do you think this whole KV eviction line of work, is compatible/a maintainable feature with state-of-the-art serving systems? More specifically, after rushing this PR, do you find frequent allocating/releasing/moving/accessing very small chunks of memory will very likely cause inefficiency and fragmentation?
> > 3. Following the tp, pp discussion, can you show the actual benchmark results? Also, as you have a vLLM draft, can you use vLLM flash_attn and flash_infer backend as baseline to show the speed-up wrt tp and pp size? I personally would think the bonus would significantly go down with tp/pp, let alone the implementation challenge.

---

> > > ### Author Response · Authors · 2024-11-25
> > >
> > > We appreciate the reviewer's recognition of the amount of work and the quality of our paper. And we would like to answer the follow-up questions to address your concerns.
> > >
> > > **Q1**: About the vLLM implementation, how does it compare to the vanilla, dense attention backend, e.g., flash_attn and flash_infer, in terms of latency and throughput? I'm more interested in a realistic setting. For example, (a) a request queue of varying length without shared prefix, (b) a request queue of similar length, all with the same lengthy prefix, e.g., a 100k doc with different questions to it, (c) a request queue of similar length, half of them shares a lengthy prefix, e.g., a 100k doc with different questions to it. You will probably need to set up a vllm server to simulate this instead of directly using the benchmark methods.
> > >
> > > **A1**:
> > >
> > > 1. To help compare the vLLM implementation with the vanilla dense attention backend in terms of throughput, we perform the experiment at Appendix Q. We present the throughput comparison between the PyramidKV vLLM implementation and the vanilla dense attention backend in a setting where the inputs have varying context lengths without shared prefixes.
> > >
> > > 2. In Figure 19, we plot the throughput of the LlaMa 8b model by varying length. We observe that relative throughput under compression decreases as the new input context length approaches the limit, causing new sequences to wait longer before being added to the decoding batch.
> > >
> > > 3. We believe that the settings (b) and (c) are very interesting and meaningful points to explore! We are now working on implementing these settings. However, given the limited time remaining in the discussion phrase, we may not be able to systematically test these settings and post a conclusive result here. We agree with your intuition that these realistic settings are important in application. This exploration likely merits a dedicated study as our revised version or future work.
> > >
> > > **Q2**: Following 1, do you think this whole KV eviction line of work, is compatible/a maintainable feature with state-of-the-art serving systems? More specifically, after rushing this PR, do you find frequent allocating/releasing/moving/accessing very small chunks of memory will very likely cause inefficiency and fragmentation?
> > >
> > > **A2**:
> > >
> > > We think that KV eviction works are compatible with SOTA serving systems. For the specific question about memory fragmentation, we have the below findings:
> > >
> > > 1. We find that allocating/releasing/moving/accessing very small chunks of memory may cause inefficiency and fragmentation in a naive implementation of PyramidKV at vLLM. As PyramidKV applies different allocation budgets for different layers. The top layers have less budget, while the bottom layers have more budget. The application of KV cache eviction with different budgets across layers at the standard paged attention frameworks (i.e., vLLM) is ineffective as it only reduces the cache size proportionally to the layer with the lowest compression rate, and all evictions beyond this rate merely increase cache fragmentation.
> > >
> > > 2. However, the problem at 1 could be solved by adapting paged attention to page out cache on a per-layer basis. We expand the block tables of each sequence to include block tables for each layer of the cache so that they can be retrieved for each layer’s KV cache during attention without the use of fixed memory offsets.
> > >
> > > To summarize, we have found that small chunks of memory will cause fragmentation, but we also find that this could be easily solved by expanding the block tables to include block tables for each layer of the cache.
> > >
> > > **Q3**: Following the tp, pp discussion, can you show the actual benchmark results? Also, as you have a vLLM draft, can you use vLLM flash_attn and flash_infer backend as baseline to show the speed-up wrt tp and pp size? I personally would think the bonus would significantly go down with tp/pp, let alone the implementation challenge.
> > >
> > > **A3**:
> > >
> > > 1. The implementation of PyramidKV is orthogonal to multi-gpu setting at vLLM because vLLM shards attention by head in tensor parallel, so the performance bonus would not change too much with tensor parallel or pipeline paralle.
> > >
> > > 2. To help see the performance of PyramidKV implementation at vLLM compared with the vLLM dense attention backend at tensor parallel and pipeline parallel, we perform the experiment at Appendix Q.
> > >
> > > 3. We present the throughput comparison between PyramidKV vLLM implementation and vanilla dense attention backend at PP=1, TP=1; PP=1, TP=2; PP=1, TP=4; and PP=2, TP=2 settings as Figure 20.
> > >
> > > 4. Results show that the performance bonus remains similar with tensor parallel and pipeline parallel.
> > > Tensor parallelism demonstrates significant improvements when the degree is increased from 1 to 2. Notably, the result also reveals a non-linear relationship between the number of GPUs and throughput; doubling the GPU count did not yield a proportional doubling of throughput.

---

> ### Author Response · Authors · 2024-11-25
>
> We would like to thank the reviewer once again for the valuable feedback, which can significantly help us improve our work. We hope these clarifications can address the concerns. We hope you find our response compelling enough to consider a re-evaluation of our score. If there are any remaining concerns, we would really appreciate the opportunity to discuss them further during the discussion period.

---

> > ### Author Response · Authors · 2024-11-28
> >
> > Dear Reviewer MCVv,
> >
> > We would appreciate it if you could let us know whether our response has adequately addressed your concerns and questions. We remain available to address any further questions you may have.
> >
> > Thank you once again for your time and effort!

---

> ### Comment · Reviewer_MCVv · 2024-11-29
> **Official Response MCVv**
>
> Thanks for the helpful discussion!
>
> Would you mind sharing the script and setting for benchmarking PyramidKV against vLLM? I'm quite surprised it can be done without big refactoring over the whole CSRC, e.g., cache_kernels.cu and attention kernels. I'll raise my score after I reproduced the result. I'll make sure to finish before Dec 2nd.

---

> ### Author Response · Authors · 2024-12-02
>
> We sincerely thank the reviewer for their insightful questions regarding the implementation of KV cache eviction in vLLM [1] and its alignment with the memory management design at vLLM. We would like to take this opportunity to clarify a few key points. While we are currently unable to update the paper, we will make sure that all the following discussions will be incorporated into the discussion section of the revised version of our paper.
>
> # Challenges and Limitations
>
> In this section, we discuss the challenges and limitations about the implementation of PyramidKV in vLLM, focusing on aligning the implementation with vLLM’s memory management design. Our primary emphasis is on redesigning the cache architecture to minimize fragmentation, which requires refactoring some components in CSRC, particularly within the cache kernels. We estimate that implementing KV cache eviction will result in a 15% slowdown due to additional overhead. Therefore, KV cache eviction works enhance total throughput by enabling larger batch sizes, but latency for each sample is about 15% slower.
>
>
> ## Fragmentation in original vLLM’s kv cache design
>
> In vLLM's original design, each block stores the keys and values generated for all attention heads across all tokens in the block. Consequently, allocating keys and values for one head requires allocating them for all heads. This design can lead to increased memory fragmentation, particularly in PyramidKV, where the number of keys and values varies across layers. Although the number of keys and values per head within a layer remains consistent, these numbers differ between layers. Under vLLM's original design, if one head requires more keys and values than the others, the system must allocate space for the same number across all heads, which would add fragmentation.
>
> ## Re-design the cache structure to avoid fragmentation
>
> To prevent fragmentation, it is necessary to redesign the KV cache structure. The fragmentation issue can be resolved by adopting a new design where each block contains KV pairs for a single head. This approach eliminates the need to allocate equal space for all heads when one head requires more KV pairs, thereby reducing fragmentation. By restructuring the cache to allocate blocks on a per-head basis, we ensure that additional KV pairs for one head do not force unnecessary space allocation for other heads, effectively mitigating fragmentation.
>
> ## Scheduling Overhead due to the re-design of cache structure
>
> **Increased size of blocks in the new-designed cache structure**
>
> Originally, each block contained one token per layer and head (32 layers and 8 KV heads for LLaMA). However, the transformation modifies this structure so that each block contains a key or value for a single head.
>
> **Increased size of blocks in the case of LlaMa**
>
> This implies that the number of blocks increases significantly. For a model with num_layer transformers with num_head heads for each layer, the number of blocks will be num_layer×num_head times larger. For instance, in the case of llama (32 layers Transformers and each key and value has 8 heads) if there were initially 128 blocks, this would expand to 128×32×8 making the number of blocks 32×8 times larger.
>
> **Increased size of blocks results in additional scheduling overhead**
>
> This increase is the primary source of additional scheduling overhead, mainly arising from operations on block tables and context-length tensors. These tensors are used to determine the number of new blocks needed and available blocks to allocate during scheduling.
>
> Previously, scheduling was performed while block tables were stored as Python lists, which were dynamically updated during block allocation. After scheduling, GPU tensors were generated from these block tables for use during the attention forward pass. To optimize this process, it is necessary to update the block manager to store block tables directly on the GPU, enabling all block allocation operations to be handled in PyTorch. As a result, the block tables can be sent directly to the model forward pass without intermediate conversions. Moving scheduling operations to the GPU significantly improves speed.
>
> Overall, the additional 15% overhead primarily stems from the increased number of blocks.
>
>
> ## IO Overhead
> The eviction process is relatively fast, and the I/O overhead may not constitute a significant portion of the total overhead, particularly because PyramidKV does not perform compression at every step. However, if the KV cache eviction were to occur at every step, the I/O overhead could be substantially higher.

---

> > ### Author Response · Authors · 2024-12-02
> >
> > ## Multi-GPU settings
> >
> > **Tensor Parallelism (TP)**: In TP, the KV caches for different attention heads are distributed across GPUs. The KV cache footprint per GPU depends on the number of KVs allocated per head. Each TP rank manages a subset of heads, typically of size num_heads/tp_size. However, TP restricts the ability to independently adjust the compression rate for individual heads, limiting flexibility in optimizing cache usage.
> >
> > **Pipeline Parallelism (PP)**: In PP, each GPU is responsible for different layers of the model. The KV cache footprint per GPU varies based on the number of KVs required per layer.
> >
> > **The Challenge to Extend KV Cache Eviction Budget Allocation Methods to TP and PP**
> >
> > TP and PP introduce unique constraints on compression flexibility. TP limits the ability to vary compression rates across heads, while PP restricts such variations across layers. In PyramidKV, the number of KVs can be adjusted dynamically across layers, and consequently, the number of KVs per head also varies between layers. These characteristics are inherently unsupported by the vLLM's original designs of TP and PP, which constrain KV cache eviction budget allocation methods. As such, KV cache eviction budget allocation methods cannot be directly applied to multi-GPU setups using TP or PP without modification.
> >
> > **Multi-GPU Solution**
> >
> > The simplest approach is to enforce uniform budget allocation per layer or head (e.g., as in SnapKV) and adopt this method for TP/PP. However, this strategy sacrifices the performance benefits of allocation methods like PyramidKV, which optimizes cache usage with effective budget allocation. This motivates a rethinking of TP and PP implementation to better accommodate flexible budget allocation. We believe the reliable solution is to is to manage KV cache eviction on a per-GPU basis, maintaining a constant compression rate across GPUs while allowing variability across the layers or heads allocated to each GPU. This approach enables each GPU to independently compress the KVs in its allocated cache while aligning with the constraints of distributed parallelism.
> >
> >
> > ## Overall Overhead
> >
> > The majority of the overhead stems from scheduling. The redesigned approach introduces a 15% slowdown overall. Therefore, KV cache eviction improves total throughput by enabling larger batches but latency for each sample is about 15% slower. IO performance and fragmentation remain consistent across both single-GPU and multi-GPU configurations.
> >
> > ## Response to “I'm surprised it can be done without big refactoring over CSRC
> > We apologize for not clearly addressing this in our previous responses. Implementing the new cache structure may require some refactoring at CSRC, particularly within the cache kernel components.
> >
> > ## Related Work
> >
> > Our discussion in this section draws significant inspiration from prior works [4][5][6][7][8][9].
> >
> > [2] addresses the I/O bottleneck by minimizing KV cache access through the identification of important tokens via critical feature channels.
> >
> > [3] introduces the Triton library, which optimizes sparse attention kernels customizable at both per-head and per-context-range levels. This approach overcomes the practical speed limitations of traditional sparse attention through hardware-accelerated optimization.
> >
> > [4] highlights the dynamic sparsity inherent in attention mechanisms, noting that most key-value pairs have negligible impact on the output while a small subset of critical pairs contribute significantly. Leveraging this sparsity, the authors employ Approximate Nearest Neighbor Search (ANNS) indexes to store KV vectors in CPU memory, retrieving only the most relevant pairs during generation instead of processing all pairs, thereby reducing GPU memory usage.
> >
> > [5] proposes a hybrid attention architecture, where one-third of the layers utilize full attention mechanisms and the remaining two-thirds adopt sparse attention, enabling the model to access all tokens while substantially reducing computational costs.
> >
> > [6] introduces a small auxiliary model to handle prompt processing, generating an approximate KV cache for the base model. This strategy avoids redundant computations and reduces overall processing overhead during autoregressive generation.
> >
> > [7] reduces computational costs by clustering fixed-context keys offline with K-means, representing them as centroids. During inference, relevant keys are dynamically selected through centroid comparison, with hierarchical lookup further reducing complexity from linear to logarithmic.

---

> > > ### Author Response · Authors · 2024-12-02
> > >
> > > ## Conclusion
> > >
> > > Naively implementing KV cache eviction in vLLM can introduce fragmentation due to the original design of the KV cache. In this design, each block stores keys and values generated for all attention heads across all tokens in the block. In PyramidKV, fragmentation arises because the number of keys and values per head differs between layers. This issue can be mitigated by redesigning the cache kernel so that each block contains KV pairs for a single head instead of for all heads. This may make the number of blocks num_layer×num_head times larger, which would result in scheduling overhead. The scheduling overhead could be reduced by moving the scheduling operations to the GPU. The overall overhead from the redesign would result in a 15% slowdown the scheduling overhead accounts for most of the overhead.

---

> > > > ### Author Response · Authors · 2024-12-02
> > > >
> > > > # vLLM implementation
> > > >
> > > > We provide the settings and scripts to evaluate the throughput of PyramidKV’s vLLM implementation at [a github repo](https://github.com/a-anonymous-user/vLLM-KV) from an anonymous account.
> > > >
> > > > During our development process, we gained valuable insights from [8][9]. [8] presents a vLLM implementation in [flash-attn.py](https://github.com/microsoft/MInference/blob/7a3e5acaaf0e83105d941a4067f53020ca1eba12/minference/modules/minference_forward.py#L812) to perform attention calculations for each attention head and data instance. At the end of the pre-filling stage, [8] profiles the attention matrix and selects a hybrid methods of eviction based on the profiling results. [9] presents a working code example for kv cache compression at vLLM.
> > > >
> > > > To reproduce the throughout results at single-GPU setting, we kindly suggest the reviewer to do the following steps:
> > > > 1. clone the github repo
> > > > 2. Use pip install -e . to compile the repo
> > > > 3. Use the [script for baseline (original vllm)](https://github.com/a-anonymous-user/vLLM-KV/blob/main/test_scripts/test_benchmark_baseline.sh) and the [script for KV cache eviction](https://github.com/a-anonymous-user/vLLM-KV/blob/main/test_scripts/test_benchmark_kv_eviction.sh) to reproduce the results.
> > > >
> > > > The settings are listed in the scripts, and we further clarify as below:
> > > > 1. **Model**: [gradientai/Llama-3-8B-Instruct-262k](https://huggingface.co/gradientai/Llama-3-8B-Instruct-262k)
> > > > 2. **compression rate**: 64, which can be changed at the [compression-rate](https://github.com/a-anonymous-user/vLLM-KV/blob/main/test_scripts/test_benchmark.py#L587) hyper-parameter. This means that the average KV cache budget across layers would be the input_length // compression_rate
> > > > 3. **Input Sequence Length**: from 1k to 10k, which can be changed at the [input-len](https://github.com/a-anonymous-user/vLLM-KV/blob/main/test_scripts/test_benchmark.py#L344) hyper-parameter
> > > > 4. **Total number of samples**: 256, which can be changed at the [num-prompts](https://github.com/a-anonymous-user/vLLM-KV/blob/main/test_scripts/test_benchmark.py#L365) hyper-parameter
> > > > 5. **max batch size**: 256, which can be changed at the [max-batch-size](https://github.com/a-anonymous-user/vLLM-KV/blob/main/test_scripts/test_benchmark.py#L500) hyper-parameter
> > > >
> > > > However, due to time constraints, our multi-GPU (i.e., tensor parallelism and pipeline parallelism) implementations still encounter memory issues in certain cases. Despite significant efforts, we have not been able to fully resolve these issues before the end of the rebuttal period. We are actively working to address these issues and are committed to sharing a stable version. We will release our implementation as an open-source multi-GPU solution for PyramidKV with vLLM.
> > > >
> > > > To connect the cache redesign discussed in the previous section with its implementation, we recommend that reviewers refer to the following files:
> > > >
> > > > **New KV Cache Structure to Prevent Fragmentation**: Please kindly refer to this [link](https://github.com/a-anonymous-user/vLLM-KV/blob/main/csrc/kvcompress_cache_kernels.cu), which implements the new KV cache structure to avoid fragmentation. This file is used during the Flash Attention forward pass from [flash_attn.py](https://github.com/a-anonymous-user/vLLM-KV/blob/main/vllm/attention/backends/flash_attn.py). The kernels are responsible for adding keys to the cache during the Flash Attention forward process.
> > > >
> > > > **Eviction Kernels for the New Cache Structure**: Please kindly refer to this [link](https://github.com/a-anonymous-user/vLLM-KV/blob/main/csrc/kvcompress_eviction_kernels.cu), which provides customized kernels to support the new cache structure, focusing on compression and eviction. These kernels are primarily invoked in [schedule.schedule_compression function](https://github.com/a-anonymous-user/vLLM-KV/blob/main/vllm/kvcompress/scheduler.py#L565) and [Metrics.schedule_evictions](https://github.com/a-anonymous-user/vLLM-KV/blob/main/vllm/kvcompress/metrics.py#L441). The file contains kernels used for eviction and adapting paged attention and caching to the new structure.
> > > >
> > > > **Block Manager**: Please kindly refer to this [link](https://github.com/a-anonymous-user/vLLM-KV/blob/main/vllm/kvcompress/block_manager.py) This fille Manages blocks in the new cache structure to support efficient operations.
> > > >
> > > > **PyramidKV Budget Allocation**: The PyramidKV budget allocation is implemented at [schedule_evictions function](https://github.com/a-anonymous-user/vLLM-KV/blob/main/kvcompress/metrics.py#L441) If we set PyramidKV=True, the budget allocation across layers will be adjusted as our proposed method.

---

> > > > > ### Author Response · Authors · 2024-12-02
> > > > >
> > > > > # References
> > > > >
> > > > > [1] Efficient Memory Management for Large Language Model Serving with PagedAttention
> > > > >
> > > > > [2] Post-Training Sparse Attention with Double Sparsity
> > > > >
> > > > > [3] S2-Attention: Hardware-Aware Context Sharding Among Attention Heads
> > > > >
> > > > > [4] RetrievalAttention: Accelerating Long-Context LLM Inference via Vector Retrieval
> > > > >
> > > > > [5] A little goes a long way: Efficient long context training and inference with partial contexts
> > > > >
> > > > > [6] KV Prediction for Improved Time to First Token
> > > > >
> > > > > [7] Squeezed Attention: Accelerating Long Context Length LLM Inference
> > > > >
> > > > > [8] MInference 1.0: Accelerating Pre-filling for Long-Context LLMs via Dynamic Sparse Attention
> > > > >
> > > > > [9] KV-Compress: Paged KV-Cache Compression with Variable Compression Rates per Attention Head
> > > > >
> > > > > Since we are currently unable to update the paper, we will include the aforementioned works as references in the discussion and related work sections, and cite them in the revised version.
> > > > >
> > > > >
> > > > > # Sincere Acknowledgments to the reviewer MCVv
> > > > >
> > > > > We sincerely appreciate the reviewer’s insightful questions and constructive feedback on our work. The reviewer’s insight have highlighted critical areas for improving our work and have provided valuable guidance for enhancing the practical relevance of our method.
> > > > >
> > > > > Specifically, we agree that limiting our implementation to Hugging Face's Transformers framework may weaken the impact of our method. Implementing our method on a serving framework such as vLLM indeed offers a broader and more compelling use case,.
> > > > >
> > > > > Furthermore, the reviewer’s suggestion to discuss the alignment between KV cache eviction works and memory management systems has illuminated a limitation of our proposed method. As mentioned discussed in the previous sections, without a re-design of the cache implementation in vLLM, issues such as fragmentation may arise. We recognize that this is an important issue to address, and we agree that incorporating a discussion on this topic will significantly enhance the quality of our work.
> > > > >
> > > > > While we are currently unable to update the paper, we are committed to revising the main body our paper to reflect the discussions and improvements inspired by your feedback. These updates will greatly enhance the impact of our work.
> > > > > Thank you once again for your invaluable feedback. Your comments have been instrumental in shaping a stronger and more comprehensive presentation of our research.

---

### Official Review · Reviewer_A1NK · 2024-11-01

**Soundness:** 2
**Presentation:** 3
**Contribution:** 2
**Rating:** 5
**Confidence:** 4

**Summary:**

This paper explores the mechanism by which Large Language Models (LLMs) handle the aggregation of information across different layers, identifying a pattern termed Pyramidal Information Funneling. The authors observe that lower layers in LLMs tend to distribute attention scores more uniformly across all tokens, whereas higher layers exhibit more peaked attention distributions. Based on this observation, the authors propose a strategy for Key-Value (KV) cache allocation where more cache budget is allocated to lower layers, while higher layers receive a reduced budget.

**Strengths:**

* The paper is well-written, easy  to follow and understand the experimental setup and results.
* The paper provides a thorough experimental evaluation, showcasing various baselines and scenarios.

**Weaknesses:**

Lack of Novelty: The main contribution of the paper, i.e., allocating a higher KV cache budget to lower layers and a smaller budget to higher layers, is not entirely new. Similar observations have already been made in prior work, such as [1], which also discussed a linearly decreasing budget allocation across layers and yielded comparable conclusions.

[1] PyramidInfer: Pyramid KV Cache Compression for High-throughput LLM Inference (https://arxiv.org/pdf/2405.12532)

**Questions:**

1. The experiments indicate that the proposed pyramid KV cache allocation does not consistently surpass other baselines across all tasks. Do the authors have insights into which types of tasks this pyramidal allocation performs best and where it tends to underperform?

2. The authors propose a linear distribution for the KV cache budget, defined as k0 = 2 * k_total / m and k_{m-1} = k_total / (beta * m) - k0. My question is: by summing the budgets across all layers and setting this sum equal to the total budget, can beta be directly calculated instead of being treated as a hyperparameter? Is there a misunderstanding in my interpretation of this allocation scheme?

3. Was the decision to use a linear allocation strategy for the pyramid KV cache budget empirically validated as the most effective approach? Did the authors conduct experiments comparing various pyramidal allocation strategies to confirm that a linear strategy is indeed optimal or preferable? Including insights from such comparisons would strengthen the rationale for choosing this specific allocation method.

---

> ### Author Response · Authors · 2024-11-23
>
> We thank the reviewer for the helpful suggestions, and we would like to clarify a few points.
>
> # Responses to Weakness
>
> **W1**: Lack of Novelty: The main contribution of the paper [1]
>
> **R1**:
>
> Based on the ICLR review guidelines (https://iclr.cc/Conferences/2025/ReviewerGuide) that papers are contemporaneous if they are published within the last four months, we regard [1] as a concurrent work.
>
> Our work differs from PyramidInfer in three key aspects:
>
> 1. **Decay Strategy**: While PyramidInfer employs a geometric decay strategy, our method adopts an arithmetic decay strategy. We argue that the relatively stable and linear nature of arithmetic decay better aligns with the behavior of the attention mechanism. This strategy is derived from empirically observed attention patterns, aiming to closely match them. Notably, our approach also achieves superior results, as demonstrated in the experimental results presented in the table below.
>
> 2. **Token Selection**: PyramidInfer discards tokens in earlier layers, preventing them from being reconsidered in later layers. In contrast, our method allows previously discarded tokens to be re-evaluated in higher layers, recognizing that these tokens may still hold relevance at different stages of the model's processing.
>
> 3. **Pyramidal Information Funneling Pattern**: A key contribution of our work lies in identifying and leveraging the pyramidal information funneling phenomenon within attention mechanisms. Through in-depth analysis, we observe that attention tends to disperse in earlier layers and progressively concentrates on crucial tokens in higher layers. This insight forms the foundation of our arithmetic decay strategy, ensuring that our method aligns more naturally with these intrinsic patterns.
>
> Despite some similarities between the two approaches, these differences lead to significantly distinct outcomes. As shown in the table below, our method consistently outperforms PyramidInfer at 64 KV cache size, highlighting the effectiveness of our design choices. We have included these differences in Appendix J.
>
>
> | Method | Avg. |
> | --- | --- |
> | PyramidInfer | 33.94 |
> | PyramidKV | 34.76 |
>
> # Responses to Questions
>
> **Q1**: The experiments indicate that the proposed pyramid KV cache allocation does not consistently surpass other baselines across all tasks. Do the authors have insights into which types of tasks this pyramidal allocation performs best and where it tends to underperform?
>
> **A1**:
>
> 1. Among the 16 datasets, the tasks where our proposed method performs slightly worse than the baseline are mostly saturated (e.g., HotpotQA, Musique, etc. under the LlaMa-3-8B-Instruct setting with KV Size = 64, as shown in Table 1). In these cases, our method is only marginally inferior to the baseline and remains competitive.
>
> 2. Conversely, on tasks with greater potential for improvement (e.g., Qasper, MF-en, TREC, TriviaQA, etc. under the same setting), our method significantly outperforms the baseline. Consequently, the overall average performance of our method surpasses that of the baselines.
>
> 3. Notably, these tasks include several In-Context Learning (ICL) tasks (i.e., TREC), our method enjoys the best performance gain at ICL tasks, where the method demonstrates the ability to leverage provided examples to adapt effectively. The importance of ICL tasks lies in their widespread use in real-world applications, where dynamic adaptation to new inputs is critical.
>
>
> **Q2**: By summing the budgets across all layers and setting this sum equal to the total budget, can beta be directly calculated instead of being treated as a hyperparameter? Is there a misunderstanding in my interpretation of this allocation scheme?
>
> **A2**:
>
> 1. The parameter $\beta$ is still required to determine the top layer.
> 2. Once the top layer is identified, the budget of the bottom layer can be calculated by summing the budgets across all layers and equating this sum to the total budget.
> 3. Finally, the step size can be determined.

---

> > ### Author Response · Authors · 2024-11-23
> >
> > **Q3**: Was the decision to use a linear allocation strategy for the pyramid KV cache budget empirically validated as the most effective approach? Did the authors conduct experiments comparing various pyramidal allocation strategies to confirm that a linear strategy is indeed optimal or preferable? Including insights from such comparisons would strengthen the rationale for choosing this specific allocation method.
> >
> > **A3**:
> >
> > 1. **Pattern Matching**: Based on our observations of the attention pattern, we find that a relatively stable, linear arithmetic decrease aligns more closely with the underlying structure of the pattern.
> >
> > 2. **Empirical Results**: We compared the linear allocation strategy against geometric and exponential decay strategies for pyramid KV cache budget allocation, using a cache size of 64. The empirical results consistently showed that the linear strategy outperformed its counterparts, establishing it as the most effective approach for our use case. We have included this ablation study in Appendix H.1.
> >
> >
> > | Method | Avg. |
> > | --- | --- |
> > | Linear decay Strategy | 34.76 |
> > | Geometric Decay Strategy | 34.36 |
> > | exponential decay Strategy | 34.23 |
> >
> >
> >
> >
> > [1] PyramidInfer: Pyramid KV Cache Compression for High-throughput LLM Inference (https://arxiv.org/pdf/2405.12532)

---

> > > ### Comment · Reviewer_A1NK · 2024-11-25
> > >
> > > Thanks for the clarification and new experiments.  I will increase my score to 5.

---

> ### Author Response · Authors · 2024-11-25
>
> We would like to thank the reviewer again for the valuable feedback, which can significantly improve our work!
> We are glad that our responses address the concerns.
> If there are any remaining concerns, we would really appreciate the opportunity to discuss them further during the discussion period and address them.

---

> > ### Author Response · Authors · 2024-12-03
> >
> > Dear Reviewer A1NK,
> >
> > Thank you for your thoughtful and constructive feedback. We deeply appreciate the time and effort you dedicated to evaluating our work. We are particularly encouraged that our responses during the rebuttal period effectively addressed your concerns regarding novelty, experimental results, and strategy decision-making.
> >
> > In light of this, we are hopeful that our revisions have adequately resolved the key issues raised in your comments, specifically weaknesses 1 and questions. However, we are sincerely disheartened to see that the overall rating for our work remains below the acceptance threshold (rating: 5; contribution: 2; soundness: 2).
> >
> > We would be extremely grateful if you could share additional insights or specific reasons behind this decision. Your perspective would be invaluable to us as we strive to further improve our work and address any lingering concerns.
> >
> > Thank you once again for your time and consideration.
> >
> > Sincerely,
> >
> > The Authors of Paper

---

### Official Review · Reviewer_UuVr · 2024-11-02

**Soundness:** 3
**Presentation:** 3
**Contribution:** 2
**Rating:** 6
**Confidence:** 4

**Summary:**

When LLMs process long texts for inference, KV Cache becomes the main bottleneck. The purpose of this paper is to reduce the GPU memory usage and computation required for KV Cache. PyramidKV is introduced as a novel approach, varying cache sizes across layers based on information flow patterns. It allocates more cache to lower layers, where information is dispersed, and less to higher layers, where it's concentrated. Experiments on LongBench demonstrate that PyramidKV maintains performance with only 12% of the KV cache and excels in extreme conditions, even with just 0.7% cache.

**Strengths:**

1）The paper analyzes Attention data from different layers of LLM and discovers that LLMs aggregate information through Pyramidal Information Funneling patterns.
2）The paper is the first to propose an algorithm using different compression rates for KV Cache at different layers, which can be used with other KV Cache algorithms.
3）In scenarios with extremely high KV Cache compression rates(like 99.3%), this method can achieve better accuracy compared to other existing algorithm.

**Weaknesses:**

1）When the KV budget is retained at 2k, the accuracy of the proposed method does not show significant advantages.
2）The paper mainly tests models with an 8k context length, lacking accuracy tests for models with sequence lengths above 128k.
3）In cases of extremely low compression ratios, it is recommended to include comparisons with new technologies such as Minference.

**Questions:**

1）Besides Llama-like models, do other models also exhibit the Pyramidal Information Funneling phenomenon?
2）When determining the KV Cache budget for different layers, how should hyperparameters be selected to ensure optimal accuracy?

---

> ### Author Response · Authors · 2024-11-23
>
> We thank the reviewer for the suggestions, and we would like to clarify a few points.
>
> # Responses to Weakness
>
> **W1**: When the KV budget is retained at 2k, the accuracy of the proposed method does not show significant advantages.
>
> **A1**:
>
> 1. With a small budget, our proposed method's effective allocation strategy better preserves useful attention information. However, with a larger budget, this allocation becomes less critical for covering the necessary information.
>
> 2. To further illustrate this phenomenon, we have included an ablation study titled "Attention Recall Rate Experiment" in Figure 8 from Appendix M. The results show that with a small budget (i.e., 64 KV cache size), PyramidKV improves the attention recall rate (the percentage of attention computed using the keys retrieved by the method and the query, relative to the attention computed using all keys and the query.). However, with a larger budget (i.e., 2k KV cache size), the improvement decreases. For 64, 128, 256, 512, 1024 and 2048 KV cache sizes, PyramidKV's average attention recall rate improvements are 1.87%, 0.64%, 0.61%, 0.56%, 0.47%, and 0.36%.
>
> **W2**: The paper mainly tests models with an 8k context length, lacking accuracy tests for models with sequence lengths above 128k.
>
> **A2**:
>
> In response, we conducted additional experiments using Llama-3-8B-Instruct-Gradient-1048k with an input length of 128k. The results, summarized in the table below, showcase the model's average performance at LongBench with extended context lengths. We have included details in Appendix N.
>
>
>
> | Method | Avg. |
> | --- | --- |
> | SnapKV | 26.65 |
> | H2O | 25.20 |
> | StreamingLLM | 24.69 |
> | PyramidKV | 27.48 |
>
> **W3**: In cases of extremely low compression ratios, it is recommended to include comparisons with new technologies such as Minference.
>
> **A3**:
>
> We would like to clarify that PyramidKV and MInference are complementary approaches addressing different aspects of KV cache optimization. Specifically:
>
> 1. MInference focuses on accelerating the generation of KV caches during the pre-filling stage of LLM inference.
>
> 2. In contrast, PyramidKV targets efficient KV cache management during LLM decoding.
>
> To evaluate their respective strengths, we compared PyramidKV and MInference on Longbench using a KV cache size of 128. The results demonstrated the superior performance of PyramidKV.
>
> Furthermore, we demonstrate that MInference and PyramidKV can be seamlessly integrated to achieve highly efficient inference while maintaining performance comparable to full attention. The LongBench average results of MInference combined with PyramidKV, evaluated on Longbench with a KV cache size of 128, are presented below:
>
> | Method | Avg. |
> | --- | --- |
> | MInference | 38.86 |
> | PyramidKV | 39.31 |
> | PyramidKV + MInference hybrid | 40.47 |
>
> In summary, we demonstrate that PyramidKV outperforms MInference on Longbench. Furthermore, when integrated with MInference, PyramidKV enhances its performance even further. We have included details in Appendix I.
>
> # Responses to Questions
>
> **Q1**: Besides Llama-like models, do other models also exhibit the Pyramidal Information Funneling phenomenon?
>
> **A1**:
>
> We further provide additional analysis at Figure 5 and Figure 6 in Appendix C, demonstrating that the Pyramidal Information Funneling phenomenon is also evident in both the Mistral model and the Mixtral mixture-of-experts model. Rather than Llama-like models, Pyramidal Information Funneling phenomenon also exhibit in Mixture-of-Expert models. This supports the universality of the Pyramidal Information Funneling phenomenon across diverse model families. We hope this addresses your concern and underscores the generalizability of our findings.
>
> **Q2**: When determining the KV Cache budget for different layers, how should hyperparameters be selected to ensure optimal accuracy?
>
> **A2**:
>
> We conducted hyperparameter testing on the original development sets of 16 datasets in LongBench. The parameter $\beta $demonstrated remarkable stability, showing minimal sensitivity to varying hyperparameter settings, which highlights its robustness. Conversely, $\alpha$ consistently produced superior results when set to 8 or 16. Consequently, these values were adopted for subsequent experiments.
>
> In Appendix H.2 and H.3, we further analyzed the impact of hyperparameter selection on KV cache budget allocation across different layers. The experiments reaffirmed that $\beta$  had negligible influence on the outcomes, underscoring its stability. Meanwhile, \alphaα continued to deliver optimal results at values of 8 and 16.

---

> > ### Author Response · Authors · 2024-11-25
> >
> > We sincerely thank the reviewer once again for the insightful feedback. We hope you find our response compelling enough to consider a re-evaluation of our score. If there are any remaining concerns, we would really appreciate the opportunity to discuss them further during the discussion period.

---

> > > ### Author Response · Authors · 2024-11-28
> > >
> > > Dear Reviewer UuVr,
> > >
> > > We would like to know if our response has addressed your concerns and questions. If you have any additional suggestions regarding the paper or our reply, please let us know. We’re happy to make further improvements.
> > >
> > > Thank you again for your time and effort!

---

> > > > ### Author Response · Authors · 2024-12-03
> > > >
> > > > Dear Reviewer UuVr,
> > > >
> > > > As the discussion period is nearing its conclusion, we kindly ask if our response has sufficiently addressed your concerns and questions. Should you have any additional inquiries, we are more than happy to provide further clarification.
> > > >
> > > > Thank you once again for your time and thoughtful feedback!

---

### Official Review · Reviewer_pUzw · 2024-11-02

**Soundness:** 3
**Presentation:** 3
**Contribution:** 3
**Rating:** 6
**Confidence:** 4

**Summary:**

The study proposes a new KV cache compression method, called PyramidKV, based on its empirical study on the pyramid pattern of attention scores across layers of language models. The proposed method dynamically allocate KV cache budget to each layer based on the identified pyramid pattern. PyramidKV shows superior performance, especially under resource-intensive circumstances, against other baselines.

**Strengths:**

1. The observation on the pyramid pattern of attention scores across layers is valuable.
2. Based on the observed pattern, the proposed method is straight-forward and performant under resource-intensive circumstances.
3. The experiment is comprehensive.

**Weaknesses:**

1. The proposed method works really well under extreme condition, i.e.e KV cache size = 128. However, under not-so-extreme cases, i.e. KV cache size = 2048, the performance is not comparable to other baselines according to Table 1 in the paper. Is there any explanation to this phenomenon? I think the paper worth a small section of ablation study to explain this phenomenon.
2. In [1], Wu et al. claims that "retrieval heads" exist across models, functioning similarly to the submission's patterns ("massive attention") seen in higher layers—such as layer 30 in Figure 2—to retrieve essential contextual information. Given this, I wonder if retrieval heads are primarily found in the higher layers or if they might also be present in lower layers but are obscured due to the study's averaging of attention scores across heads within each layer. This averaging might be masking the presence of "massive attention" in the lower layers, leading to more-than-enough allocation of KV cache for some heads in the lower layers.  Could the authors conduct additional experiments to address my concern?
3. There are many variations of NIAH tasks, e.g. haystack formed from repetitive sentences or haystack formed from a long corpus. Can the authors elaborate which setting used in the study?

[1] Wenhao Wu, Yizhong Wang, Guangxuan Xiao, Hao Peng, and Yao Fu. Retrieval head mechanistically explains long-context factuality. arXiv preprint arXiv:2404.15574, 2024.

**Questions:**

Please see weaknesses.

---

> ### Author Response · Authors · 2024-11-23
>
> We thank the reviewer for the helpful suggestions, and we would like to clarify a few points.
>
> **W1**: The proposed method works really well under extreme condition. However, under not-so-extreme cases, the performance is not comparable to other baselines according to Table 1 in the paper.
> Is there any explanation to this phenomenon? I think the paper worth a small section of ablation study to explain this phenomenon.
>
> **A1**:
>
> Thank you for your suggestions and for bringing this observation to our attention. While it may appear that the advantages of our method are less pronounced in moderate cases compared to extreme cases, we would like to highlight two key aspects of our results.
>
> 1. Firstly, with a small budget, our proposed method enables more effective allocation, better preserving useful attention information. Second, with a large budget, such allocation becomes less critical, as it is sufficient to cover the necessary information.
>
> 2. To further illustrate this phenomenon, we have included an ablation study titled "Attention Recall Rate Experiment" in Figure 8 from Appendix M of the revised version. The results show that with a small budget (i.e., 64 KV cache size), PyramidKV improves the attention recall rate (the percentage of attention computed using the keys retrieved by the method and the query, relative to the attention computed using all keys and the query.). However, with a larger budget (i.e., 2k KV cache size), the improvement decreases. For 64, 128, 256, 512, 1024, and 2048 KV cache sizes, PyramidKV's average attention recall rate improvements are 1.87%, 0.64%, 0.61%, 0.56%, 0.47%, and 0.36%.
>
> **W2**: In [1], Wu et al. claims that "retrieval heads" exist across models, functioning similarly to the submission's patterns ("massive attention") seen in higher layers—such as layer 30 in Figure 2—to retrieve essential contextual information.
> Given this, I wonder if retrieval heads are primarily found in the higher layers or if they might also be present in lower layers but are obscured due to the study's averaging of attention scores across heads within each layer.
> This averaging might be masking the presence of "massive attention" in the lower layers, leading to more-than-enough allocation of KV cache for some heads in the lower layers. Could the authors conduct additional experiments to address my concern?
>
> **A2**:
>
> 1. Retrieval heads are predominantly located in the higher layers, as illustrated in Figure 5 of [1]. Notably, no retrieval heads are observed in layers 0–8 of the LLaMA models.
>
> 2. To further investigate, we conducted additional experiments on the lower layers to analyze the attention patterns of the heads, as detailed in Appendix P. Our findings indicate the absence of "massive attention" in any individual head.
>
> **W3**: There are many variations of NIAH tasks, e.g. haystack formed from repetitive sentences or haystack formed from a long corpus. Can the authors elaborate which setting used in the study?
>
> **A3**:
>
> We adopt the haystack setting of haystack formed from a long corpus for the NIAH task. This follows the exact setup of [1]. We have included the details in Section 5.3.1.
>
> [1] Wenhao Wu, Yizhong Wang, Guangxuan Xiao, Hao Peng, and Yao Fu. Retrieval head mechanistically explains long-context factuality. arXiv preprint arXiv:2404.15574, 2024.

---

> > ### Comment · Reviewer_pUzw · 2024-11-24
> >
> > Thanks for the rebuttal, and I'm satisfied with it, so I will maintain my score.

---

> > > ### Author Response · Authors · 2024-11-25
> > >
> > > We are glad that you are satisfied with our response! If our response addressed your concerns, would you like to consider updating the review rating accordingly?

---

> > > > ### Author Response · Authors · 2024-11-28
> > > >
> > > > Dear reviewer pUzw,
> > > >
> > > > We’re delighted to hear that our response has been helpful! If you feel that we’ve addressed your concerns and you are satisfied with the response, we would sincerely appreciate it if you could consider updating your review rating accordingly. Thank you for your support!
> > > >
> > > > Thank you again for your time and effort to help us improve our work!

---

> > > > > ### Author Response · Authors · 2024-12-03
> > > > >
> > > > > Dear Reviewer pUzw,
> > > > >
> > > > > Thank you for taking the time to volunteer as a reviewer and review our response. We’re glad to hear that it has been helpful in addressing your concerns. If you believe we’ve successfully resolved the issues raised, we would greatly appreciate it if you could consider reflecting this in your review rating.
> > > > >
> > > > > Since the discussion period is ending soon, if you have any additional suggestions regarding the paper or our reply, please let us know. We’re happy to make further improvements.
> > > > >
> > > > > We are deeply grateful for your valuable feedback and support, which have been instrumental in improving our work.

---

### Official Review · Reviewer_ieep · 2024-11-04

**Soundness:** 2
**Presentation:** 1
**Contribution:** 1
**Rating:** 3
**Confidence:** 4

**Summary:**

This paper proposes PyramidKV to conduct KV cache compression for LLM inference. The insight is that the attention scores are more uniform in the first layers but become more skewed in the last layers. As such, PyramidKV selects more tokens for the first layers and fewer tokens for the final layers. Experiments are conducted on two sets of tasks.

**Strengths:**

1.	Key cache compression is an important topic.

2.	The idea of PyramidKV is explained clearly.

**Weaknesses:**

1.	The observation that the attention scores are more uniform in the first layers but become more skewed in the last layers is NOT new, see [1][2] for example. With the observation, it is straightforward to extend existing KV cache selection methods to use different sampling ratios for different layers. This limits the novelty of the paper.

[1] InfiniGen: Efficient Generative Inference of Large Language Models with Dynamic KV Cache Management
[2] MagicPIG: LSH Sampling for Efficient LLM Generation

2.	The choice of sampling rate schedule, i.e., how many tokens to sample for each layer, is not discussed clearly? Why use an arithmetic sequence? What are the observations driving this choice? Will also schedule also work?

3.	The empirical results are not impressive. As shown in Table 1, PyramidKV does not outperform the baselines in many cases.

4.	The presentation needs to be significantly improved. (a) Most figures are not vector illustrations and become blurred when enlarged. Figure 3 is repetitive w.r.t. to Figure 1, where the idea of PyramidKV is already illustrated. (b) Figure 2 is difficult to understand, to show the skewness of attention scores, histogram or CDF (see [1][2]) may be used. (c) Section 5 is partitioned into too many subsections. You can present the experiment settings in one subsection, the main experiment result in one subsection, and some insight experiment in one subsection. (d) I failed to understand what the grids mean in Figure 5, and the axis is too small to read. (e) What is the right half of Table 2 reporting?

**Questions:**

See weakness part

---

> ### Author Response · Authors · 2024-11-23
>
> We thank the reviewer for constructive advices, and we would like to clarify a few points.
>
> **W1**: The observation that the attention scores are more uniform in the first layers but become more skewed in the last layers is NOT new, see [1][2].
>
> [1] InfiniGen: Efficient Generative Inference of Large Language Models with Dynamic KV Cache Management [2] MagicPIG: LSH Sampling for Efficient LLM Generation
>
> **R1**:
> 1. We respectfully refer the reviewer to the ICLR review guideline (https://iclr.cc/Conferences/2025/ReviewerGuide) regarding comparisons to papers published within four months. Papers  [1] and [2] fall within this timeframe and are considered contemporaneous.
>
> 2. However, we would like to emphasize the key distinctions between our findings and theirs. While [1] and [2] observe a general trend where attention scores are uniform in the initial layers and become increasingly skewed in the later layers, our analysis uncovers a more intricate evolution of attention patterns across the model's layers. Specifically, our findings reveal:
>
> - **Lower Layers**: The model operates in a broad-spectrum mode, uniformly distributing attention across the input to capture diverse information.
>
> - **Middle Layers**: Attention narrows its focus on individual components (See Figure 2. Layers 6 and 12), reflecting a process of refined information aggregation.
>
> - **Upper Layers**: Attention transitions into a "massive attention" mechanism, heavily concentrating on a few critical tokens to extract key information efficiently for answer generation.
>
> This nuanced perspective provides a clearer and more comprehensive understanding of how attention dynamics evolve throughout the model's layers, highlighting the unique insights offered by our work.
>
> **W2**: The choice of sampling rate schedule, i.e., how many tokens to sample for each layer, is not discussed clearly? Why use an arithmetic sequence? What are the observations driving this choice? Will also schedule also work?
>
> **R2**:
>
> 1. **The choice of sampling rate schedule**: For the sampling schedule, the number of tokens allocated to each layer is determined based on the hyperparameter $\beta$. Once the cache budget and $\beta$ are set, the token count for each layer can be calculated using Eq. (1).
>
> 2. **Why an Arithmetic Sequence? Observations Driving this Choice**: Our decision to use an arithmetic sequence is driven by three key factors:
>
> - **Alignment with Pyramidal Information Funneling Pattern**:
> Empirical observations reveal a pyramidal information funneling pattern, where lower layers exhibit dispersed attention while higher layers concentrate on fewer tokens. Inspired by this, we adopt the arithmetic sequence design to align with this natural progression.
>
> - **Superior Empirical Performance**:
> Through extensive experimentation across diverse datasets, we compared various methods, including the arithmetic sequence and adaptive approaches. Results consistently showed that the arithmetic sequence method outperformed others.
>
> - **Computational Efficiency**:
> The arithmetic sequence method introduces minimal computational overhead compared to adaptive approaches, which require dynamically computing cache budgets across layers.
>
> **W3**: The empirical results are not impressive. As shown in Table 1, PyramidKV does not outperform the baselines in many cases.
>
> **R3**:
>
> 1. Among the 16 datasets, the tasks where our proposed method performs slightly worse than the baseline (i.e., HotpotQA, Musique, SAMSum, and PCount under the LLaMA-3-8B-Instruct, KV Size = 64 setting, as shown in Table 1) are actually saturated. In these cases, our method is only marginally inferior to the baseline and remains highly competitive.
>
> 2. Conversely, on tasks with more room for improvement (e.g., Qasper, MF-en, TREC, and TriviaQA under the same LLaMA-3-8B-Instruct, KV Size = 64 setting), our method significantly outperforms the baseline. Consequently, the average performance of our method is substantially better than the baselines.
>
> 3. Notably, these tasks include several In-Context Learning (ICL) tasks (i.e., TREC), our method enjoys the best performance gain at ICL tasks, where the method demonstrates the ability to leverage provided examples to adapt effectively. The importance of ICL tasks lies in their widespread use in real-world applications, where dynamic adaptation to new inputs is critical.

---

> > ### Author Response · Authors · 2024-11-23
> >
> > **W4**: The presentation needs to be significantly improved.
> >
> > **R4**:
> >
> > Thank you for your valuable suggestions. We have made the corresponding revisions based on your feedback.
> >
> > 1. The figures have been updated to vector illustrations, and Figure 3 has been removed from the main body of the paper.
> >
> > 2. We have added a figure of attention recall rates at Figure 8 from Appendix M. Figure 8 shows the skewness of attention scores and attention recall rates (the percentage of attention computed using the keys retrieved by the method and the query, relative to the attention computed using all keys and the query.), which may help understand our motivation and the superiority of the proposed method.
> >
> > 3. Section 5 has been reorganized into three subsections: Experimental Setup, Main Results, and Other Insight Experiments.
> >
> > 4. Figure 4 (previous Figure 5 for NIAH evaluation results) is now a larger figure for better clarity. The color of each cell in the grid represents the score. Green indicates a high score, while red indicates a low score. The x-axis represents the input length, and the y-axis represents the position where important information is inserted into the input.
> >
> > 5. The right half of Table 2 reports the performance results for specific tasks.
> >
> > We appreciate your insightful comments, which have helped improve the quality of our work.

---

> > > ### Comment · Reviewer_ieep · 2024-11-25
> > > **After the author response**
> > >
> > > I have read the author response. I agree that the novelty of the paper is not affected by recently published papers. However, I still think the solution is overly simple, the experiment results are not good, and the presentation is not carefully polished.
> > >
> > > I have some questions regarding the author response.
> > >
> > > (1) In the reponse of R1, why the observation is " clearer and more comprehensive" than existing works? The three points basically say the same thing as existing work, i.e., the attention is uniform in the initial layers but more skewed in latter layers.
> > >
> > > (2) The explantion of using the arithmetic sequence does not provide real content. "pyramidal information funneling pattern, where lower layers exhibit dispersed attention while higher layers concentrate on fewer tokens", any sequence with a dimishing trend matches this pattern, why an arithmetic sequence? To support using arithmetic sequence, detailed profiling needs to show that to reach for instance, 80% of the full attention weightm, the number of required top tokens really resembles an arithmetic sequence. Moreover, why do " adaptive approaches" acutually mean in the author response? How do they work? Regarding the computation efficency, any algebra computation with reasonable computation will be much faster than the actual attention computation, which will not make attention weight scheduling the bottleneck. This is NOT the reason to choose the arithmetic sequence.

---

> ### Author Response · Authors · 2024-11-28
>
> We thank the reviewer for the insightful questions. And we would like to clarify a few points.
>
> # Response to Weakness
>
> **W1**: I still think the solution is overly simple, the experiment results are not good, and the presentation is not carefully polished
>
> **A1**:
>
> **Response to the solution is overly simple:**
>
> Our work focuses on developing simple, efficient, and effective methods that are easily applicable to a variety of models and adaptable to diverse tasks. The simplicity and effectiveness of our approach represent its unique strength, as it can be seamlessly integrated with pre-filling stage acceleration methods (e.g., MInference, as noted by Reviewer `UuVr` ) and serving frameworks (e.g., vLLM, as highlighted by Reviewer `MCVv` ). In contrast, adaptive methods that require additional computation may lack this versatility and ease of integration.
>
> **Response to the experiment results are not good**:
>
> We claim that our method demonstrates generally superior performance compared to the baselines, particularly in extreme scenarios. This strong performance has been highlighted as a strength of our work by reviewers `MCVv` , `A1NK` , `UuVr` , and `pUzw`. However, we understand the reviewer's concern that our method may underperform relative to baselines on some datasets. To address this, we clarify the following points:
>
> - Among the 16 datasets, the few instances where our method slightly underperforms the baselines typically occur in saturated tasks (e.g., HotpotQA, Musique, etc., under the LLaMA-3-8B-Instruct setting with KV Size = 64, as shown in Table 1). In these cases, the margin of difference is minimal, and our method remains competitive.
>
> - Conversely, on tasks with greater potential for improvement (e.g., Qasper, MF-en, TREC, TriviaQA, etc., under the same setting), our method achieves significantly better results than the baselines.
>
> - As a result, the overall average performance of our method significantly exceeds that of the baselines, especially in extreme scenarios.
>
> **Response to the presentation is not carefully polished**:
>
> Regarding the reviewer’s feedback about the presentation not being polished, we have carefully revised the paper to address the concerns that the reviewer has about the presentation of our paper. And we would like to sincerely ask the reviewer to further point out the specific issues that are unclear. We are eager to make corresponding improvements. We value the reviewer’s feedback and appreciate the opportunity to improve our work.

---

> ### Author Response · Authors · 2024-11-28
>
> # Response to Questions
>
> **Q1**: In the reponse of R1, why the observation is " clearer and more comprehensive" than existing works? The three points basically say the same thing as existing work, i.e., the attention is uniform in the initial layers but more skewed in latter layers.
>
> **A1**:
>
> We believe that [1] and [2] are both high-quality works on this topic. Despite that [1] and [2] are considered contemporaneous with our work, we humbly assert that our work remains novel and more comprehensive than other studies. Our analysis uniquely examines attention metrics across all transformer layers, from 0 to 30, leading to the discovery of a key phenomenon we term Pyramidal Information Funneling.
>
> [1] conducted a limited investigation into attention patterns, focusing only on the lower layer (layer 0) and a single upper layer (layer 18). While [1] noted that attention becomes more skewed in upper layers, it did not provide a fine-grained observation of attention patterns across all layers. In contrast, our study reveals several novel findings:
>
> - **Localized Attention**: We observe that attention progressively narrows its focus, targeting specific components within the input sequence.
>
> - **Massive Attention Mechanism**: In the upper layers, attention heavily concentrates on a small set of critical tokens. Notably, these tokens are not limited to the leading positions, as observed in [1], but also appear at regular intervals across the sequence. The discrepancy arises from differences in input settings, with [1] identifying massive attention only at the initial tokens.
>
> These insights motivated us to propose a token-selection method based on the highest attention scores in the upper layers, rather than solely relying on tokens from earlier positions.
>
> To the best of our knowledge, [2] has not analyzed attention patterns across transformer layers.
>
> Therefore, although [1] and [2] are considered contemporaneous with our work, making a comparison unnecessary, the perspective of our observation is considered novel compared with [1] and [2]. Moreover, although [1] also observed attention patterns, the method we proposed based on our observations is significantly different from [1], further highlighting the novelty of our work.
>
> In summary, our primary objective is to uncover the flow of information across transformer layers and identify effective compression strategies. By analyzing attention patterns at all levels—lower, middle, and upper—we elucidate localized attention mechanisms and the Pyramidal Information Funneling phenomenon. Our observations partially explain how large language models process information from different components. Additionally, we identify that massive activations in the upper layers occur at diverse positions, driving us to propose the method for selecting tokens with the highest attention scores even in the upper layers.
>
> We have included the above discussion and these two contemporaneous works as citations in Appendix C.
>
> [1] InfiniGen: Efficient Generative Inference of Large Language Models with Dynamic KV Cache Management
>
> [2] MagicPIG: LSH Sampling for Efficient LLM Generation

---

> ### Author Response · Authors · 2024-11-28
>
> **Q2**: The explantion of using the arithmetic sequence does not provide real content, why an arithmetic sequence? Moreover, why do " adaptive approaches" acutually mean in the author response? How do they work? Regarding the computation efficency, any algebra computation with reasonable computation will be much faster than the actual attention computation, which will not make attention weight scheduling the bottleneck. This is NOT the reason to choose the arithmetic sequence.
>
> **A2**:
>
> **Why an arithmetic sequence? **
>
> - **Pattern Matching**: Based on our analysis of the attention patterns, we observe that a relatively stable, linearly decreasing trend aligns closely with the underlying structure of the patterns.
>
> - **Empirical Results**: We compared the arithmetic sequence strategy against logarithmic and exponential sequence approaches for pyramid KV cache budget allocation, using a cache size of 64 as below. The empirical results consistently showed that the arithmetic sequence strategy outperformed its counterparts, establishing it as the most effective approach for our use case.
>
>
> | Method | Avg. |
> | --- | --- |
> | Arithmetic Sequence Strategy | 34.76 |
> | Geometric Sequence Strategy | 34.36 |
> | Exponential Sequence Strategy | 34.23 |
>
> We have included this ablation study in Appendix H.1.
>
> **why do " adaptive approaches" acutually mean in the author response? How do they work?**
>
> We propose two adaptive allocation baselines, which are based on the entropy and Gini coefficient of the attention values at each layer. The weight of each layer is calculated based on its corresponding metric (i.e., entropy and Gini coefficient), and the budget is allocated accordingly. Specifically:
>
> - **Entropy-based allocation**: Layers with higher entropy receive fewer weights. Each layer's entropy is calculated based on the layer's attention.
>
> - **Gini coefficient-based allocation**: Layers with higher Gini coefficients receive fewer weights. Each layer's Gini coefficient is calculated based on the layer's attention.
>
> We compared the arithmetic sequence-based allocation against entropy-based allocation and Gini coefficient-based allocation, using a cache size of 64 as below. The empirical results consistently showed that the arithmetic sequence-based allocation outperformed adaptive methods, establishing it as the most effective approach for our use case.
>
> | Method | Avg. |
> | --- | --- |
> | arithmetic sequence-based allocation | 34.76 |
> | Entropy-based allocation | 32.71 |
> | Gini coefficient-based allocation | 32.58 |
>
> The results show that all of these methods are far less effective than PyramidKV, which uses an arithmetic sequence-based allocation approach. We have included this ablation study in Appendix H.1.
>
> **computation efficency**
>
> Although algebraic computations with reasonable complexity are faster than attention computations, attention itself remains an unavoidable component of the transformer forward pass. Therefore, the critical issue lies in the additional computation introduced by the proposed method rather than the attention computation, which is inherent to the model. We present a detailed computation time analysis for the proposed method compared with adaptive methods in a llama-3-8b model as below. Allocation time means the time needed to compute for budget allocation.
>
>
> |  | Allocation Time |
> | --- | --- |
> | PyramidKV | 0.0000003 |
> | Entropy-based allocation | 0.35 |
> | Gini coefficient-based allocation | 1.13 |
>
> PyramidKV as an arithmetic sequence demonstrates significant efficiency advantages at computation overhead over adaptive methods.
>
> Based on the above analysis, PyramidKV demonstrates both effectiveness and efficiency advantages over adaptive methods.
>
> We sincerely thank the reviewer once again for their valuable feedback, which has greatly contributed to improving the clarity and rigor of our work. We have carefully addressed each of the raised concerns and hope that our clarifications effectively resolve them. We kindly ask you to consider re-evaluating our score based on our responses. If there are any remaining concerns, we would deeply appreciate the opportunity to engage further during the discussion period. Your insights are invaluable to us, and we really appreciate the opportunity to improve our work.

---

> > ### Author Response · Authors · 2024-12-03
> >
> > Dear Reviewer ieep,
> >
> > We would appreciate it if you could let us know whether our response has adequately addressed your concerns and questions. We remain available to address any further questions you may have.
> >
> > Thank you once again for your time and effort!

---

### Author Response · Authors · 2024-12-03

Dear reviewers,

Thank you again for your volunteer service as reviewers and your efforts to propose helpful advices! We have added many revisions to address your concerns. Since the rebuttal period is closing very soon, can you please check the response to see whether it mitigates your concerns? We would greatly appreciate that!

Thank you,

The Authors

---

### Author Response · Authors · 2024-12-04
**General Response to all Reviewers**

Dear reviewers,

We thank all reviewers for their insightful comments and acknowledgment of our contributions. We are glad to hear reviewer  `MCVv`'s **recognition that “the amount of work this paper and quality are well above the ICLR threshold”**, underscoring its significance and potential impact in the field.

We greatly appreciate your recognition of the strengths of our work as follows:

- **Observation of Pyramidal Information Funneling and Proposal of PyramidKV**
  - Our observations reveal that LLMs aggregate information through **Pyramidal Information Funneling, recognized by `pUzw` and `MCVv` as valuable and insightful**.
  - Based on the observed pattern, we propose **PyramidKV, recognized by `UuVr` as a novel method**, which is the first to propose an algorithm using different compression rates for KV Cache at different layers, which can be used with other KV Cache algorithms.
- **Comprehensive Experiments and Superior Performance**
  - **The experiment is recognized by as comprehensive, thorough and through by `pUzw`, `A1NK` and `MCVv`**, showcasing various baselines and scenario, which exams a few challenging tasks in the long context scenarios.
  - The comprehensive experiments are a strong indication that PyramidKV not only **shows superior performance, especially under resource-intensive circumstances, against other baselines** as mentioned by reviewer `UuVr` and `pUzw`; but also **effectively preserves long context capabilities much better compared to methods** in the class as highlighted by reviewer `MCVv`
- **Well-Writing and Good Presentation**
  - The paper recognized is **well-written, easy to follow and understand** the experimental setup and results by `A1NK`.

**We've revised our manuscript per the reviewers' suggestions (highlighted in red in the uploaded revision pdf)**. Detailed responses to each reviewer's concerns are carefully addressed point-by-point.

Below summarize the major updates we've made:

- **Presentation**:
    - `A1NK` (**Section 4.2.1**) Includes additional clarification on the proposed method.
    - `ieep` (**Section 5**) Re-organizes the sections.
    - `ieep` (**Figure 4**)  Includes a larger figure for better clarity.
    - `UuVr` (**Figure 5 and Figure 6 in Appendix C**) Includes the analysis of ** Pyramidal Information Funneling in Mistral model (i.e., Decoder-Only architecture) and Mixtral (i.e., MoE architecture).**
    - `ieep` (**Figure 7**)  Figure 7 is removed from main body to Appendix C.
- **Experiments**:
    - `ieep` and `A1NK` (**Appendix H.1**) Includes **ablation studies on diverse allocation strategies**.
    - `UuVr` (**Appendix I**) Includes **comparison and integration with MInference**.
    - `A1NK` (**Appendix J**) Includes **clarification and empirical comparison between our work and a contemporaneous work.**
    - `pUzw` (**Figure 8**) Includes **a study to explain the results that PyramidKV outperforms more in extreme resource-intensive settings.**
    - `UuVr` (**Appendix N**) Includes an additional experiment with **input length of 128k.**
    - `pUzw` (**Appendix P**) Includes a study showing **the absence of "massive attention" in any individual head from low layers.**
    - `MCVv` (**Appendix Q**) Includes a study shows the **implementation of PyramidKV in vLLM and the efficiency comparison.**
- **Explaination**:
    - `ieep` (**Appendix C**) Includes the **clarification and comparison of the observation and analysis between our work and two contemporaneous works.**
    - `ieep` and `A1NK` (**Section 5.2**) Includes additional explanations of the experiment results.
    - `ieep` (**Appendix D**) Includes illustrations of reasons of the method design.

Specifically, below are the updates that we have not updated in the revised version yet. **We are committed to update the below details in the revised version.**

- vLLM Implementation
  - `MCVv` Adds the **challenges, solutions and limitations of implementing PyramidKV in vLLM.**
  - `MCVv` Adds the **details of PyramidKV’s implementation in vLLM.**

We believe our work could make a novel contribution to the KV cache compression community and offer a novel perspective on leveraging attention patterns to propose effective compression methods.

Best,

Authors.

---

### Meta-Review · Area_Chair_KUvd · 2024-12-21

**Metareview:**

This paper introduces PyramidKV, a KV cache compression method designed to enhance memory efficiency in large language model (LLM) inference. By leveraging an observed pyramidal attention pattern, the authors allocate larger cache sizes to lower layers and progressively smaller sizes to higher layers. Experimental results on the LongBench benchmark demonstrate that PyramidKV achieves comparable or superior performance to models with full KV caches, even at significantly reduced cache sizes. However, the reviewers highlight critical limitations: the novelty of the method is undermined by its similarity to existing works, the experimental results lack consistency in demonstrating superiority across diverse settings, and the explanation for key design choices (e.g., the arithmetic allocation sequence) remains insufficiently validated. Strengths of the submission include a well-motivated application area, a clear exposition of the compression approach, and competitive performance in resource-constrained scenarios. Nonetheless, the lack of broader novelty, limited generalization tests, and weaknesses in presentation weigh against acceptance.

The primary reason for the recommendation to reject this paper is its limited contribution beyond existing works in KV cache optimization. While the method is practical, its design choices are not adequately justified, and its performance advantages are context-specific, lacking robustness across tasks.

**Additional Comments On Reviewer Discussion:**

The reviewers’ concerns about the method’s novelty and experimental consistency remain salient, even after the authors’ rebuttals. While some reviewers acknowledged the authors’ responses, others pointed out unresolved issues, particularly regarding the originality of the approach and justification for specific design choices. Overall, the discussion highlights both the potential and the limitations of the work, suggesting that further refinement and validation would significantly enhance its impact.

---

### Decision · Program_Chairs · 2025-01-22

Reject